# Towards Understanding Cooperative Multi-Agent Q-Learning with Value Factorization

**Jianhao Wang**[1]*, **Zhizhou Ren**[2]*[†], **Beining Han**[1], **Jianing Ye**[1], **Chongjie Zhang**[1]

[1]Institute for Interdisciplinary Information Sciences, Tsinghua University
[2]Department of Computer Science, University of Illinois at Urbana-Champaign
wjh19@mails.tsinghua.edu.cn, zhizhour@illinois.edu
{hbn18, yejn21}@mails.tsinghua.edu.cn
chongjie@tsinghua.edu.cn

## Abstract

Value factorization is a popular and promising approach to scaling up multi-agent reinforcement learning in cooperative settings, which balances the learning scalability and the representational capacity of value functions. However, the theoretical understanding of such methods is limited. In this paper, we formalize a multi-agent fitted Q-iteration framework for analyzing factorized multi-agent Q-learning. Based on this framework, we investigate linear value factorization and reveal that multi-agent Q-learning with this simple decomposition implicitly realizes a powerful counterfactual credit assignment, but may not converge in some settings. Through further analysis, we find that on-policy training or richer joint value function classes can improve its local or global convergence properties, respectively. Finally, to support our theoretical implications in practical realization, we conduct an empirical analysis of state-of-the-art deep multi-agent Q-learning algorithms on didactic examples and a broad set of StarCraft II unit micromanagement tasks.

## 1 Introduction

Cooperative multi-agent reinforcement learning (MARL) has great promise for addressing coordination problems in a variety of applications, such as robotic systems [1], autonomous cars [2], and sensor networks [3]. Such complex tasks often require MARL to learn decentralized policies for agents to jointly optimize a global cumulative reward signal, which posts a number of challenges, including multi-agent credit assignment [4, 5], non-stationarity [6, 7], and scalability [3, 8]. Recently, by leveraging the strength of deep learning techniques, cooperative MARL has made great progress [9–19], particularly in value-based methods that demonstrate state-of-the-art performance on challenging tasks such as StarCraft unit micromanagement [20].

Value factorization is a popular approach to effectively scaling up cooperative multi-agent Q-learning in complex domains. One fundamental problem of factorized multi-agent algorithms is the trade-off between the learning scalability and the representational capacity of value functions. As a representative method, value-decomposition network (VDN) [10] is proposed to obtain excellent scalability based on a popular paradigm called *centralized training with decentralized execution* (CTDE) [21], which learns a centralized but factorizable joint value function $Q_{\text{tot}}$ represented as the summation of individual value functions $Q_i$. During the execution, decentralized policies can be easily derived by greedily selecting individual actions from the local value function $Q_i$. An implicit multi-agent credit assignment is realized because $Q_i$ is learned by neural network backpropagation from

---

*Equal contribution.
[†]Work done while Zhizhou was an undergraduate at Tsinghua University.

35th Conference on Neural Information Processing Systems (NeurIPS 2021).

the total temporal-difference error on the single global reward signal. This linear value factorization structure can significantly improve the scalability of multi-agent joint policy training and individual policy execution. This linear value factorization adopted by VDN also realizes a sufficient condition for an important principle of CTDE, the IGM (*Individual-Global-Max*) principle [13], which asserts the consistency between joint and individual greedy action selection and ensures the consistent policy learning under the CTDE paradigm. Following this principle, most recent advances focus on enriching the function expressiveness of the factorization from $Q_{\text{tot}}$ to $Q_i$. QMIX [12] uses a monotonic function to represent the joint value function $Q_{\text{tot}}$ through local values $Q_i$, whose function expressiveness is further improved by QTRAN [13] and QPLEX [19] that aim to achieve full function expressiveness induced by the IGM principle. These approaches show promise and achieve state-of-the-art performance in complicated cooperative multi-agent domains.

Despite these impressive empirical successes, theoretical understandings for these MARL approaches are still limited. To bridge this gap, this paper is the first to consider a general framework for theoretical studies. We formulate *Factorzed Multi-Agent Fitted Q-Iteration* (FMA-FQI) for formally analyzing cooperative MARL with value factorization. This framework generalizes single-agent Fitted Q-Iteration, a popular model for studying Q-learning algorithms with function approximation [22–25]. FMA-FQI models the iterative training procedure of multi-agent Q-learning using empirical Bellman error minimization. Under this framework, we investigate two popular value factorization methods: the linear factorization used by VDN [10], and the IGM factorization adopted by QTRAN [13] and QPLEX [19]. Our analyses reveal algorithmic properties of these approaches and provide insights on fundamental questions: Why do they work? Do they converge? When may they not perform well?

The main contribution of this paper can be summarized as follows:

1. We formalize *Factorized Multi-Agent Fitted Q-Iteration* (FMA-FQI) as a general theoretical framework for analyzing cooperative multi-agent Q-learning with value factorization.

2. Based on FMA-FQI, we study linear value factorization and derive a closed-form solution to its Bellman error minimization. We then reveal two novel insights that: 1) linear value factorization implicitly realizes a powerful counterfactual credit assignment and 2) on-policy training is beneficial to its stability and local convergence near the optimal solutions.

3. Using FMA-FQI, we also study IGM value factorization and prove its global convergence and optimality.

4. Empirical analysis is conducted to connect our theoretical implications to practical scenarios by evaluating state-of-the-art deep MARL approaches.

## 2 Related Work

Deep Q-learning algorithms that use neural networks as function approximators have shown great promise in solving complicated decision-making problems [26]. One of the core components of such methods is iterative Bellman error minimization, which can be modelled by a classical framework called Fitted Q-Iteration (FQI) [27]. FQI utilizes a specific $Q$-function class to iteratively optimize empirical Bellman error on a dataset $D$ with respect to a frozen target. Great efforts have been made towards theoretically characterizing the behavior of FQI with finite samples and imperfect function classes [22–24]. From an empirical perspective, there is also a growing trend to adopt FQI for empirical analysis of deep offline Q-learning algorithms [25, 28]. In MARL, the joint $Q$-function class grows exponentially with the number of agents, leading many algorithms [10, 13, 19] to utilize different value factorization structures to balance scalability and optimality. In section 4, we generalize FQI to formalize a factorized multi-agent fitted Q-iteration framework for analyzing cooperative multi-agent Q-learning algorithms with value factorization.

To achieve superior effectiveness and scalability in multi-agent settings, centralized training with decentralized executing (CTDE) has become a popular MARL paradigm [29, 30]. *Individual-Global-Max* (IGM) principle [13] is a critical concept for value-based CTDE [31], that ensures the consistency between joint and local greedy action selections and enables effective performance in both training and execution phases. VDN [10] utilizes linear value factorization to satisfy a sufficient condition of IGM. This simple linear structure of VDN has become very popular in MARL due to its excellent scalability [13, 19, 32] and inspired many follow-up methods. QMIX [12] proposes a monotonic $Q$-network structure to improve the expressiveness of the factorized function class. QTRAN [13]

aims to realize the entire IGM function class, but its proposed objective is computationally intractable and requires two extra soft regularizations to approximate IGM, which actually loses rigorous IGM guarantees. QPLEX [19], the state-of-the-art multi-agent Q-learning algorithm, encodes the IGM principle into the $Q$-network architecture and realize a complete IGM function class, but may have potential limitations in scalability. This paper focuses on the theoretical and empirical understanding of multi-agent Q-learning with linear and IGM value factorization to explore the underlying implications of their corresponding factorized value structures.

## 3 Notations and Preliminaries

### 3.1 Decentralized Rich-Observation Markov Decision Process (Dec-ROMDP)

To support theoretical analysis on cooperative multi-agent Q-learning, we consider the problem formulation of *Decentralized Rich-Observation Markov Decision Process* (Dec-ROMDP). Dec-ROMDP is an interpolation of ROMDP [33, 34] and Dec-POMDP [35], in which the latent state space is finite, and the observation space could be arbitrarily large. A Dec-ROMDP is defined as a tuple $\mathcal{M} = \langle \mathcal{N}, \mathcal{S}, \mathcal{X}, \mathcal{A}, P, \Lambda, r, \gamma \rangle$. $\mathcal{N} \equiv \{1, \ldots, n\}$ is a finite set of agents. $\mathcal{S}$ is a finite set of global latent states. $\mathcal{X}$ denotes the observation space (a.k.a. context space) which is larger than the latent state space. $\mathcal{A}$ denotes the action space for an individual agent. The joint action $\mathbf{a} \in \mathbf{A} \equiv \mathcal{A}^n$ is a collection of individual actions $[a_i]_{i=1}^n$. At each timestep $t$, a selected joint action $\mathbf{a}_t$ results in a latent-state transition $s_{t+1} \sim P(\cdot|s_t, \mathbf{a}_t)$ and a global reward signal $r(s_t, \mathbf{a}_t)$. The observation of agent $i$ is generated from the emission distribution $x_{t,i} \sim \Lambda(\cdot|i, s_t)$.

The major difference between Dec-ROMDP and Dec-POMDP is the concept of rich observation. Dec-ROMDP ensures that the observation emission distributions are disjoint across latent states, i.e., there exists an inverse function $\Lambda^{-1} : \mathcal{N} \times \mathcal{X} \to \mathcal{S}$ that decodes full information of latent states from observations. This assumption enables us to consider *reactive function classes* [36, 37], which only takes the latest observation as inputs. Formally, the goal for MARL in Dec-ROMDP is to construct a joint policy $\boldsymbol{\pi} = \langle \pi_1, \ldots, \pi_n \rangle$ maximizing expected discounted rewards $V^{\boldsymbol{\pi}}(\boldsymbol{x}) = \mathbb{E}\left[\sum_{t=0}^\infty \gamma^t r(s_t, \boldsymbol{\pi}(\boldsymbol{x}_t)) | \boldsymbol{x}_0 = \boldsymbol{x}\right]$, where $\pi_i : \mathcal{X} \mapsto \mathcal{A}$ denotes an individual policy of agent $i$ which depends on its local observation. The corresponding action-value function is denoted as $Q^{\boldsymbol{\pi}}(\boldsymbol{x}_t, \mathbf{a}_t) = \mathbb{E}[r(s_t, \mathbf{a}_t) + \gamma V^{\boldsymbol{\pi}}(\boldsymbol{x}_{t+1}) | \boldsymbol{x}_t]$. More discussions are deferred to Appendix A.

**Why Dec-ROMDP?** The most widely-used environment setting in deep MARL for empirical studies is Dec-POMDP. However, as implied by prior work, finding an optimal policy in infinite-horizon Dec-POMDPs is an incomputable problem in the classical computation model [38]. There is no algorithm that can achieve strong theoretical guarantees in the Dec-POMDP setting. Following recent advances in reinforcement learning with function approximation [33, 34, 39, 40], we consider the rich-observation setting to derive implications from a theoretical perspective. This setting is also adopted by a concurrent work to establish algorithm analysis for cooperative multi-agent reinforcement learning [41].

### 3.2 Centralized Training with Decentralized Execution (CTDE)

Most deep multi-agent Q-learning algorithms with value factorization adopt the paradigm of centralized training with decentralized execution [21]. In the training phase, the centralized trainer can access all global information, including joint trajectories, shared global rewards, agents' policies, and value functions. In the decentralized execution phase, every agent makes individual decisions based on its local information. *Individual-Global-Max* (IGM) [13] is a common principle to realize effective decentralized policy execution. It enforces the action selection consistency between the global joint action-value $Q_{\text{tot}}$ and individual action-values $[Q_i]_{i=1}^n$, which are specified as follows:

$$\textbf{(IGM)} \quad \arg\max_{\mathbf{a} \in \mathbf{A}} Q_{\text{tot}}(\boldsymbol{x}, \mathbf{a}) = \left\langle \arg\max_{a_1 \in \mathcal{A}} Q_1(x_1, a_1), \cdots, \arg\max_{a_n \in \mathcal{A}} Q_n(x_n, a_n) \right\rangle. \quad (1)$$

QPLEX [19] is the state-of-the-art deep multi-agent Q-learning algorithm that realizes a complete IGM factorization. QTRAN [13] is another method to approximate the IGM principle by two extra soft regularizations. Moreover, as stated in Eq. (2), the linear value factorization adopted by VDN [10] is a sufficient condition for the IGM constraint stated in Eq. (1) but is not a necessary condition, which induces a limited joint action-value function class.

$$\textbf{(Additivity)} \quad Q_{\text{tot}}(\boldsymbol{x}, \mathbf{a}) = \sum_{i=1}^n Q_i(x_i, a_i). \quad (2)$$

# 4 Factorized Multi-Agent Fitted Q-Iteration

Constructing a specific factorized value function class satisfying the IGM condition is a critical step to realize the centralized training with decentralized execution (CTDE) paradigm. In the literature of Q-learning with function approximation, *Fitted Q-Iteration* (FQI) [27] is a popular framework for both theoretical and empirical studies [22–25, 28]. In this section, we generalize single-agent FQI to formalize a *Factorzed Multi-Agent Fitted Q-Iteration* (FMA-FQI) framework, which enables us to characterize the algorithmic properties of different factorization structures in the CTDE paradigm.

For multi-agent Q-learning with value factorization, we use $Q_{\text{tot}}$ to denote the global but factorized value function, which can be represented as a function of individual value functions $[Q_i]_{i=1}^n$. In other words, we can use $[Q_i]_{i=1}^n$ to represent $Q_{\text{tot}}$. For brevity, we overload $Q = \langle Q_{\text{tot}}, [Q_i]_{i=1}^n \rangle$ to denote their association. Following fitted Q-iteration, FMA-FQI considers an iterative optimization framework based on a given dataset $D = \{(\boldsymbol{x}, \mathbf{a}, r, \boldsymbol{x}')\}$ as presented in Algorithm 1.

---

**Algorithm 1** Factorized Multi-Agent Fitted Q-Iteration (FMA-FQI)

1: **Input:** dataset $D$, the number of iterations $T$, factorized multi-agent value function class $\mathcal{Q}^{\text{FMA}}$
2: Randomly initialize $Q^{(0)}$ from $\mathcal{Q}^{\text{FMA}}$
3: **for** $t = 0 \ldots T - 1$ **do**
4:     Iteratively update

$$Q^{(t+1)} \leftarrow \mathcal{T}_D^{\text{FMA}} Q^{(t)} \equiv \underset{Q \in \mathcal{Q}^{\text{FMA}}}{\arg\min} \; \underset{(\boldsymbol{x}, \mathbf{a}, r, \boldsymbol{x}') \sim D}{\mathbb{E}} \left( \hat{y}^{(t)}(\boldsymbol{x}, \mathbf{a}, \boldsymbol{x}') - Q_{\text{tot}}(\boldsymbol{x}, \mathbf{a}) \right)^2 \quad (3)$$

    where $\hat{y}^{(t)}(\boldsymbol{x}, \mathbf{a}, \boldsymbol{x}') = r + \gamma \max_{\mathbf{a}' \in \mathbf{A}} Q_{\text{tot}}^{(t)}(\boldsymbol{x}', \mathbf{a}')$ denotes the one-step TD target.
5:     Construct policies by individual value functions

$$\forall i \in \mathcal{N}, \; \pi_i^{(t+1)}(x_i) = \underset{a_i \in \mathcal{A}}{\arg\max} \; Q_i^{(t+1)}(x_i, a_i) \quad (4)$$

6: **Return:** $[\pi_i^{(T)}]_{i=1}^n$

---

Formally, FMA-FQI specifies the $Q$-function class through a multi-agent value factorization structure:

$$\mathcal{Q}^{\text{FMA}} = \left\{ Q \; \middle| \; \exists f \in \mathcal{F}^{\text{FMA}}, \; Q_{\text{tot}}(\boldsymbol{x}, \mathbf{a}) = f\left(\boldsymbol{x}, \mathbf{a}, [Q_i]_{i=1}^n\right), \; [Q_i]_{i=1}^n \in \mathbb{R}^{|\mathcal{X} \times \mathcal{A}|^n} \right\}, \quad (5)$$

where $\mathcal{F}^{\text{FMA}}$ denotes the function class characterizing the factorization structure. For example, the IGM constraint stated as Eq. (1) and the additivity constraint stated as Eq. (2) correspond to the value factorization structures used by QPLEX [19] and VDN [10], respectively.

The empirical Bellman error minimization in FMA-FQI is established using the global reward signals. To facilitate further discussions, we rewrite the regression objective as follows:

$$Q^{(t+1)} \leftarrow \mathcal{T}_D^{\text{FMA}} Q^{(t)} \equiv \underset{Q \in \mathcal{Q}^{\text{FMA}}}{\arg\min} \; \underset{(\boldsymbol{x}, \mathbf{a}) \sim D}{\mathbb{E}} \left( y^{(t)}(\boldsymbol{x}, \mathbf{a}) - Q_{\text{tot}}(\boldsymbol{x}, \mathbf{a}) \right)^2 \quad (6)$$

where $y^{(t)}(\boldsymbol{x}, \mathbf{a}) = r + \gamma \mathbb{E}_{\boldsymbol{x}'} \left[ \max_{\mathbf{a}'} Q_{\text{tot}}^{(t)}(\boldsymbol{x}', \mathbf{a}') \right]$ denotes the expected one-step TD target. Assume the dataset is adequate, the equivalence between Eq. (3) and Eq. (6) is proved in Appendix A.3.

**Modeling CTDE by FMA-FQI.** Different from the single-agent case, FMA-FQI considers the multi-agent value factorization structure, $Q_{\text{tot}}$ and $[Q_i]_{i=1}^n$, to support effective training and scalable execution. In the global reward setting, the shared reward signal can only supervise the training of the joint value function $Q_{\text{tot}}$ (see Eq. (6)). The learned joint value function $Q_{\text{tot}}$ would be automatically decomposed to individual value functions $[Q_i]_{i=1}^n$ through the factorized function structure, which can be further used to perform decentralized execution (see Eq. (4)). In practice, FMA-FQI illustrated in Eq. (5) provides the end-to-end learning of $[Q_i]_{i=1}^n$ as the iterative optimization of joint value function $Q_{\text{tot}}$. The greedy action selection can be searched in the individual action space $\mathcal{A}$ rather than the joint action space $\mathcal{A}^n$, which significantly reduces the computation costs.

**Relation to Prior Work.** Most prior work of FQI focus on the single-agent RL with finite-sample analyses and exploring minimum requirements for performance guarantees (e.g., data distribution and function capacity) in a general sense [22–24]. In comparison, this paper is motivated by recent

advances in the multi-agent area. To our best knowledge, we are the first to generalize single-agent FQI to the multi-agent setting (i.e., FMA-FQI) for analyzing value factorization. For instance, we characterize the algorithmic properties of two specific value function classes, linear value factorization and IGM value factorization, which are widely used in deep MARL [10, 13, 19]. These factorization structures are implemented by certain network architectures and thus can be well modelled by FMA-FQI from the perspective of function approximation.

## 5 Multi-Agent Q-Learning with Linear Value Factorization

Linear value factorization proposed by VDN [10] is a simple yet effective method to realize a sufficient condition of the IGM principle in the CTDE paradigm. In this section, we provide theoretical analysis towards a deeper understanding of this popular factorization structure. Our result is based on FMA-FQI with linear value factorization, named FQI-LVF. We derive the closed-form update rule of FQI-LVF, and then reveal the underlying credit assignment mechanism realized by linear value factorization learning. We find that FQI-LVF has potential risks of unbounded divergence and may not have any fixed-point solutions in the general case. To improve its training stability, we prove that on-policy data collection can ensure the existence of fixed-point Q-values and provide local convergence guarantees near the optimal value function.

### 5.1 Multi-Agent Fitted Q-Iteration with Linear Value Factorization (FQI-LVF)

We define multi-agent fitted Q-iteration with linear value factorization as follows.

**Definition 1** (FQI-LVF). *FQI-LVF is an instance of FMA-FQI stated in Algorithm 1, which specifies the action-value function class with linear value factorization:*

$$\mathcal{Q}^{LVF} = \left\{ Q \;\Big|\; Q_{tot}(\boldsymbol{x}, \mathbf{a}) = \sum_{i=1}^{n} Q_i(x_i, a_i),\; [Q_i]_{i=1}^{n} \in \mathbb{R}^{|\mathcal{X} \times \mathcal{A}|^n} \right\}. \tag{7}$$

FQI-LVF considers a popular linear value factorization structure in MARL [10, 13, 19], which reduces the action-value function class to provide attractive scalability for policy training [32]. Value-decomposition network (VDN) [10] provides a deep-learning-based implementation of FQI-LVF, in which individual value functions $[Q_i]_{i=1}^{n}$ are parameterized by deep neural networks, and the joint value function $Q_{\text{tot}}$ can be simply formed by their summation.

### 5.2 Implicit Counterfactual Credit Assignment in Linear Value Factorization

In the formulation of FQI-LVF, the empirical Bellman error minimization with linear value function class $\mathcal{Q}^{\text{LVF}}$ can be regarded as a weighted linear least-squares problem, which contains $n|\mathcal{X} \times \mathcal{A}|$ variables to form individual value functions $[Q_i]_{i=1}^{n}$ and $|\mathcal{X} \times \mathcal{A}|^n$ data points corresponding to all entries of the regression target $y^{(t)}(\boldsymbol{x}, \mathbf{a})$. Formally, by plugging the definition of $\mathcal{Q}^{\text{LVF}}$ into multi-agent fitted Q-iteration, FQI-LVF iteratively optimizes the following least-squares problem:

$$Q^{(t+1)} \leftarrow \mathcal{T}_D^{\text{LVF}} Q^{(t)} \equiv \underset{Q \in \mathcal{Q}^{\text{LVF}}}{\arg\min} \sum_{\boldsymbol{x}, \mathbf{a}} p_D(\boldsymbol{x}, \mathbf{a}) \left( y^{(t)}(\boldsymbol{x}, \mathbf{a}) - \sum_{i=1}^{n} Q_i^{(t+1)}(x_i, a_i) \right)^2,$$

where $p_D$ denotes the probability measured by the dataset $D$. The closed-form solution of this least-squares problem is presented in Theorem 1 with the following assumption:

**Assumption 1** (Decentralized Data Collection). *The dataset $D$ is collected by a decentrailized and exploratory policy $\boldsymbol{\pi}^D$ satisfying:*

$$\forall (\boldsymbol{x} \times \boldsymbol{a}) \in \mathcal{X} \times \mathbf{A},\; \boldsymbol{\pi}^D(\mathbf{a}|\boldsymbol{x}) = \prod_{i \in \mathcal{N}} \pi_i^D(a_i|x_i) > 0.$$

**Theorem 1.** *Let $Q^{(t+1)} = \mathcal{T}_D^{LVF} Q^{(t)}$ denote a single iteration of the empirical Bellman operator. $\forall i \in \mathcal{N}, \forall(\boldsymbol{x}, \mathbf{a}) \in \mathcal{X} \times \mathbf{A}$, the individual action-value function $Q_i^{(t+1)}(x_i, a_i)$ is updated to*

$$\underbrace{\mathbb{E}_{(x'_{-i}, a'_{-i}) \sim p_D(\cdot|x_i)} \left[ y^{(t)}\left(x_i \oplus x'_{-i}, a_i \oplus a'_{-i}\right) \right]}_{\text{evaluation of the individual action } a_i} - \frac{n-1}{n} \underbrace{\mathbb{E}_{\boldsymbol{x}', \mathbf{a}' \sim p_D(\cdot|\Lambda^{-1}(x_i))} \left[ y^{(t)}\left(\boldsymbol{x}', \mathbf{a}'\right) \right]}_{\text{counterfactual baseline}} + w_i(x_i), \tag{8}$$

*where $z_i \oplus z'_{-i}$ denotes $\langle z'_1, \cdots, z'_{i-1}, z_i, z'_{i+1}, \cdots, z'_n \rangle$, and $z'_{-i}$ denotes the elements of all agents except for agent $i$. $\Lambda^{-1}(x_i)$ denotes the inverse of observation emission, which decodes the current latent state from $x_i$. The residue term $\mathbf{w} \equiv [w_i]_{i=1}^n$ is an arbitrary function satisfying $\forall x \in \mathcal{X}$, $\sum_{i=1}^n w_i(x_i) = 0$.*

The proof of Theorem 1 is based on *Moore-Penrose inverse* [42] for weighted linear regression analysis. The detailed proofs for all theory statements in this paper are deferred to Appendix. To serve intuitions, we present a simplified version of Eq. (8) on Multi-agent MDP (MMDP) [43] to make the underlying insights more accessible:

$$Q^{(t+1)}(s, a) = \mathop{\mathbb{E}}_{a'_{-i} \sim p_D(\cdot|s)} \left[ y^{(t)} \left( s, a_i \oplus a'_{-i} \right) \right] - \frac{n-1}{n} \mathop{\mathbb{E}}_{\mathbf{a}' \sim p_D(\cdot|s)} \left[ y^{(t)} \left( s, \mathbf{a}' \right) \right] + w_i(s), \quad (9)$$

where MMDP refers to a special case of Dec-ROMDP with all individual observations $x_i = s$, and $s$ denotes the latent state. The residue term $w_i(s)$ satisfies $\forall s \in \mathcal{S}$, $\sum_{i=1}^n w_i(s) = 0$. Eq. (9) is derived from Eq. (8) by translating notations.

**Solution Space.** The last term of Eq. (8), vector $\mathbf{w}$, indicates the entire valid individual action-value function space. We can ignore this term because $\mathbf{w}$ does not affect the local action selection of each agent and will be eliminated in the summation operator of linear value factorization (see Eq. (2)), which indicates that joint action-value $Q_{\text{tot}}^{(t+1)}$ has a unique closed-form solution.

**Implicit Counterfactual Credit Assignment.** To interpret the underlying credit assignment of linear value factorization, we regard the empirical probability $p_D(\mathbf{a}|s)$ within the dataset $D$ as a *default policy*. The first term of Eq. (8) is the expected value of an individual action $a_i$ over the actions of other agents, which evaluates the expected return of executing an individual action $a_i$. The second term of Eq. (8) is the expected value of the default policy, which is considered as the *counterfactual baseline*. Their difference corresponds to a classical credit assignment mechanism called *counterfactual difference rewards* [4, 44]. This mechanism is usually adopted by policy-based methods such as counterfactual multi-agent policy gradients (COMA) [11]. A slight difference from Eq. (8) is the extra importance weight $(n-1)/n$. It makes our derived credit assignment of FQI-LVF to be more meaningful in the sense that all global rewards should be assigned to agents. Consider a simple case where all joint actions generate the same reward signals, Eq. (8) will assign $1/n$ unit of rewards to each agent, but *counterfactual difference rewards* used by [11] will assign 0. The policy-gradient-based methods are not sensitive to such a constant gap, but it is critical to the value estimation of Q-learning algorithms.

**Remark on Assumptions.** The closed-form solution derived in Theorem 1 relies on two assumptions, decentralized data collection and rich-observation problem formulation. These two assumptions correspond to the least requirement we found to make the closed-form solution meaningful and explainable. In Appendix B.3, we prove that if we have a closed-form solution for FQI-LVF without either of the above assumptions, we can obtain a closed-form solution for arbitrary linear least-squares problems with binary weight matrices. Since the general least-squares problem does not have existing analytical solutions, it is unlikely to derive a general closed-form solution of FQI-LVF without any assumptions. It remains an open question whether we can relax current assumptions a little while keeping the simplicity of derived formulas.

### 5.3 Data Distribution Matters for Linear Value Factorization

The closed-form update rule of FQI-LVF stated in Theorem 1 enables us to investigate more algorithmic properties of linear value factorization in multi-agent Q-learning. In the empirical literature, the major strength of linear value factorization is its high scalability [32], since the additive constraint between the global and local values is non-parametric which does not require additional learnable parameters. Meanwhile, the limitation of linear value factorization structure is also induced by its simplicity. The function capacity of $Q^{\text{LVF}}$ cannot express all valid global values in $\mathbb{R}^{|\mathcal{X} \times \mathcal{A}|^n}$ [13, 19, 31]. As suggested by prior work [24, 45], the performance of Q-learning algorithms would become sensitive to the training data distribution when the function approximator is not perfect. In this section, we investigate how linear value factorization structure interacts with different data distributions. Our theoretical results contain two aspects:

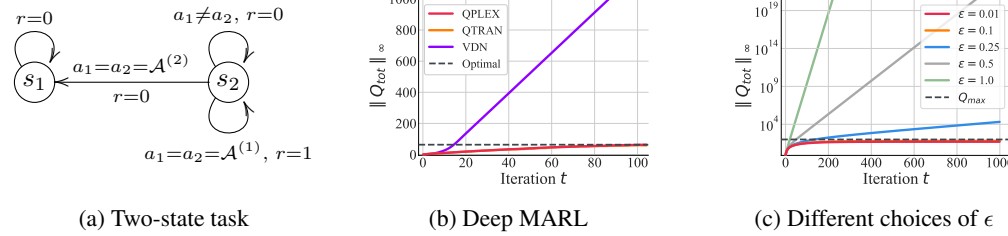

| (a) Two-state task | (b) Deep MARL | (c) Different choices of $\epsilon$ |

Figure 1: (a) A two-state MMDP where FQI-LVF will diverge to infinity when $\gamma \in \left(\frac{4}{5}, 1\right)$ from arbitrary initialization $Q^{(0)}$. (b) The learning curves of $\|Q_{\text{tot}}\|_\infty$ produced by running several deep multi-agent Q-learning algorithms. (c) The learning curves of $\|Q_{\text{tot}}\|_\infty$ of on-policy FQI-LVF on the given task where the dataset is generated by different choices of hyper-parameters $\epsilon$ for $\epsilon$-greedy.

1. When the dataset is collected in a fully offline manner, e.g., by a uniform random policy, FQI-LVF would suffer from unbounded divergence and do not have any fixed points in the most unfavorable environment.

2. By iteratively updating the dataset to a nearly on-policy data distribution, FQI-LVF would have fixed-point Q-values that expresses the optimal policy.

### 5.3.1 Divergence Risk in Offline Training

To begin with the discussion on linear value factorization in offline training settings, we first investigate an important property, named $\gamma$-contraction, in the FQI-LVF framework (see Proposition 1).

**Proposition 1.** *The empirical Bellman operator $\mathcal{T}_D^{LVF}$ is not a $\gamma$-contraction, i.e., the following important property of the standard Bellman optimality operator $\mathcal{T}$ does not hold for $\mathcal{T}_D^{LVF}$ anymore.*

$$(\gamma\text{-contraction}) \qquad \forall Q_{tot}, Q_{tot}' \in \mathcal{Q}, \ \ \|\mathcal{T}Q_{tot} - \mathcal{T}Q_{tot}'\|_\infty \leq \gamma \|Q_{tot} - Q_{tot}'\|_\infty$$

For the standard *Bellman optimality operator* $\mathcal{T}$, $\gamma$-contraction property is critical to deriving the convergence guarantee of Q-learning algorithms [46]. In the context of FQI-LVF, the additivity constraint limits the joint action-value function class that it can express, which deviates the empirical Bellman operator $\mathcal{T}_D^{\text{LVF}}$ from the original *Bellman optimality operator* $\mathcal{T}$ (see Theorem 1). This deviation is also known as *inherent Bellman error* [22] or projection error, which corrupts a broad set of stability properties, including $\gamma$-contraction.

To serve a concrete example, we construct a simple MMDP with two agents, two global states, and two actions (see Figure 1a). The optimal policy of this task is simply executing the action $\mathcal{A}^{(1)}$ at state $s_2$, which is the only way for two agents to obtain a positive reward. The learning curve of $\epsilon = 1.0$ (green one) in Figure 1c refers to an offline setting with uniform data distribution, in which an unbounded divergence can be observed as depicted by the following proposition.

**Proposition 2.** *There exist an MMDP such that, when using uniform data distribution, the value function of FQI-LVF diverges to infinity from an arbitrary initialization $Q^{(0)}$.*

In the proof of Proposition 2, we will show that the unbounded divergence would happen to an arbitrary initialization $Q^{(0)}$ in this example. To provide an implication for practical scenarios, we also investigate the performance of several deep multi-agent Q-learning algorithms in this two-state task. As shown in Figure 1b, VDN [10], a deep-learning-based implementation of FQI-LVF, also results in unbounded divergence.

### 5.3.2 Benefits of On-Policy Data Collection

As shown in Theorem 1, the choice of training data distribution affects the output of the empirical Bellman operator $\mathcal{T}_D^{\text{LVF}}$. Proposition 2 indicates that, with a fully offline data distribution, FQI-LVF may have no fixed points in the worst case. In this section, we show that FQI-LVF with on-policy data collection guarantees to have fixed-point Q-values expressing the optimal policy. It highlights the importance of training data distribution for linear value factorization. More specifically, we consider the dataset $D_t$ is accumulated by running an $\epsilon$-greedy policy [26] at $t$-th iteration, i.e., each agent performs a random action with probability $\epsilon$. To serve intuitions, we present an informal

statement here and defer the detailed version, its proof, and the algorithm box of on-policy FQI-LVF to Appendix C.2.

**Theorem 2** (Informal). *When the hyper-parameter $\epsilon$ is sufficiently small, on-policy FQI-LVF has at least one fixed-point Q-value that derives the optimal policy.*

The proof of this statement is based on Brouwer's fixed-point theorem [47], in which we construct a bounding box in the value function space such that $\mathcal{T}_D^{\mathrm{LVF}}$ is a closed operator. Within this bounding box, the induced policies of these value functions are equal to the optimal policy $\pi^*$ in the given MDP. It indicates that multi-agent Q-learning with linear value factorization has a convergent region, in which all included value function induces optimal actions. It ensures the local convergence guarantee near the optimal solution.

Figure 1c visualizes the performance of on-policy FQI-LVF with different values of the hyper-parameter $\epsilon$. With a smaller $\epsilon$ (such as 0.1 or 0.01), on-policy FQI-LVF demonstrates excellent training stability, and their corresponding collected datasets are closer to on-policy data distribution.

**Relation to Prior Work.** In the literature of Q-learning with function approximation, there is a long history of studying the behavior of Q-learning with non-universal function classes. Many counterexamples have been proposed to indicate certain function classes suffering from oscillation [48] or even unbounded divergence [49, 50]. Our analysis corresponds to a case study of the function class $\mathcal{Q}^{\mathrm{LVF}}$ with linear value factorization structure in the multi-agent setting, which is restricted by the additivity constraint stated in Eq. (2). Several prior works suggest that on-policy data distribution is beneficial to the learning stability in some settings [51–53], and our theorems in this section also indicate this implication in the multi-agent linear value factorization. A recent work, Weighted QMIX [45], proposes a technique called *idealised central weighting* that re-weights the training data to a nearly on-policy distribution. Their results show that training with on-policy data can also benefit monotonic value factorization.

# 6 Multi-Agent Q-Learning with IGM Value Factorization

Recently, advanced deep multi-agent Q-learning algorithms, QTRAN [13] and QPLEX [19], aim to utilize the IGM factorization method to formulate their value function class for CTDE paradigm and achieve state-of-the-art performance in StarCraft II benchmark [20] with online data collection. In this section, we introduce FMA-FQI with IGM factorization, named FQI-IGM. Compared with the unbounded divergence risk of linear value factorization analyzed in section 5, we find that FQI-IGM provides a global optimality convergence guarantee in Dec-ROMDPs, which implies IGM value factorization is a stable choice for cooperative multi-agent Q-learning.

## 6.1 Multi-Agent Fitted Q-Iteration with IGM Value Factorization (FQI-IGM)

We define multi-agent fitted Q-iteration with IGM value factorization as follows.

**Definition 2** (FQI-IGM). *FQI-IGM is an instance of FMA-FQI stated in Algorithm 1, which specifies the action-value function class with a complete IGM principle realization*

$$\mathcal{Q}^{IGM} = \left\{ Q \ \middle| \ \arg\max_{\mathbf{a} \in \mathbf{A}} Q_{tot}(\boldsymbol{x}, \mathbf{a}) = \left\langle \arg\max_{a_1 \in \mathcal{A}} Q_i(x_i, a_i) \right\rangle_{i=1}^n, \ [Q_i]_{i=1}^n \in \mathbb{R}^{|\mathcal{X} \times \mathcal{A}|^n} \right\}. \quad (10)$$

Compared with FQI-LVF stated in Definition 1, the differences are the factorized value function classes, i.e, $\mathcal{Q}^{\mathrm{LVF}}$ vs. $\mathcal{Q}^{\mathrm{IGM}}$. Note that $\mathcal{Q}^{\mathrm{LVF}} \subset \mathcal{Q}^{\mathrm{IGM}}$ indicates that the linear factorization structure realizes a subspace of IGM value functions. In empirical studies [19], the enriched value function class $\mathcal{Q}^{\mathrm{IGM}}$ is observed to have better training stability and overall performance than that of linear value factorization $\mathcal{Q}^{\mathrm{LVF}}$. In the following section, we will provide theoretical supports regarding the differences between the algorithmic properties of $\mathcal{Q}^{\mathrm{LVF}}$ and $\mathcal{Q}^{\mathrm{IGM}}$.

## 6.2 Global Convergence Guarantee of IGM Value Factorization

The major difference between $\mathcal{Q}^{\mathrm{LVF}}$ and $\mathcal{Q}^{\mathrm{IGM}}$ is their representation power of function expressiveness. Recall that the unbounded divergence of linear value factorization is caused by the projection error induced from the limited function expressiveness of $\mathcal{Q}^{\mathrm{LVF}}$. In this section, we show that the divergence

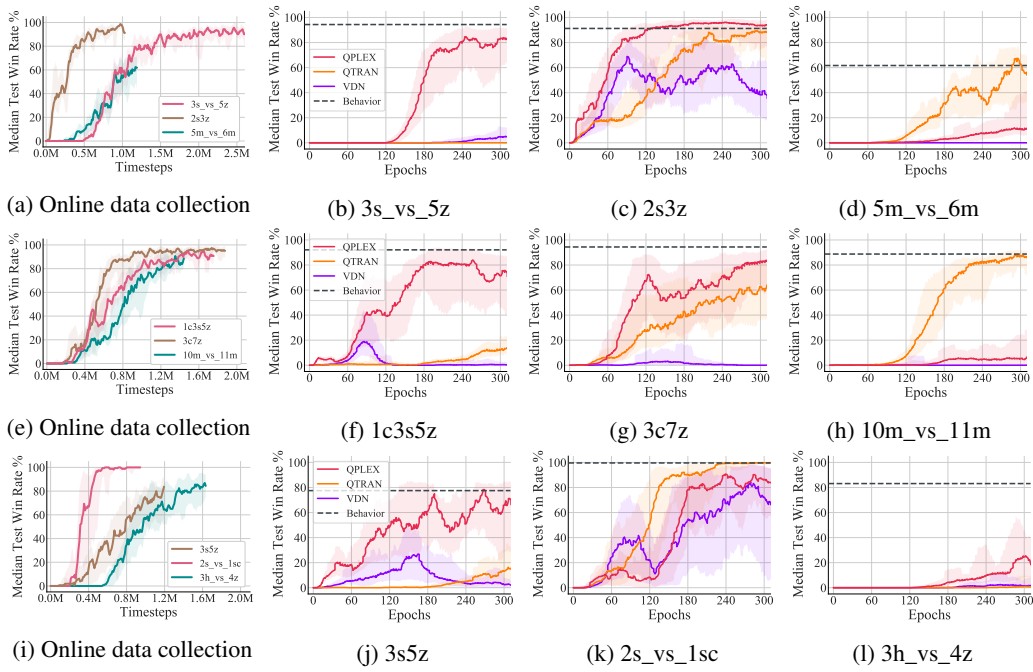

Figure 2: (a,c,i) The learning curve of VDN using online data collection when constructing the static datasets. (b-d,f-h,j-l) Evaluating the performance of deep multi-agent Q-learning algorithms with a given static dataset.

issues would not happen to FQI-IGM using $\mathcal{Q}^{\text{IGM}}$ in Dec-ROMDPs. Formally, the convergence guarantee of FQI-IGM can be derived based on the following assumption:

**Assumption 2** (Exploratory Data Collection). *The dataset $D$ is collected by an exploratory policy $\boldsymbol{\pi}^D$ satisfying $\forall (\boldsymbol{x} \times \boldsymbol{a}) \in \boldsymbol{\mathcal{X}} \times \boldsymbol{A}, \boldsymbol{\pi}^D(\mathbf{a}|\boldsymbol{x}) > 0$.*

**Theorem 3.** *FQI-IGM globally converges to the optimal value function in arbitrary Dec-ROMDPs.*

Theorem 3 relies on a fact that $\mathcal{Q}^{\text{IGM}}$ is closed under the Bellman operator, i.e., the empirical Bellman error minimization discussed in Eq. (6) can reach zero, and the empirical Bellman operator $\mathcal{T}_D^{\text{IGM}}$ satisfies the $\gamma$-contraction property. The global convergence presented in Theorem 3 is the same as the convergence property of tabular value iteration.

**Remark on Assumptions.** The only purpose of introducing the assumption of the exploratory dataset as Assumption 2 is to make the outputs of empirical Bellman error minimization well-defined in analysis. This assumption is weaker than the assumptions used in Theorem 1 and Theorem 2 for analyzing linear value factorization. It indicates that multi-agent Q-learning with IGM value factorization is not sensitive to the training data distribution. In addition, we also consider the problem formulation of Dec-ROMDP for FQI-IGM, since prior work [38] suggests that, there is no algorithm that can guarantee global convergence in general infinite-horizon Dec-POMDPs.

## 6.3 IGM Value Factorization Provides Better Learning Stability in Offline Deep MARL

In this subsection, we conduct an empirical study to connect our theoretical implications to practical scenarios of deep multi-agent Q-learning algorithms. We evaluate three deep-learning-based counterparts of FQI-LVF and FQI-IGM, i.e., VDN [10], QTRAN [13], and QPLEX [19], respectively. VDN can be regarded as a deep-learning-based implementation of FQI-LVF. QTRAN and QPLEX correspond to two different implementations of FQI-IGM. As suggested by Proposition 2 that linear value factorization may suffer from training stability in an offline setting, we utilize StarCraft Multi-Agent Challenge (SMAC) benchmark [20] with offline data collection to investigate the effect of the expressiveness of a factorized value function, i.e., which value factorization is suitable for multi-agent offline reinforcement learning.

We evaluate the performance of VDN, QTRAN, and QPLEX on nine common maps of StarCraft II. The results are shown in Figure 2. The datasets are collected by training a VDN with $\epsilon$-greedy online data collection, which corresponds to the learning curves of *Behavior* line shown in Figure 2(a,e,i). Figure 2(b-d,f-h,j-l) illustrate that QPLEX and QTRAN with $Q^{\text{IGM}}$ function class achieve the state-of-the-art performance, but offline VDN performs poorly and cannot utilize well the offline dataset collected by an unfamiliar behavior policy. This considerable performance gap between deep multi-agent Q-learning with IGM and linear value factorization indicates that the expressiveness of value factorization structures dramatically affects the performance, and IGM value factorization is a promising choice of value factorization for offline multi-agent reinforcement learning.

**Remark on Experiment Settings.** Similar experiment results are observed in Dec-POMDP settings by prior work [19]. To make the empirical study rigorous, we slightly modify the observation emission rule of SMAC tasks to create a suite of Dec-ROMDP environments. More specifically, we concatenate the local observations with the global latent state vector to ensure rich observations. We follow the evaluation setting used by [19] to establish a clear comparison under offline learning. The offline dataset is collected by training a VDN agent and adopting its full experience buffer. This buffer would include the mixture of a series of policies along with the whole learning procedure and also contain the exploration trajectories. Each curve is plotted by the median performance of running with multiple random seeds on three independently collected datasets. Such diverse datasets can reduce the unpredictable effects of irrelevant factors [25, 54] such as extrapolation error and the variance of dataset collection. A detailed description of the experiment setting is deferred to Appendix E.

## 7 Conclusion

This paper proposes a unified framework for analyzing cooperative multi-agent Q-learning with value factorization, which is a first work to make an initial effort to provide a theoretical understanding of this branch of methods. We analyze two classes of value factorization and investigate their algorithmic properties. The derived implications of our theoretical results are supported by experiments with deep-learning-based implementations. To close this paper, we connect our results with additional related literature as a discussion of future work.

**Other Value Factorization Approaches.** In addition to linear and IGM value decomposition discussed in this paper, there are a variety of value factorization methods have been proposed to trade off the computational complexity and function expressiveness. The monotonic value decomposition used by QMIX [12] is a popular factorization structure that induces many variants [45, 55–57]. The graph-based value factorization used by deep coordination graph (DCG) [58] provides a unified framework to extend high-order value factorization. These factorization mechanisms aim to capture some special structures of multi-agent learning. Investigating their algorithmic properties would provide deeper insights to understand state-of-the-art MARL algorithms.

**Generic Algorithm Formulation.** In this paper, we focus on the analysis of multi-agent fitted Q-iteration, which gives a theoretical characterization for the functionality of value factorization. One limitation of our analyses is that the formulation of fitted Q-iteration excludes several algorithmic components of reinforcement learning, such as exploration and optimization. As a first-step result, FQI serves a clear framework to support analyses of multi-agent value factorization but does not capture the effects of other algorithmic components of general Q-learning algorithms. An important future work is to investigate how value factorization methods interact with poorly explored datasets and gradient-based optimization [25]. In addition, value factorization is also adopted by actor-critic-based MARL algorithms [32, 59, 60], which poses another problem for future studies.

**Sample Complexity Analysis.** Finite-sample analysis of fully centralized multi-agent algorithms has been studied in the literature of PAC reinforcement learning. Several prior work [61–64] proposed provably efficient reinforcement learning algorithms for factored MDPs in terms of regret and sample complexity. However, these algorithms are established in an information-theoretic manner and are not computationally efficient. From this perspective, introducing the concept of value factorization is a promising way to improve the computational efficiency of multi-agent algorithms in PAC learning settings.

## Acknowledgments and Disclosure of Funding

The authors would like to thank the anonymous reviewers and Kefan Dong for their insightful discussions and helpful suggestions. This work is supported in part by Science and Technology Innovation 2030 – "New Generation Artificial Intelligence" Major Project (No. 2018AAA0100904), a grant from the Institute of Guo Qiang, Tsinghua University, and a grant from Turing AI Institute of Nanjing.

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
