# A  Formal Descriptions of Problem Formulation and Assumptions

Note that, learning policies for sequential decision making in Dec-POMDP [35] is known to be undecidable in the standard computation model [38]. i.e., deciding whether there is a policy achieving a constant performance lower bound cannot be computed in finite time. To bypass these impossibility results, we adopt some common assumptions to make the analysis accessible and rigorous. A formal description of our assumptions is included in Appendix A.1, which are widely assumed in prior work on analyzing single-agent PAC reinforcement learning [24, 33, 34]. The main purpose of introducing these assumptions is to avoid discussing trajectories with infinite length, which is not well-defined in the context of data-driven learning paradigm.

## A.1  Decentralized Rich-Observation Markov Decision Process

The first step to exclude the discussion of infinitely long trajectories is to ensure that the belief states can always be constructed using finite recent observations. A widely-used formulation of such assumptions is Rich-Observation MDPs [33, 34], in which the observation would be noisy but guarantee to contain full information. Formally, we extend the formulation Rich-Observation MDPs (ROMDP) to a decentralized multi-agent setting as the following definition:

**Definition 3** (Dec-ROMDP). *An instance of Decentralized Rich-Observation Markov Decision Process (Dec-ROMDP) is defined as a tuple $\mathcal{M} = \langle \mathcal{N}, \mathcal{S}, \mathcal{X}, \mathcal{A}, P, \Lambda, r, \gamma \rangle$, in which*

- *$\mathcal{N} \equiv \{1, \dots, n\}$ denotes a finite set of agents.*

- *$\mathcal{S}$ denotes a finite set of global latent states.*

- *$\mathcal{X}$ denotes the observation space (a.k.a. context space) which is larger than the latent state space.*

- *$\mathcal{A}$ denotes the individual action space. The joint action $\mathbf{a} \in \mathbf{A} \equiv \mathcal{A}^n$ is a collection of individual actions $[a_i]_{i=1}^n$.*

- *$P(s' \mid s, \mathbf{a})$ denotes the transition function on latent states.*

- *$\Lambda(x_i \mid i, s)$ denotes the observation emission distribution, which may defer in different agents.*

- *$r(s, \mathbf{a})$ denotes the global reward function.*

- *$\gamma$ denotes the discount factor.*

*The concept of rich-observation assumes that the observation emission distributions $\Lambda$ are disjoint across states. Formally,*

- *$\forall s_1 \neq s_2 \in \mathcal{S}, \forall i \in \mathcal{N}, \forall x \in \mathcal{X}, (\Lambda(x \mid i, s_1) > 0) \Rightarrow (\Lambda(x \mid i, s_2) = 0)$.*

Note that, in rich-observation models [33, 34], the size of observation space $\mathcal{X}$ can be much larger than the latent state space $\mathcal{S}$. This rich-observation model is a formulation of real-world scenarios where sensors suffer from systematical errors, and it is usually used to study how function approximators can help to decode the state information from observations [39, 40]. The formulation of Dec-ROMDP stated as Definition 3 considers an extension of ROMDP upon Dec-POMDP, where different agents would observe the global information under different noises. This decentralized observation emission distribution $\Lambda(x \mid i, s)$ is an important characteristic and also a critical subtlety for analyzing a multi-agent system.

The definition of Dec-ROMDP enables the agent extract the belief state information without storing the whole infinite-length trajectory. Formally, we adopt the definition of reactive function classes [33, 36, 37] to simplify the notation for analyses.

**Definition 4** (Reactive Function Class). *Reactive function classes consider a set of memoryless functions that only takes the most recent observations as inputs to compute the output values.*

The same function class structure is widely-used in prior work studying single-agent ROMDP [24, 33, 65]. Note that, in Dec-ROMDPs, decoding information from last-step observations guarantees to retain full information of the latent states. In addition, this simplification is equivalent to storing recent observations in a constant-size time window, since we can convert such multi-step model to last-step model by encoding recent steps into the latent state representation and overwriting the observation emission to a window of observations. Regarding this equivalence, we adopt the most simplified notations as related work to make the underlying insights more accessible.

## A.2  Connection to Practical Scenarios

The main purpose to introduce the concept of rich observations is to bypassing the barrier of partial observability in theoretical analyses, since general POMDP-based problem formulation can hardly support algorithms for strong performance guarantees. In practice, although most scenarios suffer from some extend of partial observability, the formulation of richly observable environments can approximate many application scenarios.

**Micromanagement Tasks.** In Micromanagement tasks such as SMAC benchmark tasks [20], agents are working in a bounded region with a sufficiently large receptive field. In such tasks, the observations of agents may be high-dimensional and redundant (e.g., visual observations with various camera directions), the global latent state space is bounded. For example, in StarCraft II benchmark tasks, agents are assigned large receptive fields to provide sufficient information for decision making, i.e., every alive agent can nearly observe all other agents. In this situation, the observation history of each agent can nearly decode the global latent state, which can be approximated by the problem formulation of Dec-ROMDPs.

**Coordination with Communication.** In practice, partial observability is not a hard constraint. When agents can only access their local observations, learning communication is a common approach to reducing partial observability issues. e.g., NDQ [15] aims to learn an efficient communication protocol that transmits useful and necessary information to address partial observability. When such communication channels are available, every agent can collect redundant message information from other agents to transform Dec-POMDPs to Dec-ROMDPs.

## A.3 Omitted Proofs for the Equivalence Mentioned in Section 4

**Lemma 1.** *The empirical Bellman operator $\mathcal{T}_D^{FMA}$ defined in Eq. (3) and Eq. (6) are equivalent. Formally,*

$$
\begin{aligned}
\mathcal{T}_D^{FMA} Q^{(t)} &\equiv \underset{Q \in \mathcal{Q}^{FMA}}{\arg\min} \ \underset{(\boldsymbol{x},\mathbf{a},r,\boldsymbol{x}')\sim D}{\mathbb{E}} \left( \hat{y}^{(t)}(\boldsymbol{x},\mathbf{a},\boldsymbol{x}') - Q_{tot}(\boldsymbol{x},\mathbf{a}) \right)^2 \\
&= \underset{Q \in \mathcal{Q}^{FMA}}{\arg\min} \ \underset{(\boldsymbol{x},\mathbf{a},r)\sim D}{\mathbb{E}} \left( y^{(t)}(\boldsymbol{x},\mathbf{a}) - Q_{tot}(\boldsymbol{x},\mathbf{a}) \right)^2,
\end{aligned}
\tag{11}
$$

*where*

$$
\begin{aligned}
\hat{y}^{(t)}(\boldsymbol{x},\mathbf{a},\boldsymbol{x}') &= r + \gamma \max_{\mathbf{a}'} Q_{tot}^{(t)}\left(\boldsymbol{x}',\mathbf{a}'\right), \\
y^{(t)}(\boldsymbol{x},\mathbf{a}) &= r + \gamma \mathbb{E}_{\boldsymbol{x}'}\left[ \max_{\mathbf{a}'} Q_{tot}^{(t)}\left(\boldsymbol{x}',\mathbf{a}'\right) \right].
\end{aligned}
\tag{12}
$$

*Proof.* Consider

$$
\begin{aligned}
\mathcal{T}_D^{\text{FMA}} Q^{(t)} &\equiv \underset{Q \in \mathcal{Q}^{\text{FMA}}}{\arg\min} \ \underset{(\boldsymbol{x},\mathbf{a},\boldsymbol{x}')\sim D}{\mathbb{E}} \left[ \left( \hat{y}^{(t)}(\boldsymbol{x},\mathbf{a},\boldsymbol{x}') - Q_{\text{tot}}(\boldsymbol{x},\mathbf{a}) \right)^2 \right] \\
&= \underset{Q \in \mathcal{Q}^{\text{FMA}}}{\arg\min} \ \underset{(\boldsymbol{x},\mathbf{a},\boldsymbol{x}')\sim D}{\mathbb{E}} \left[ \left( \hat{y}^{(t)}(\boldsymbol{x},\mathbf{a},\boldsymbol{x}') - y^{(t)}(\boldsymbol{x},\mathbf{a}) + y^{(t)}(\boldsymbol{x},\mathbf{a}) - Q_{\text{tot}}(\boldsymbol{x},\mathbf{a}) \right)^2 \right] \\
&= \underset{Q \in \mathcal{Q}^{\text{FMA}}}{\arg\min} \ \underset{(\boldsymbol{x},\mathbf{a},\boldsymbol{x}')\sim D}{\mathbb{E}} \left[ \left( \hat{y}^{(t)}(\boldsymbol{x},\mathbf{a},\boldsymbol{x}') - y^{(t)}(\boldsymbol{x},\mathbf{a}) \right)^2 \right] \\
&\quad + \underset{(\boldsymbol{x},\mathbf{a},\boldsymbol{x}')\sim D}{\mathbb{E}} \left[ 2 \left( \hat{y}^{(t)}(\boldsymbol{x},\mathbf{a},s') - y^{(t)}(\boldsymbol{x},\mathbf{a}) \right) \left( y^{(t)}(\boldsymbol{x},\mathbf{a}) - Q_{\text{tot}}(\boldsymbol{x},\mathbf{a}) \right) \right] \\
&\quad + \underset{(\boldsymbol{x},\mathbf{a},\boldsymbol{x}')\sim D}{\mathbb{E}} \left[ \left( y^{(t)}(\boldsymbol{x},\mathbf{a}) - Q_{\text{tot}}(\boldsymbol{x},\mathbf{a}) \right)^2 \right].
\end{aligned}
\tag{13}
$$

The first term is a constant since $y^{(t)}$ and $\hat{y}^{(t)}$ are fixed targets.

The second term is equal to zero since

$$
\begin{aligned}
&\underset{(\boldsymbol{x},\mathbf{a},\boldsymbol{x}')\sim D}{\mathbb{E}} \left[ 2 \left( \hat{y}^{(t)}(\boldsymbol{x},\mathbf{a},\boldsymbol{x}') - y^{(t)}(\boldsymbol{x},\mathbf{a}) \right) \left( y^{(t)}(\boldsymbol{x},\mathbf{a}) - Q_{\text{tot}}(\boldsymbol{x},\mathbf{a}) \right) \right] \\
&= 2 \underset{(\boldsymbol{x},\mathbf{a})\sim D}{\mathbb{E}} \Bigg[ \underbrace{\underset{\boldsymbol{x}'\sim P(\cdot|\boldsymbol{x},\mathbf{a})}{\mathbb{E}} \left[ \hat{y}^{(t)}(\boldsymbol{x},\mathbf{a},\boldsymbol{x}') - y^{(t)}(\boldsymbol{x},\mathbf{a}) \right]}_{=0} \left( y^{(t)}(\boldsymbol{x},\mathbf{a}) - Q_{\text{tot}}(\boldsymbol{x},\mathbf{a}) \right) \Bigg] \\
&= 0.
\end{aligned}
\tag{14}
$$

The third term exactly corresponds to Eq. (11). □

# B  Omitted Proofs in Section 5.2

## B.1  The Closed-Form Solution to A Special Weighted Linear Regression

**Lemma 2.** *Considering following weighted linear regression problem*

$$\min_{\mathbf{x}} \| \sqrt{\mathbf{p}^\top} \cdot (\mathbf{A}\mathbf{x} - \mathbf{b}) \|_2^2 \tag{15}$$

*where $\mathbf{A} \in \mathbb{R}^{m^n \times mn}, \mathbf{x} \in \mathbb{R}^{mn}, \mathbf{b}, \mathbf{p} \in \mathbb{R}^{m^n}, m, n \in \mathbb{Z}^+$. Besides, $\mathbf{A}$ is m-ary encoding matrix namely $\forall i \in [m^n], j \in [mn]$*

$$\mathbf{A}_{i,j} = \begin{cases} 1, & if \quad \exists u \in [n], j = m \times u + (\lfloor i/m^u \rfloor \bmod m), \\ 0, & otherwise. \end{cases} \tag{16}$$

*For simplicity, $j^{th}$ row of $\mathbf{A}$ corresponds to a m-ary number $\vec{a}_j = (j)_m$ where $\vec{a} = a_0 a_1 \ldots a_{n-1}$, with $a_u \in [m], \forall u \in [n]$. Assume $\mathbf{p}$ is a positive vector which follows that*

$$\mathbf{p}_j = \mathbf{p}(\vec{a}_j) = \prod_{u \in [n]} p_u(a_{u,j}), \text{ where } p_u : [m] \to (0,1) \text{ and } \sum_{a_u \in [m]} p_u(a_u) = 1, \forall u \in [n] \tag{17}$$

*The optimal solution of this problem is the following. Denote $i = u \times m + v, v \in [m], u \in [n]$ and an arbitrary vector $\mathbf{w} \in \mathbb{R}^{mn}$*

$$\mathbf{x}_i^* = \sum_{\vec{a}} \frac{\mathbf{p}(\vec{a})}{p_u(a_u)} \mathbf{b}_{\vec{a}} \cdot \mathbf{1}(a_u = v) - \frac{n-1}{n} \mathbf{p}(\vec{a}) \mathbf{b}_{\vec{a}} - \frac{1}{mn} \sum_{i' \in [mn]} \mathbf{w}_{i'} + \frac{1}{m} \sum_{v' \in [m]} \mathbf{w}_{um+v'} \tag{18}$$

*Proof.* For brevity, denote

$$\mathbf{A}^p = \sqrt{\mathbf{p}^\top} \cdot \mathbf{A}, \qquad \mathbf{b}^p = \sqrt{\mathbf{p}^\top} \cdot \mathbf{b} \tag{19}$$

Then the weighted linear regression becomes a standard Linear regression problem w.r.t $\mathbf{A}^p, \mathbf{b}^p$. To compute the optimal solutions, we need to calculate the Moore-Penrose inverse of $\mathbf{A}^p$. The sufficient and necessary condition of this inverse matrix $\mathbf{A}^{p,\dagger} \in \mathbb{R}^{mn \times m^n}$ is the following three statements [42]:

$$(1) \ \mathbf{A}^p \mathbf{A}^{p,\dagger} \text{ and } \mathbf{A}^{p,\dagger} \mathbf{A}^p \text{ are self-adjoint} \tag{20}$$

$$(2) \ \mathbf{A}^p = \mathbf{A}^p \mathbf{A}^{p,\dagger} \mathbf{A}^p \tag{21}$$

$$(3) \ \mathbf{A}^{p,\dagger} = \mathbf{A}^{p,\dagger} \mathbf{A}^p \mathbf{A}^{p,\dagger} \tag{22}$$

We consider the following matrix as $\mathbf{A}^{p,\dagger}$ and we prove that it satisfies all three statements. For $\forall i \in [mn], i = u \times m + v, u \in [n], v \in [m], j \in [m^n]$

$$\begin{aligned}
\mathbf{A}_{i,j}^{p,\dagger} &= \mathbf{A}_{i,\vec{a}_j}^{p,\dagger} \\
&= \sqrt{\frac{\mathbf{p}(\vec{a}_{-u,j})}{p_u(a_{u,j})}} \cdot \mathbf{1}(a_{u,j} = v) - \frac{n-1}{n} \sqrt{\mathbf{p}(\vec{a}_j)} - \frac{1}{m} \sqrt{\frac{\mathbf{p}(\vec{a}_{-u,j})}{p_u(a_{u,j})}} + \frac{1}{mn} \sum_{u'=0}^{n-1} \sqrt{\frac{\mathbf{p}(\vec{a}_{-u',j})}{p_{u'}(a_{u',j})}}
\end{aligned} \tag{23}$$

where $\mathbf{p}(\vec{a}_{-u}) = \prod_{u' \neq u} p_{u'}(a_{u'})$.

First, we verify that $\mathbf{A}^p \mathbf{A}^{p,\dagger}$ is a $m^n \times m^n$ self-adjoint matrix in statement (1). For simplicity, $O(\vec{a}_i, \vec{a}_j) = \{u | a_{u,i} = a_{u,j}, u \in [n]\}$.

$$\begin{aligned}
(\mathbf{A}^p \mathbf{A}^{p,\dagger})_{i,j} &= \sum_{u \in [n]} \sqrt{\mathbf{p}(\vec{a}_i)} \Big[ \sqrt{\frac{\mathbf{p}(\vec{a}_{-u,j})}{p_u(a_{u,j})}} \cdot \mathbf{1}(a_{u,j} = a_{u,i}) - \frac{n-1}{n} \sqrt{\mathbf{p}(\vec{a}_j)} - \frac{1}{m} \sqrt{\frac{\mathbf{p}(\vec{a}_{-u,j})}{p_u(a_{u,j})}} \\
&\quad + \frac{1}{mn} \sum_{u'=0}^{n-1} \sqrt{\frac{\mathbf{p}(\vec{a}_{-u',j})}{p_{u'}(a_{u',j})}} \Big] \\
&= \sum_{u \in O(\vec{a}_i, \vec{a}_j)} \frac{\sqrt{\mathbf{p}(\vec{a}_j)\mathbf{p}(\vec{a}_i)}}{p_u(a_{u,j})} - \frac{n-1}{n} \sum_{u \in [n]} \sqrt{\mathbf{p}(\vec{a}_i)\mathbf{p}(\vec{a}_j)} - \frac{1}{m} \sum_{u \in [n]} \frac{\sqrt{\mathbf{p}(\vec{a}_j)\mathbf{p}(\vec{a}_i)}}{p_u(a_{u,j})}
\end{aligned}$$

$$+ \sum_{u \in [n]} \frac{1}{mn} \sum_{u'=0}^{n-1} \frac{\sqrt{\mathbf{p}(\vec{a}_j)\mathbf{p}(\vec{a}_i)}}{p_{u'}(a_{u',j})}$$

$$= \sum_{u \in O(\vec{a}_i, \vec{a}_j)} \frac{\sqrt{\mathbf{p}(\vec{a}_{j'})\mathbf{p}(\vec{a}_i)}}{p_u(a_{u,j})} - (n-1)\sqrt{\mathbf{p}(\vec{a}_i)\mathbf{p}(\vec{a}_j)} - \frac{1}{m} \sum_{u \in [n]} \frac{\sqrt{\mathbf{p}(\vec{a}_j)\mathbf{p}(\vec{a}_i)}}{p_u(a_{u,j})}$$

$$+ \frac{1}{m} \sum_{u \in [n]} \frac{\sqrt{\mathbf{p}(\vec{a}_j)\mathbf{p}(\vec{a}_i)}}{p_u(a_{u,j})}$$

$$= \sum_{u \in O(\vec{a}_i, \vec{a}_j)} \frac{\sqrt{\mathbf{p}(\vec{a}_j)\mathbf{p}(\vec{a}_i)}}{p_u(a_{u,j})} - (n-1)\sqrt{\mathbf{p}(\vec{a}_i)\mathbf{p}(\vec{a}_j)} \tag{24}$$

Observe that $p_u(a_{u,j}) = p_u(a_{u,i})$ if $a_{u,i} = a_{u,j}$, thus $(\mathbf{A}^p \mathbf{A}^{p,\dagger})_{i,j} = (\mathbf{A}^p \mathbf{A}^{p,\dagger})_{j,i}$ for any $i, j \in [m^n]$. This proves that $\mathbf{A}^p \mathbf{A}^{p,\dagger}$ is self-adjoint.

Second, we prove that $\mathbf{A}^{p,\dagger} \mathbf{A}^p$ is a $mn \times mn$ self-adjoint matrix and has surprisingly succinct form. Let $i = u \times m + v, u \in [n], v \in [m]$.

1. $i = i'$. Besides, $O(i) = \{\vec{a} \in [m^n] | a_u = v\}$

$$(\mathbf{A}^{p,\dagger} \mathbf{A}^p)_{i,i} = \sum_{\vec{a} \in O(i)} \sqrt{\mathbf{p}(\vec{a})}[\sqrt{\frac{\mathbf{p}(\vec{a}_{-u})}{p_u(a_u)}} \cdot \mathbf{1}(a_u = v) - \frac{n-1}{n}\sqrt{\mathbf{p}(\vec{a})} - \frac{1}{m}\sqrt{\frac{\mathbf{p}(\vec{a}_{-u})}{p_u(a_u)}}$$

$$+ \frac{1}{mn} \sum_{u'=0}^{n-1} \sqrt{\frac{\mathbf{p}(\vec{a}_{-u'})}{p_{u'}(a_{u'})}}]$$

$$= \sum_{\vec{a} \in O(i)} \frac{\mathbf{p}(\vec{a})}{p_u(a_u)} - \frac{n-1}{n}\mathbf{p}(\vec{a}) - \frac{1}{m}\frac{\mathbf{p}(\vec{a})}{p_u(a_u)} + \frac{1}{mn}\sum_{u'=0}^{n-1}\frac{\mathbf{p}(\vec{a})}{p_{u'}(a_{u'})}$$

$$= \sum_{\vec{a} \in O(i)} \left( \mathbf{p}(\vec{a}_{-u}) - \frac{1}{m}\mathbf{p}(\vec{a}_{-u}) + \frac{1}{mn}\sum_{u'=0}^{n-1}\mathbf{p}(\vec{a}_{-u'}) \right) - \frac{n-1}{n}p_u(a_u = v)$$

$$= 1 - \frac{1}{m} - \frac{n-1}{n}p_u(a_u = v) + \frac{1}{mn}\sum_{\substack{u' \in [n] \\ u' \neq u}} \sum_{\vec{a} \in O(i)} \mathbf{p}(\vec{a}_{-u'})$$

$$+ \frac{1}{mn} \sum_{\vec{a} \in O(i)} \mathbf{p}(\vec{a}_{-u})$$

$$= 1 - \frac{1}{m} - \frac{n-1}{n}p_u(a_u = v) + \frac{1}{mn} + \frac{n-1}{mn}mp_u(a_u = v)$$

$$= 1 - \frac{1}{m} + \frac{1}{mn} \tag{25}$$

2. $i = u \times m + v, i' = u \times m + v', v \neq v'$. This implies that $Q(i) \cap O(i') = \emptyset$

$$(\mathbf{A}^{p,\dagger} \mathbf{A}^p)_{i,i'} = \sum_{\vec{a} \in O(i')} \sqrt{\mathbf{p}(\vec{a})}[\sqrt{\frac{\mathbf{p}(\vec{a}_{-u})}{p_u(a_u)}} \cdot \mathbf{1}(a_u = v) - \frac{n-1}{n}\sqrt{\mathbf{p}(\vec{a})}$$

$$- \frac{1}{m}\sqrt{\frac{\mathbf{p}(\vec{a}_{-u})}{p_u(a_u)}} + \frac{1}{mn}\sum_{u'=0}^{n-1}\sqrt{\frac{\mathbf{p}(\vec{a}_{-u'})}{p_{u'}(a_{u'})}}]$$

$$= \sum_{\vec{a} \in O(i) \cap O(i')} \frac{\mathbf{p}(\vec{a})}{p_u(a_u)} - \frac{n-1}{n}\sum_{\vec{a} \in O(i')}\mathbf{p}(\vec{a}) - \frac{1}{m}\sum_{\vec{a} \in O(i')}\frac{\mathbf{p}(\vec{a})}{p_u(a_u)}$$

$$+ \frac{1}{mn}\sum_{\substack{u' \in [n] \\ u' \neq u}}\sum_{\vec{a} \in O(i')}\frac{\mathbf{p}(\vec{a})}{p_{u'}(a_{u'})} + \frac{1}{mn}\sum_{\vec{a} \in O(i')}\frac{\mathbf{p}(\vec{a})}{p_u(a_u)}$$

$$= -\frac{n-1}{n}p_u(a_u = v') - \frac{1}{m} + \frac{n-1}{mn}\sum_{\vec{a}\in O(i')}\mathbf{p}(\vec{a}_{-u'}) + \frac{1}{mn}$$

$$= -\frac{1}{m} + \frac{1}{mn} \tag{26}$$

3. $i = u_1 \times m + v_1, i' = u_2 \times m + v_2, u_1 \neq u_2$.

$$\begin{aligned}
(\mathbf{A}^{p,\dagger}\mathbf{A}^p)_{i,i'} &= \sum_{\vec{a}\in O(i')}\sqrt{\mathbf{p}(\vec{a})}\Big[\sqrt{\frac{\mathbf{p}(\vec{a}_{-u_1})}{p_{u_1}(a_{u_1})}}\cdot\mathbf{1}(a_{u_1}=v) - \frac{n-1}{n}\sqrt{\mathbf{p}(\vec{a})} \\
&\quad - \frac{1}{m}\sqrt{\frac{\mathbf{p}(\vec{a}_{-u_1})}{p_{u_1}(a_{u_1})}} + \frac{1}{mn}\sum_{u'=0}^{n-1}\sqrt{\frac{\mathbf{p}(\vec{a}_{-u'})}{p_{u'}(a_{u'})}}\Big] \\
&= \sum_{\vec{a}\in O(i)\cap O(i')}\frac{\mathbf{p}(\vec{a})}{p_{u_1}(a_{u_1})} - \frac{n-1}{n}\sum_{\vec{a}\in O(i')}\mathbf{p}(\vec{a}) - \frac{1}{m}\sum_{\vec{a}\in O(i')}\frac{\mathbf{p}(\vec{a})}{p_{u_1}(a_{u_1})} \\
&\quad + \frac{1}{mn}\sum_{\substack{u'\in[n]\\u'\neq u_2}}\sum_{\vec{a}\in O(i')}\frac{\mathbf{p}(\vec{a})}{p_{u'}(a_{u'})} + \frac{1}{mn}\sum_{\vec{a}\in O(i')}\frac{\mathbf{p}(\vec{a})}{p_{u_2}(a_{u_2})} \\
&= p_{u_2}(a_{u_2}) - \frac{n-1}{n}p_{u_2}(a_{u_2}) - p_{u_2}(a_{u_2}) + \frac{n-1}{mn}mp_{u_2}(a_{u_2}) + \frac{1}{mn} \\
&= \frac{1}{mn} \tag{27}
\end{aligned}$$

Observe that $\mathbf{A}^{p,\dagger}\mathbf{A}^p$ is self-adjoint by equation (2,3,4) and the expression is succinct.

Third, we verify statement (2). Since we have computed $\mathbf{A}^{p,\dagger}\mathbf{A}^p$, the verification is straightforward. For brevity, denote $\mathbf{A}^{p,\dagger}\mathbf{A}^p$ as $\mathbf{A}_0^p$

$$\begin{aligned}
(\mathbf{A}^p\mathbf{A}_0^p)_{\vec{a},i} &= \sqrt{\mathbf{p}(\vec{a})}\sum_{u\in[n]}(\mathbf{A}_0^p)_{um+a_u,i} \\
&= \sqrt{\mathbf{p}(\vec{a})}\left(\mathbf{1}(\exists u\in[n], i = um+a_u) - \frac{1}{m} + \frac{1}{mn} + (n-1)\frac{1}{mn}\right) \\
&= \sqrt{\mathbf{p}(\vec{a})}\cdot\mathbf{1}(\exists u\in[n], i = um+a_u) \tag{28}
\end{aligned}$$

Thus, $\mathbf{A}^p\mathbf{A}^{p,\dagger}\mathbf{A}^p = \mathbf{A}^p$.

Similarly, we can verify statement (3). Suppose $i_0 = u_0 \times m + v_0$, we have

$$\begin{aligned}
(\mathbf{A}_0^p\mathbf{A}^{p,\dagger})_{i_0,\vec{a}} &= \frac{1}{mn}\sum_{\substack{u\neq u_0\\u\in[n]}}\sum_{v\in[m]}\Big[\sqrt{\frac{\mathbf{p}(\vec{a}_{-u})}{p_u(a_u)}}\cdot\mathbf{1}(a_u = v) \\
&\quad - \frac{n-1}{n}\sqrt{\mathbf{p}(\vec{a})} - \frac{1}{m}\sqrt{\frac{\mathbf{p}(\vec{a}_{-u})}{p_u(a_u)}} + \frac{1}{mn}\sum_{u'=0}^{n-1}\sqrt{\frac{\mathbf{p}(\vec{a}_{-u'})}{p_{u'}(a_{u'})}}\Big] \\
&\quad + \sum_{v\in[m]}(\mathbf{1}(v = v_0) - \frac{1}{m} + \frac{1}{mn})\Big[\sqrt{\frac{\mathbf{p}(\vec{a}_{-u_0})}{p_{u_0}(a_{u_0})}}\cdot\mathbf{1}(a_{u_0} = v) \\
&\quad - \frac{n-1}{n}\sqrt{\mathbf{p}(\vec{a})} - \frac{1}{m}\sqrt{\frac{\mathbf{p}(\vec{a}_{-u_0})}{p_{u_0}(a_{u_0})}} + \frac{1}{mn}\sum_{u'=0}^{n-1}\sqrt{\frac{\mathbf{p}(\vec{a}_{-u'})}{p_{u'}(a_{u'})}}\Big] \\
&= \frac{1}{mn}\sum_{u\in[n]}\sum_{v\in[m]}\Big[\sqrt{\frac{\mathbf{p}(\vec{a}_{-u})}{p_u(a_u)}}\cdot\mathbf{1}(a_u = v)
\end{aligned}$$

$$
-\frac{n-1}{n}\sqrt{\mathbf{p}(\vec{a})}-\frac{1}{m}\sqrt{\frac{\mathbf{p}(\vec{a}_{-u})}{p_u(a_u)}}+\frac{1}{mn}\sum_{u'=0}^{n-1}\sqrt{\frac{\mathbf{p}(\vec{a}_{-u'})}{p_{u'}(a_{u'})}}]
$$

$$
+\sum_{v\in[m]}(\mathbf{1}(v=v_0)-\frac{1}{m})[-\frac{n-1}{n}\sqrt{\mathbf{p}(\vec{a})}-\frac{1}{m}\sqrt{\frac{\mathbf{p}(\vec{a}_{-u_0})}{p_{u_0}(a_{u_0})}}
$$

$$
+\frac{1}{mn}\sum_{u'=0}^{n-1}\sqrt{\frac{\mathbf{p}(\vec{a}_{-u'})}{p_{u'}(a_{u'})}}]+\sum_{v\in[m]}(\mathbf{1}(v=v_0)-\frac{1}{m})\sqrt{\frac{\mathbf{p}(\vec{a}_{-u_0})}{p_{u_0}(a_{u_0})}}\cdot\mathbf{1}(a_{u_0}=v)
$$

$$
=\frac{1}{mn}\sum_{u\in[n]}\sqrt{\frac{\mathbf{p}(\vec{a}_{-u})}{p_u(a_u)}}-\frac{n-1}{n}\sqrt{\mathbf{p}(\vec{a})}
$$

$$
+\frac{1}{n}\sum_{u\in[n]}[-\frac{1}{m}\sqrt{\frac{\mathbf{p}(\vec{a}_{-u})}{p_u(a_u)}}+\frac{1}{mn}\sum_{u'=0}^{n-1}\sqrt{\frac{\mathbf{p}(\vec{a}_{-u'})}{p_{u'}(a_{u'})}}]
$$

$$
+\left(\sum_{v\in[m]}(\mathbf{1}(v=v_0)-\frac{1}{m})\right)[-\frac{n-1}{n}\sqrt{\mathbf{p}(\vec{a})}-\frac{1}{m}\sqrt{\frac{\mathbf{p}(\vec{a}_{-u_0})}{p_{u_0}(a_{u_0})}}
$$

$$
+\frac{1}{mn}\sum_{u'=0}^{n-1}\sqrt{\frac{\mathbf{p}(\vec{a}_{-u'})}{p_{u'}(a_{u'})}}]+(\mathbf{1}(a_{u_0}=v_0)-\frac{1}{m})\sqrt{\frac{\mathbf{p}(\vec{a}_{-u_0})}{p_{u_0}(a_{u_0})}}
\tag{29}
$$

Clearly, we have the following relations

$$
\sum_{u\in[n]}[-\frac{1}{m}\sqrt{\frac{\mathbf{p}(\vec{a}_{-u})}{p_u(a_u)}}+\frac{1}{mn}\sum_{u'=0}^{n-1}\sqrt{\frac{\mathbf{p}(\vec{a}_{-u'})}{p_{u'}(a_{u'})}}]=0
\tag{30}
$$

$$
\sum_{v\in[m]}(\mathbf{1}(v=v_0)-\frac{1}{m})=0
\tag{31}
$$

Thus

$$
(\mathbf{A}_0^p\mathbf{A}^{p,\dagger})_{i_0,\vec{a}}=\frac{1}{mn}\sum_{u\in[n]}\sqrt{\frac{\mathbf{p}(\vec{a}_{-u})}{p_u(a_u)}}-\frac{n-1}{n}\sqrt{\mathbf{p}(\vec{a})}+(\mathbf{1}(a_{u_0}=v_0)-\frac{1}{m})\sqrt{\frac{\mathbf{p}(\vec{a}_{-u_0})}{p_{u_0}(a_{u_0})}}
\tag{32}
$$

$$
=\mathbf{A}_{i_0,\vec{a}}^{p,\dagger}
\tag{33}
$$

This proves $\mathbf{A}^{p,\dagger}=\mathbf{A}^{p,\dagger}\mathbf{A}^p\mathbf{A}^{p,\dagger}$ in statement (3) and $\mathbf{A}^{p,\dagger}$ is the Moore-Penrose inverse of $\mathbf{A}^p$. Since the optimal solution $\mathbf{x}^*=\mathbf{A}^{p,\dagger}\mathbf{b}^p+(\mathbf{I}_{mn\times mn}-\mathbf{A}^{p,\dagger}\mathbf{A}^p)\mathbf{w}$ where $w\in\mathrm{R}^{mn}$ is any vector [42].

Denote $\mathbf{x}^p=\mathbf{A}^{p,\dagger}\mathbf{b}^p$. We have $\forall i=u\times m+v$

$$
\mathbf{x}_i^p=\sum_{\vec{a}}\mathbf{A}_{i,\vec{a}}^{p,\dagger}\sqrt{\mathbf{p}(\vec{a})}\mathbf{b}_{\vec{a}}
$$

$$
=\sum_{\vec{a}}[\sqrt{\frac{\mathbf{p}(\vec{a}_{-u})}{p_u(a_u)}}\cdot\mathbf{1}(a_u=v)-\frac{n-1}{n}\sqrt{\mathbf{p}(\vec{a})}-\frac{1}{m}\sqrt{\frac{\mathbf{p}(\vec{a}_{-u})}{p_u(a_u)}}
$$

$$
+\frac{1}{mn}\sum_{u'=0}^{n-1}\sqrt{\frac{\mathbf{p}(\vec{a}_{-u'})}{p_{u'}(a_{u'})}}]\sqrt{\mathbf{p}(\vec{a})}\mathbf{b}_{\vec{a}}
$$

$$
=\sum_{\vec{a}}\left[\frac{\mathbf{p}(\vec{a})}{p_u(a_u)}\cdot\mathbf{1}(a_u=v)-\frac{n-1}{n}\mathbf{p}(\vec{a})-\frac{1}{m}\frac{\mathbf{p}(\vec{a})}{p_u(a_u)}+\frac{1}{mn}\sum_{u'=0}^{n-1}\frac{\mathbf{p}(\vec{a})}{p_{u'}(a_{u'})}\right]\mathbf{b}_{\vec{a}}
\tag{34}
$$

From equation (2, 3, 4), we have $i=u\times m+v, i'=u'\times m+v'$

$$
(\mathbf{I}-\mathbf{A}^{p,\dagger}\mathbf{A}^p)_{i,i'}=\begin{cases}\frac{1}{m}-\frac{1}{mn}&if\ u=u'\\-\frac{1}{mn}&if\ u\neq u'\end{cases}
\tag{35}
$$

If we consider $\mathbf{w}$ as the following $i_0 = u_0 \times m + v_0$

$$\mathbf{w}_{i_0} = \sum_{\vec{a} \in O(i_0)} \frac{\mathbf{p}(\vec{a})}{p_{u_0}(a_{u_0})} \mathbf{b}_{\vec{a}} \qquad (36)$$

Then for $i = u \times m + v$

$$((\mathbf{I} - \mathbf{A}^{p,\dagger}\mathbf{A}^p)\mathbf{w})_i = \sum_{\substack{i_0 \in [mn] \\ u \neq u_0}} -\frac{1}{mn}\mathbf{w}_{i_0} + \sum_{i_0:u_0=u} (\frac{1}{m} - \frac{1}{mn})\mathbf{w}_{i_0} \qquad (37)$$

$$= \sum_{\vec{a}} -\frac{1}{mn} \sum_{u' \in [n]} \frac{\mathbf{p}(\vec{a})}{p_{u'}(a_{u'})}\mathbf{b}_{\vec{a}} + \frac{1}{m} \sum_{\vec{a}} \frac{\mathbf{p}(\vec{a})}{p_u(a_u)}\mathbf{b}_{\vec{a}} \qquad (38)$$

Notice that this is exactly the last two terms in equation (5). Therefore, the optimal solutions of this weighted linear regression problem can be written as: $i = u \times m + v, v \in [m], u \in [n]$ and an arbitrary vector $\mathbf{w} \in \mathbb{R}^{mn}$.

$$\mathbf{x}_i^* = \sum_{\vec{a}} \frac{\mathbf{p}(\vec{a})}{p_u(a_u)} \mathbf{b}_{\vec{a}} \cdot \mathbf{1}(a_u = v) - \frac{n-1}{n}\mathbf{p}(\vec{a})\mathbf{b}_{\vec{a}} - \frac{1}{mn} \sum_{i' \in [mn]} \mathbf{w}_{i'} + \frac{1}{m} \sum_{v' \in [m]} \mathbf{w}_{um+v'} \qquad (39)$$

This completes the proof. $\qquad\qquad\qquad\qquad\qquad\qquad\qquad\qquad\qquad\qquad\qquad\qquad\qquad\qquad\square$

### B.2 Omitted Proofs for Theorem 1

**Definition 1** (FQI-LVF). *FQI-LVF is an instance of FMA-FQI stated in Algorithm 1, which specifies the action-value function class with linear value factorization:*

$$\mathcal{Q}^{LVF} = \left\{ Q \mid Q_{tot}(\boldsymbol{x}, \mathbf{a}) = \sum_{i=1}^n Q_i(x_i, a_i), \ [Q_i]_{i=1}^n \in \mathbb{R}^{|\mathcal{X} \times \mathcal{A}|^n} \right\}. \qquad (7)$$

**Theorem 1.** *Let $Q^{(t+1)} = \mathcal{T}_D^{LVF} Q^{(t)}$ denote a single iteration of the empirical Bellman operator. $\forall i \in \mathcal{N}, \forall (\boldsymbol{x}, \mathbf{a}) \in \mathcal{X} \times \mathbf{A}$, the individual action-value function $Q_i^{(t+1)}(x_i, a_i)$ is updated to*

$$\underbrace{\mathbb{E}_{(x'_{-i}, a'_{-i}) \sim p_D(\cdot | x_i)}\left[ y^{(t)}\left(x_i \oplus x'_{-i}, a_i \oplus a'_{-i}\right)\right]}_{\text{evaluation of the individual action } a_i} - \frac{n-1}{n} \underbrace{\mathbb{E}_{\boldsymbol{x}', \mathbf{a}' \sim p_D(\cdot | \Lambda^{-1}(x_i))}\left[ y^{(t)}\left(\boldsymbol{x}', \mathbf{a}'\right)\right]}_{\text{counterfactual baseline}} + w_i(x_i), \qquad (8)$$

*where $z_i \oplus z'_{-i}$ denotes $\langle z'_1, \cdots, z'_{i-1}, z_i, z'_{i+1}, \cdots, z'_n \rangle$, and $z'_{-i}$ denotes the elements of all agents except for agent $i$. $\Lambda^{-1}(x_i)$ denotes the inverse of observation emission, which decodes the current latent state from $x_i$. The residue term $\mathbf{w} \equiv [w_i]_{i=1}^n$ is an arbitrary function satisfying $\forall \boldsymbol{x} \in \mathcal{X}, \sum_{i=1}^n w_i(x_i) = 0$.*

*Proof.* In the formulation of FQI-LVF stated in Definition 1, the empirical Bellman error minimization in FQI-LVF can be regarded as a set of weighted linear least squares problems as the following form:

$$\min_{\mathbf{x}} \| \sqrt{\mathbf{p}^\top} \cdot (\mathbf{A}\mathbf{x} - \mathbf{b}) \|_2^2. \qquad (40)$$

To construct such a linear regression problem, we first fix a latent state $s$.

- $n$ denotes the number of agents;

- $m$ denotes the number of individual observation-action pairs $(x_i, a_i)$ such that $x_i$ encodes the same latent state as $s$. We assume this amount is symmetric among agents to simplify the notations;

- $\mathbf{A} \in \mathbb{R}^{m^n \times mn}$ denotes the multi-agent credit assignment coefficient matrix of action-value functions with linear value decomposition;

- $\mathbf{x} \in \mathbb{R}^{mn}$ denotes individual action-value functions $\left[Q_i^{(t)}(x_i, a_i) \in \mathbb{R}^m\right]_{i=1}^n$ under the empirical Bellman error minimization;

- According to Lemma 1, $\mathbf{b} \in \mathbb{R}^{m^n}$ denotes the regression target $y^{(t)}(\boldsymbol{x}, \mathbf{a})$ derived by the *Bellman optimality operator*;

- $\mathbf{p} \in \mathbb{R}^{m^n}$ denotes the empirical probability of joint action $\mathbf{a}$ executed on observation $\boldsymbol{x}$, $p_D(\mathbf{a}|\boldsymbol{x})$, which can be factorized to the production of individual components illustrated in Assumption 1.

Besides, $\mathbf{A}$ is $m$-ary encoding matrix namely $\forall i \in [m^n], j \in [mn]$

$$\mathbf{A}_{i,j} = \begin{cases} 1, & \text{if} \quad \exists u \in [n], j = m \times u + (\lfloor i/m^u \rfloor \bmod m), \\ 0, & \text{otherwise.} \end{cases} \tag{41}$$

For simplicity, $j^{th}$ row of $\mathbf{A}$ corresponds to a m-ary number $\vec{a}_j = (j)_m$ where $\vec{a} = a_0 a_1 \ldots a_{n-1}$, with $a_u \in [m], \forall u \in [n]$. According to the factorizable empirical probability $p_D$ shown in Assumption 1, $\mathbf{p}$ is a corresponding positive vector which follows that

$$\mathbf{p}_j = \mathbf{p}(\vec{a}_j) = \prod_{u \in [n]} p_u(a_{u,j}), \; \text{where } p_u : [m] \to (0,1) \; and \sum_{a_u \in [m]} p_u(a_u) = 1, \forall u \in [n] \tag{42}$$

According to Lemma 2, we derive the optimal solution of this problem is the following. Denote $i = u \times m + v, v \in [m], u \in [n]$ and an arbitrary vector $\mathbf{w} \in \mathbb{R}^{mn}$

$$\mathbf{x}_i^* = \sum_{\vec{a}} \frac{\mathbf{p}(\vec{a})}{p_u(a_u)} \mathbf{b}_{\vec{a}} \cdot \mathbf{1}(a_u = v) - \frac{n-1}{n} \mathbf{p}(\vec{a}) \mathbf{b}_{\vec{a}} - \frac{1}{mn} \sum_{i' \in [mn]} \mathbf{w}_{i'} + \frac{1}{m} \sum_{v' \in [m]} \mathbf{w}_{um+v'} \tag{43}$$

which means $\forall i \in \mathcal{N}, \forall (\boldsymbol{x}, \mathbf{a}) \in \mathcal{X} \times \mathbf{A}$, the individual action-value function $Q_i^{(t+1)}(x_i, a_i) =$

$$\mathop{\mathbb{E}}_{(x'_{-i}, a'_{-i}) \sim p_D(\cdot | x_i)} \left[ y^{(t)} \left( x_i \oplus x'_{-i}, a_i \oplus a'_{-i} \right) \right] - \frac{n-1}{n} \mathop{\mathbb{E}}_{\boldsymbol{x}', \mathbf{a}' \sim p_D(\cdot | \Lambda^{-1}(x_i))} \left[ y^{(t)} \left( \boldsymbol{x}', \mathbf{a}' \right) \right] + w_i(x_i), \tag{44}$$

where $z_i \oplus z'_{-i}$ denotes $\langle z'_1, \cdots, z'_{i-1}, z_i, z'_{i+1}, \cdots, z'_n \rangle$, and $z'_{-i}$ denotes the elements of all agents except for agent $i$. $\Lambda^{-1}(x_i)$ denotes the inverse of observation emission, which decodes the current latent state from $x_i$. The residue term $\mathbf{w} \equiv [w_i]_{i=1}^n$ is an arbitrary function satisfying $\forall \boldsymbol{x} \in \mathcal{X}, \sum_{i=1}^n w_i(x_i) = 0$.

$\square$

## B.3 Equivalent between Least-Squares Problem and FQI-LVF in Dec-POMDPs

Recall that the closed-form solution derived in Theorem 1 relies on two assumptions, decentralized data collection and rich-observation problem formulation. In this section, we argue that, without either of these assumptions, an intuitive closed-form solution unlikely exists. Formally, we show that the problem complexity of solving one iteration of FQI-LVF is equivalent to solving general least-squares problems with binary weight matrices, which does not have existing analytical solutions. Proposition 3 characterizes the hardness of solving the closed-form solution of FQI-LVF without the rich-observation assumption. Proposition 4 characterizes the hardness without the decentralized data collection assumption.

**Proposition 3.** *Computing linear least-squares problems with binary weight matrices can be reduced to computing one iteration of FQI-LVF with some dataset collected in a Dec-POMDP with a decentralized policy.*

*Proof.* Consider the following linear least-squares problem:

$$(\mathbf{x}^*, b^*) = \arg\min_{\mathbf{x}, b} \sum_{j=1}^m w_j \cdot \left( \mathbf{c}_j^\top \mathbf{x} + b - y_j \right)^2, \tag{45}$$

where $\{\mathbf{c}_j\}_{j=1}^m$ are a set of given binary vectors, $\{y_j\}_{j=1}^m$ are given labels, and $\{w_j\}_{j=1}^m$ denote the weights of each data point.

Using linear least-squares to solve one-iteration of FQI-LVF is directed. We only need to consider how to solve the above least-squares problem using an oracle that can compute one-iteration of FQI-LVF without the assumption of rich observability.

We construct the dataset by executing a decentralized policy in the following Dec-POMDP:

- This Dec-POMDP contains $2^n$ latent state. Each latent state is represented by a binary string with length $n$. Recall that $n$ denotes the number of agents.

- Agent $i$ can observe the $i^{\text{th}}$ dimension of the latent state.

- At state $s = \mathbf{c}_j$, no matter executing what action, the transition would lead to a termination signal and the reward is equal to $y_j$. i.e., the transition function does not depend on the action. For brevity, we consider the individual action space $|\mathcal{A}| = 1$.

- The initial state distribution of state $s = \mathbf{c}_j$ is proportional to $w_j$.

Note that all actions are equivalent in this Dec-POMDP. Let $Q_i(0)$ and $Q_i(1)$ denote the individual value function of agent $i$ w.r.t. two local observations computed by FQI-LVF, i.e.,

$$\left\{ \langle Q_i(0), Q_i(1) \rangle \right\}_{i=1}^n = \arg\min_Q \sum_{j=1}^m w_j \cdot \left( \sum_{i=1}^n Q_i(c_{j,i}) - y_j \right)^2.$$

The solution of the given least-squares problem can be computed as follows:

$$\mathbf{x}^* = \langle Q_i(1) - Q_i(0) \rangle_{i=1}^n,$$

$$b^* = \sum_{i=1}^n Q_i(0).$$

Thus computing linear least-squares problems with binary weight matrices is equivalent to computing one iteration of FQI-LVF with some dataset collected in a Dec-POMDP with a decentralized policy. $\square$

**Proposition 4.** *Computing linear least-squares problems with binary matrices can be reduced to computing one iteration of FQI-LVF with some dataset collected in an MMDP with a centralized policy.*

*Proof.* Consider the following linear least-squares problem:

$$(\mathbf{x}^*, b^*) = \arg\min_{\mathbf{x}, b} \sum_{j=1}^m w_j \cdot \left( \mathbf{a}_j^\top \mathbf{x} + b - y_j \right)^2, \tag{46}$$

where $\{\mathbf{a}_j\}_{j=1}^m$ are a set of given binary vectors, $\{y_j\}_{j=1}^m$ are given labels, and $\{w_j\}_{j=1}^m$ denote the weights of each data point.

Using linear least-squares to solve one-iteration of FQI-LVF is directed. We only need to consider how to solve the above least-squares problem using an oracle that can compute one-iteration of FQI-LVF without the assumption of decentralized data collection.

We construct the dataset by executing a centralized policy in the following MMDP:

- This MMDP contains only one latent state. Each agent have two individual actions, i.e., $|\mathcal{A}| = 2$.

- All actions lead to a termination signal. The joint action $\mathbf{a}_j$ produces a reward $y_j$.

- The probability to executing joint action $\mathbf{a}_j$ is proportional to $w_j$, which is a centralized policy.

Note that all actions are states in this MMDP. Let $Q_i(0)$ and $Q_i(1)$ denote the individual value function of agent $i$ w.r.t. two actions computed by FQI-LVF, i.e.,

$$\left\{ \langle Q_i(0), Q_i(1) \rangle \right\}_{i=1}^n = \arg\min_Q \sum_{j=1}^m w_j \cdot \left( \sum_{i=1}^n Q_i(a_{j,i}) - y_j \right)^2.$$

The solution of the given least-squares problem can be computed as follows:

$$\mathbf{x}^* = \langle Q_i(1) - Q_i(0) \rangle_{i=1}^n,$$

$$b^* = \sum_{i=1}^n Q_i(0).$$

Thus computing linear least-squares problems with binary matrices is equivalent to computing one iteration of FQI-LVF with some dataset collected in an MMDP with a centralized policy. $\square$

# C Omitted Proofs in Section 5.3

## C.1 Omitted Proofs in Section 5.3.1

Note that, since we adopt the reactive function classes for analyses (see Appendix A.1), the infinite-norm of value functions is defined over a finite set.

**Proposition 1.** *The empirical Bellman operator $\mathcal{T}_D^{LVF}$ is not a $\gamma$-contraction, i.e., the following important property of the standard Bellman optimality operator $\mathcal{T}$ does not hold for $\mathcal{T}_D^{LVF}$ anymore.*

$$(\gamma\text{-contraction}) \qquad \forall Q_{tot}, Q'_{tot} \in \mathcal{Q}, \ \ \|\mathcal{T}Q_{tot} - \mathcal{T}Q'_{tot}\|_\infty \leq \gamma\|Q_{tot} - Q'_{tot}\|_\infty$$

*Proof.* Suppose the empirical Bellman operator $\mathcal{T}_D^{\text{LVF}}$ is a $\gamma$-contraction. For any Dec-ROMDPs, when using a uniform data distribution, the value function of FQI-LVF will converge [27] because of the contraction of the distance (infinity norm) between any pair of $Q$. However, one counterexample is indicated in Proposition 2, which shows that there exists Dec-ROMDPs such that, when using a uniform data distribution, the value function of FQI-LVF diverges to infinity from an arbitrary initialization $Q^{(0)}$. The assumption of $\gamma$-contraction is not hold and the empirical Bellman operator $\mathcal{T}_D^{\text{LVF}}$ is not a $\gamma$-contraction. □

**Proposition 2.** *There exist an MMDP such that, when using uniform data distribution, the value function of FQI-LVF diverges to infinity from an arbitrary initialization $Q^{(0)}$.*

*Proof.* We consider the following environment with 2 agents, 2 states $(s_1, s_2)$ and each agent $(i = 1, 2)$ has 2 actions $\mathcal{A} \equiv \{\mathcal{A}^{(1)}, \mathcal{A}^{(2)}\}$. Both agents can directly observe full state information without noises. The reward function is listed below where $r(s_j, \mathbf{a})$ denotes the reward of $(s_j, \mathbf{a})$, and $\mathbf{a} = \langle a_1, a_2 \rangle$.

$$r(s_1) = \begin{pmatrix} 0 & 0 \\ 0 & 0 \end{pmatrix} \quad r(s_2) = \begin{pmatrix} 1 & 0 \\ 0 & 0 \end{pmatrix} \tag{47}$$

Besides, the transition is deterministic.

$$T(s_1) = \begin{pmatrix} s_1 & s_1 \\ s_1 & s_1 \end{pmatrix} \quad T(s_2) = \begin{pmatrix} s_2 & s_2 \\ s_2 & s_1 \end{pmatrix} \tag{48}$$

Furthermore, $\gamma \in (\frac{4}{5}, 1)$. (In practice, $\gamma$ is usually chosen as 0.99 or 0.95.) The following proves that this example will diverge for any initialization.

To make the notations more accessible, we let the value function explicitly depend on the global state within this example. Denote $Q_i^t(s_j, a_i)$ as the decomposed Q-value of agent $i$ after $t^{\text{th}}$ value-iteration at state $s_j$ with action $a_i$. Then, the total Q-value can be described as $Q_{\text{tot}}^t(s_j, \mathbf{a}) = Q_1^t(s_j, a_1) + Q_2^t(s_j, a_2)$. For brevity, $0^{\text{th}}$ Q-value is its initialization.

First, we clarify the process of each iteration. Since the value-iteration for linear decomposed function class is solving the MSE problem in Lemma 2. $\mathbf{b}$ is target one-step TD-value w.r.t the Q-value of the last iteration. Through described in Lemma 2, the optimal solution of this MSE problem is not unique. We can ignore the term of an arbitrary vector $\mathbf{w}$ when considering the joint action-value functions because $\mathbf{w}$ does not affect the local action selection of each agent and will be eliminated in the summation operator of linear value decomposition. In addition, under uniformed sampling, we observe that $p_u(a_u) = \frac{1}{2}$ for any $\vec{a}, u$. Then, in equation 34

$$-\frac{1}{m}\frac{\mathbf{p}(\vec{a})}{p_u(a_u)} + \frac{1}{mn}\sum_{u'=0}^{n-1}\frac{\mathbf{p}(\vec{a})}{p_{u'}(a_{u'})} = 0 \tag{49}$$

Second, we denote $V_{\text{tot}}^t(s_j) = \max_{\mathbf{a}} Q_{\text{tot}}^t(s_j, \mathbf{a})$ and observe that $\forall t \geq 1, s_j$

$$Q_1^t(s_j, a_1) = \frac{1}{2}\sum_{a_2 \in \mathcal{A}}\left(r(s_j, \mathbf{a}) + \gamma V_{\text{tot}}^{t-1}(T(s_j, \mathbf{a}))\right) - \frac{1}{2}\sum_{\mathbf{a} \in \mathcal{A}}\frac{1}{4}\left(r(s_j, \mathbf{a}) + \gamma V_{\text{tot}}^{t-1}(T(s_j, \mathbf{a}))\right) \tag{50}$$

$$= Q_2^t(s_j, a_2) \tag{51}$$

The second equation holds because the transition $T$ and the reward $R$ are symmetric for both agents. Thus, we omit the subscript of local Q-values as $Q^t(s_j, a)$ when $t \geq 1$.

Third, we analyze the Q-values on state $s_1$. Clearly, its iteration is irrelevant to $s_2$. According to equation 50, $\forall a \in \mathcal{A}, t \geq 1$

$$Q^t(s_1, a) = \frac{\gamma}{2}V_{\text{tot}}^{t-1}(s_1) \tag{52}$$

$$= \frac{\gamma}{2} \max_{a_1, a_2 \in \mathcal{A}} \left( Q^{t-1}(s_1, a_1) + Q^{t-1}(s_1, a_2) \right) \tag{53}$$

Clearly, when $t \geq 1$, $Q^t \left( s_1, \mathcal{A}^{(1)} \right) = Q^t \left( s_1, \mathcal{A}^{(2)} \right)$. Therefore, we observe that $Q^t(s_1, \cdot) = \gamma^t q_1, \forall t \geq 1$ where $q_1$ is determined by the initialization $Q_{\text{tot}}^0(s_1, \mathbf{a}), \forall \mathbf{a} \in \mathcal{A}$.

Last, we consider state $s_2$. It is straightforward to observe the following recursion for $t \geq 2$ from equation 50

$$Q^t \left( s_2, \mathcal{A}^{(1)} \right) = \frac{1}{2}(1 + 2\gamma V_{\text{tot}}^{t-1}(s_2)) - \frac{1}{8}[1 + \gamma(3V_{\text{tot}}^{t-1}(s_2) + V_{\text{tot}}^{t-1}(s_1))]$$

$$= \frac{5\gamma}{8} V_{\text{tot}}^{t-1}(s_2) + \frac{3}{8} - \frac{1}{4}\gamma^t q_1$$

$$= \frac{5\gamma}{4} \max_{a \in \mathcal{A}} Q^{t-1}(s_2, a) + \frac{3}{8} - \frac{1}{4}\gamma^t q_1 \tag{54}$$

$$Q^t \left( s_2, \mathcal{A}^{(2)} \right) = \frac{1}{2}(\gamma V_{\text{tot}}^{t-1}(s_2) + \gamma V_{\text{tot}}^{t-1}(s_1)) - \frac{1}{8}[1 + \gamma(3V_{\text{tot}}^{t-1}(s_2) + V_{\text{tot}}^{t-1}(s_1))]$$

$$= \frac{\gamma}{8} V_{\text{tot}}^{t-1}(s_2) - \frac{1}{8} + \frac{3}{4}\gamma^t q_1$$

$$= \frac{\gamma}{4} \max_{a \in \mathcal{A}} Q^{t-1}(s_2, a) - \frac{1}{8} + \frac{3}{4}\gamma^t q_1 \tag{55}$$

We consider some $\delta > 0$ and $t_\delta = \left\lceil \log_\gamma \frac{\delta}{6|q_1|} \right\rceil$. Then, $t > t_\delta$

$$Q^t \left( s_2, \mathcal{A}^{(2)} \right) \geq \frac{\gamma}{4} \max_{a \in \mathcal{A}} Q^{t-1}(s_2, a) - \frac{1+\delta}{8} \geq \frac{\gamma}{4} Q^{t-1} \left( s_2, \mathcal{A}^{(2)} \right) - \frac{1+\delta}{8} \tag{56}$$

Denote $\widehat{Q}^t \left( s_2, \mathcal{A}^{(2)} \right) = \frac{\gamma}{4} \widehat{Q}^{t-1} \left( s_2, \mathcal{A}^{(2)} \right) - \frac{1+\delta}{8}, \forall t > t_\delta$ and $\widehat{Q}^{t_\delta} \left( s_2, \mathcal{A}^{(2)} \right) = Q^{t_\delta} \left( s_2, \mathcal{A}^{(2)} \right)$. Consequently, $Q^t(s_2, a_2) \geq \widehat{Q}^{t_\delta} \left( s_2, \mathcal{A}^{(2)} \right), \forall t \geq t_\delta$ by equation 56. Since $t \geq t_\delta$

$$\widehat{Q}^t \left( s_2, \mathcal{A}^{(2)} \right) = \left( \frac{\gamma}{4} \right)^{t-t_\delta} \left( Q^{t_\delta} \left( s_2, \mathcal{A}^{(2)} \right) - \frac{1+\delta}{2\gamma - 8} \right) + \frac{1+\delta}{2\gamma - 8} \tag{57}$$

Furthermore, $\gamma \in (\frac{4}{5}, 1)$. There exists some $T_\delta \geq t_\delta$ which

$$Q^{T_\delta} \left( s_2, \mathcal{A}^{(2)} \right) \geq \widehat{Q}^{T_\delta} \left( s_2, \mathcal{A}^{(2)} \right) \geq \frac{1+2\delta}{2\gamma - 8} > -\frac{1+2\delta}{6} \tag{58}$$

According to equation 54 and let $\delta < \frac{1}{11}$.

$$Q^{T_\delta+1} \left( s_2, \mathcal{A}^{(1)} \right) \geq \frac{5\gamma}{4} Q^{T_\delta} \left( s_2, \mathcal{A}^{(2)} \right) + \frac{3}{8} - \frac{1}{4}\gamma^t q_1 \tag{59}$$

$$> -\frac{5+10\delta}{24} + \frac{3}{8} - \frac{1}{24}\delta \tag{60}$$

$$> \frac{1}{8} \tag{61}$$

Similar to equation 56, we observer from equation 54 that $\forall t > T_{\delta=\frac{1}{11}} + 1$

$$Q^t \left( s_2, \mathcal{A}^{(1)} \right) \geq \frac{5\gamma}{4} Q^{t-1} \left( s_2, \mathcal{A}^{(1)} \right) + \frac{1}{4} \tag{62}$$

and

$$V_{\text{tot}}^t \left( s_2 \right) = 2Q^t \left( s_2, \mathcal{A}^{(1)} \right) \tag{63}$$

$$\geq 2 \left( \frac{5\gamma}{4} Q^{t-1} \left( s_2, \mathcal{A}^{(1)} \right) + \frac{1}{4} \right) \tag{64}$$

$$= \frac{5\gamma}{4} V_{\text{tot}}^{t-1} \left( s_2 \right) + \frac{1}{4} \tag{65}$$

Since $\frac{5\gamma}{4} > 1$ and the initial point at $T_{\delta=\frac{1}{11}} + 1$ is larger than $\frac{1}{8}$, this suggests that $V_{\text{tot}}^t \left( s_2 \right)$ will eventually diverge.

Noticing that our proof holds with respect to any $\{Q_{\text{tot}}^0(s_j, \mathbf{a}) | \forall j \in \mathcal{S}, \mathbf{a} \in \mathcal{A}\}$. Thus, value-iteration on linear decomposed function class w.r.t this MDP will diverge evnetually under any circumstances. $\qquad \square$

## C.2 Omitted Statements in Section 5.3.2

---

**Algorithm 2** On-Policy Fitted Q-Iteration with $\epsilon$-greedy Exploration

---

1: Initialize $Q^{(0)}$.
2: **for** $t = 0 \ldots T - 1$ **do**          $\triangleright$ $T$ denotes the computation budget
3:      Construct an exploratory policy $\tilde{\pi}_t$ based on $Q^{(t)}$.      $\triangleright$ i.e., $\epsilon$-greedy exploration

$$\tilde{\pi}_t(\mathbf{a}|\boldsymbol{x}) = \prod_{i=1}^{n} \left( \frac{\epsilon}{|\mathcal{A}|} + (1 - \epsilon) \mathbb{I} \left[ a_i = \arg\max_{a_i' \in \mathcal{A}} Q_i^{(t)}(x_i, a_i') \right] \right) \tag{66}$$

4:      Collect a new dataset $D_t$ by running $\tilde{\pi}_t$.
5:      Operate an on-policy Bellman operator $Q^{(t+1)} \leftarrow \mathcal{T}_\epsilon^{\mathrm{LVF}} Q^{(t)} \equiv \mathcal{T}_{D_t}^{\mathrm{LVF}} Q^{(t)}$.

---

Algorithm 2 is a variant of fitted Q-iteration which adopts an on-policy sample distribution. At line 3, an exploratory noise is integrated into the greedy policy, since the function approximator generally requires an extensive set of samples to regularize extrapolation values. Particularly, we investigate a standard exploration module called $\epsilon$-greedy, in which every agent takes a small probability to explore actions with non-maximum values. To make the underlying insights more accessible, we assume the data collection procedure at line 4 can obtain infinite samples, which makes the dataset $D_t$ become a sufficient coverage over the state-action space (see Assumption 1). This algorithmic framework serves as a foundation for discussions on local stability.

We consider an additional assumption stated as follows.

**Assumption 3** (Unique Optimal Policy). *The optimal policy $\boldsymbol{\pi}^*$ is unique.*

The intuitive motivation of this assumption is to have the optimal policy $\boldsymbol{\pi}^*$ be a potential stable solution. In situations where the optimal policy is not unique, most Q-learning algorithms will oscillate around multiple optimal policies [66], and Assumption 3 helps us to rule out these non-interesting cases. Based on this setting, the local stability of FQI-LVF can be characterized by the following lemma.

**Lemma 3.** *There exists a threshold $\delta > 0$ such that the on-policy Bellman operator $\mathcal{T}_\epsilon^{LVF}$ is closed in the following subspace $\mathcal{B} \subset \mathcal{Q}^{LVF}$, when the hyper-parameter $\epsilon$ is sufficiently small.*

$$\mathcal{B} = \left\{ Q \in \mathcal{Q}^{LVF} \ \middle| \ \boldsymbol{\pi}_Q = \boldsymbol{\pi}^*, \ \max_{\boldsymbol{x} \in \mathcal{X}} |Q_{tot}(\boldsymbol{x}, \boldsymbol{\pi}^*(\boldsymbol{x})) - V^*(\boldsymbol{x})| \leq \delta \right\}$$

*Formally, $\exists \delta > 0$, $\exists \epsilon > 0$, $\forall Q \in \mathcal{B}$, there must be $\mathcal{T}_\epsilon^{LVF} Q \in \mathcal{B}$.*

Lemma 3 indicates that once the value function $Q$ steps into the subspace $\mathcal{B}$, the induced policy $\pi_Q$ will converge to the optimal policy $\boldsymbol{\pi}^*$. By combining this local stability with Brouwer's fixed-point theorem [47], we can further verify the existence of a fixed-point solution for the on-policy Bellman operator $\mathcal{T}_\epsilon^{\mathrm{LVF}}$ (see Theorem 4).

**Theorem 4** (Formal version of Theorem 2). *Besides Lemma 3, Algorithm 2 will have a fixed point value function expressing the optimal policy if the hyper-parameter $\epsilon$ is sufficiently small.*

Theorem 4 indicates that, multi-agent Q-learning with linear value decomposition has a convergent region, where the value function induces optimal actions. Note that $\mathcal{Q}^{\mathrm{LVF}}$ is a limited function class, which even cannot guarantee to contain the one-step TD target $\mathcal{T}_D^{\mathrm{LVF}} Q$. From this perspective, on-policy data distribution becomes necessary to make the one-step TD target projected to a small set of critical observation-action pairs, which help construct the stable subspace $\mathcal{B}$ stated in Lemma 3.

## C.3 Omitted Proofs in Section 5.3.2

In this section, we only consider the data distribution generated by the optimal joint policy $\boldsymbol{\pi}^*$.

To simplify the notations, we use $\varepsilon = \frac{\epsilon}{|\mathcal{A}|}$ to reformulate the exploratory policy generated by $\epsilon$-greedy exploration as follows

$$\tilde{\pi}(\mathbf{a}|\boldsymbol{x}) = \prod_{i=1}^{n} \left( \varepsilon + (1 - \hat{\varepsilon}) \mathbb{I} \left[ a_i = \arg\max_{a_i' \in \mathcal{A}} Q_i^*(x_i, a_i') \right] \right) \tag{67}$$

where $\hat{\varepsilon} = (|\mathcal{A}| - 1)\varepsilon$.

In addition, we use $f(\boldsymbol{x}, \cdot, \cdot)$ to denote the corresponding coefficient in the closed-form updating

$$(\mathcal{T}_D^{\mathrm{LVF}} Q)_{\mathrm{tot}}(\boldsymbol{x}, \mathbf{a}) = \sum_{\mathbf{a}' \in \mathcal{A}^n} f(\boldsymbol{x}, \mathbf{a}, \mathbf{a}')(\mathcal{T} Q)_{\mathrm{tot}}(\boldsymbol{x}, \mathbf{a}') \tag{68}$$

where $(\mathcal{T}Q)_{\text{tot}} = r(s, \mathbf{a}') + \gamma \mathbb{E}[V_{\text{tot}}(\boldsymbol{x}')]$ denote the precise target values derived by Bellman optimality equation.
Formally, according to Eq. (8),

$$f(\boldsymbol{x}, \mathbf{a}, \mathbf{a}') = \left( \frac{h^{(1)}(\boldsymbol{x}, \mathbf{a}, \mathbf{a}')}{1 - \hat{\varepsilon}} + \frac{h^{(0)}(\boldsymbol{x}, \mathbf{a}, \mathbf{a}')}{\varepsilon} - (n-1) \right) (1 - \hat{\varepsilon})^{h^{\boldsymbol{\pi}^*}(\boldsymbol{x}, \mathbf{a}')} \varepsilon^{n - h^{\boldsymbol{\pi}^*}(\boldsymbol{x}, \mathbf{a}')}, \tag{69}$$

in which

$$h^{\boldsymbol{\pi}^*}(\boldsymbol{x}, \mathbf{a}) = \sum_{i=1}^{n} \mathbb{I}[a_i = \pi_i^*(x_i)] \tag{70}$$

$$h^{(1)}(\boldsymbol{x}, \mathbf{a}, \mathbf{a}') = \sum_{i=1}^{n} \mathbb{I}[a_i = \pi_i^*(x_i)] \mathbb{I}[a_i = a_i'] \tag{71}$$

$$h^{(0)}(\boldsymbol{x}, \mathbf{a}, \mathbf{a}') = \sum_{i=1}^{n} \mathbb{I}[a_i \neq \pi_i^*(x_i)] \mathbb{I}[a_i = a_i'] \tag{72}$$

As a reference indicating whether the learned value function produces the optimal policy, we denote

$$\mathcal{E}(Q) = \max_{s \in \mathcal{S}} \left[ \max_{\mathbf{a} \in (\mathcal{A}^n \setminus \{\boldsymbol{\pi}^*(\boldsymbol{x})\})} (Q_{\text{tot}}(\boldsymbol{x}, \boldsymbol{\pi}^*(\boldsymbol{x})) - Q_{\text{tot}}(\boldsymbol{x}, \mathbf{a})) \right] \tag{73}$$

Notice that $\boldsymbol{\pi}^*$ denotes the optimal policy of the given MDP, so $\mathcal{E}(Q)$ might be negative for a non-optimal or inaccurate value function $Q$.

**Lemma 4.** *Given a dataset $D$ generated by the optimal policy $\boldsymbol{\pi}^*$ with $\epsilon$-greedy exploration, for any target value function $Q$,*

$$\forall \delta > 0, \ \forall 0 < \varepsilon \le \frac{\delta}{n^2 |\mathcal{A}|^n 2^{n+1} (R_{max} + \gamma \|V_{tot}\|_\infty)}, \tag{74}$$

*we have*

$$\forall s \in \mathcal{S}, \ \left| (\mathcal{T}_D^{LVF} Q)_{tot}(\boldsymbol{x}, \boldsymbol{\pi}^*(\boldsymbol{x})) - (\mathcal{T}Q)_{tot}(\boldsymbol{x}, \boldsymbol{\pi}^*(\boldsymbol{x})) \right| \le \delta, \tag{75}$$

*where $(\mathcal{T}Q)_{tot}(\boldsymbol{x}, \mathbf{a}) = r(s, \mathbf{a}) + \gamma \mathbb{E}[V_{tot}(\boldsymbol{x}')]$ denotes the regression target generated by $Q$.*

*Proof.* $\forall \boldsymbol{x} \in \mathcal{X}$,

$$\left| (\mathcal{T}_D^{LVF} Q)_{\text{tot}}(\boldsymbol{x}, \boldsymbol{\pi}^*(\boldsymbol{x})) - (\mathcal{T}Q)_{\text{tot}}(\boldsymbol{x}, \boldsymbol{\pi}^*(\boldsymbol{x})) \right|$$

$$\le |(f(\boldsymbol{x}, \boldsymbol{\pi}^*(\boldsymbol{x}), \boldsymbol{\pi}^*(\boldsymbol{x})) - 1)(\mathcal{T}Q)_{\text{tot}}(\boldsymbol{x}, \boldsymbol{\pi}^*(\boldsymbol{x}))| + \left| \sum_{a' \in \mathcal{A}^n \setminus \{\boldsymbol{\pi}^*(\boldsymbol{x})\}} f(\boldsymbol{x}, \boldsymbol{\pi}^*(\boldsymbol{x}), \mathbf{a}')(\mathcal{T}Q)_{\text{tot}}(\boldsymbol{x}, \mathbf{a}') \right|$$

$$\le \left( |f(\boldsymbol{x}, \boldsymbol{\pi}^*(\boldsymbol{x}), \boldsymbol{\pi}^*(\boldsymbol{x})) - 1| + \sum_{a' \in \mathcal{A}^n \setminus \{\boldsymbol{\pi}^*(\boldsymbol{x})\}} |f(\boldsymbol{x}, \boldsymbol{\pi}^*(\boldsymbol{x}), \mathbf{a}')| \right) \|(\mathcal{T}Q)_{\text{tot}}\|_\infty. \tag{76}$$

In the first term, $\forall \boldsymbol{x} \in \mathcal{X}$,

$$|f(\boldsymbol{x}, \boldsymbol{\pi}^*(\boldsymbol{x}), \boldsymbol{\pi}^*(\boldsymbol{x})) - 1| = \left| \left( \frac{n}{1 - \hat{\varepsilon}} - (n-1) \right) (1 - \hat{\varepsilon})^n - 1 \right|$$

$$= \left| (n - (n-1)(1 - \hat{\varepsilon}))(1 - \hat{\varepsilon})^{n-1} - 1 \right|$$

$$= \left| (1 + (n-1)\hat{\varepsilon})(1 - \hat{\varepsilon})^{n-1} - 1 \right|$$

$$= \left| (1 + (n-1)\hat{\varepsilon}) \left( \sum_{\ell=0}^{n-1} \binom{n-1}{\ell} (-1)^\ell \hat{\varepsilon}^\ell \right) - 1 \right|$$

$$= \left| (1 + (n-1)\hat{\varepsilon}) \left( 1 - (n-1)\hat{\varepsilon} + \sum_{\ell=2}^{n-1} \binom{n-1}{\ell} (-1)^\ell \hat{\varepsilon}^\ell \right) - 1 \right|$$

$$= \left| 1 - (n-1)^2 \hat{\varepsilon}^2 + (1 + (n-1)\hat{\varepsilon}) \left( \sum_{\ell=2}^{n-1} \binom{n-1}{\ell} (-1)^\ell \hat{\varepsilon}^\ell \right) - 1 \right|$$

$$= \left| \hat{\varepsilon}^2 \left( (n-1)^2 - (1 + (n-1)\hat{\varepsilon}) \sum_{\ell=2}^{n-1} \binom{n-1}{\ell} (-1)^\ell \hat{\varepsilon}^{\ell-2} \right) \right|$$

$$\leq |\mathcal{A}|^2 \varepsilon^2 \left( n^2 + 2 \sum_{\ell=2}^{n-1} \binom{n-1}{\ell} \right)$$

$$\leq |\mathcal{A}|^2 \varepsilon^2 \left( n^2 + 2^n \right)$$

$$\leq \varepsilon^2 n^2 |\mathcal{A}|^2 2^n. \tag{77}$$

In the second term, $\forall \boldsymbol{x} \in \boldsymbol{\mathcal{X}}$,

$$\sum_{\mathbf{a}' \in \mathcal{A}^n \setminus \{\boldsymbol{\pi}^*(\boldsymbol{x})\}} |f(\boldsymbol{x}, \boldsymbol{\pi}^*(\boldsymbol{x}), \mathbf{a}')|$$

$$\leq \sum_{\mathbf{a}' \in \mathcal{A}^n \setminus \{\boldsymbol{\pi}^*(\boldsymbol{x})\}} \left| \left( \frac{h^{\boldsymbol{\pi}^*}(\boldsymbol{x}, \mathbf{a}')}{1-\hat{\varepsilon}} - (n-1) \right) (1-\hat{\varepsilon})^{h^{\boldsymbol{\pi}^*}(\boldsymbol{x}, \mathbf{a}')} \varepsilon^{n-h^{\boldsymbol{\pi}^*}(\boldsymbol{x}, \mathbf{a}')} \right|$$

$$= \sum_{\mathbf{a}' \in \mathcal{A}^n \setminus \{\boldsymbol{\pi}^*(\boldsymbol{x})\}} \left| \left( h^{\boldsymbol{\pi}^*}(\boldsymbol{x}, \mathbf{a}') - (n-1)(1-\hat{\varepsilon}) \right) (1-\hat{\varepsilon})^{h^{\boldsymbol{\pi}^*}(\boldsymbol{x}, \mathbf{a}')-1} \varepsilon^{n-h^{\boldsymbol{\pi}^*}(\boldsymbol{x}, \mathbf{a}')} \right|$$

$$\leq \sum_{\mathbf{a}' \in \mathcal{A}^n \setminus \{\boldsymbol{\pi}^*(\boldsymbol{x})\}} \left| 2n(1-\hat{\varepsilon})^{h^{\boldsymbol{\pi}^*}(\boldsymbol{x}, \mathbf{a}')-1} \varepsilon^{n-h^{\boldsymbol{\pi}^*}(\boldsymbol{x}, \mathbf{a}')} \right|$$

$$\leq \sum_{\mathbf{a}' \in \mathcal{A}^n \setminus \{\boldsymbol{\pi}^*(\boldsymbol{x})\}} 2n\varepsilon$$

$$\leq 2n\varepsilon |\mathcal{A}|^n. \tag{78}$$

Thus $\forall \boldsymbol{x} \in \boldsymbol{\mathcal{X}}$,

$$\left| (\mathcal{T}_D^{\text{LVF}} Q)_{\text{tot}}(\boldsymbol{x}, \boldsymbol{\pi}^*(\boldsymbol{x})) - (\mathcal{T} Q)_{\text{tot}}(\boldsymbol{x}, \boldsymbol{\pi}^*(\boldsymbol{x})) \right|$$

$$\leq \left( |f(\boldsymbol{x}, \boldsymbol{\pi}^*(\boldsymbol{x}), \boldsymbol{\pi}^*(\boldsymbol{x})) - 1| + \sum_{\mathbf{a}' \in \mathcal{A}^n \setminus \{\boldsymbol{\pi}^*(\boldsymbol{x})\}} |f(\boldsymbol{x}, \boldsymbol{\pi}^*(\boldsymbol{x}), \mathbf{a}')| \right) \|(\mathcal{T} Q)_{\text{tot}}\|_\infty$$

$$\leq (\varepsilon^2 n^2 |\mathcal{A}|^2 2^n + 2n\varepsilon |\mathcal{A}|^n) \|(\mathcal{T} Q)_{\text{tot}}\|_\infty$$

$$\leq \varepsilon n^2 |\mathcal{A}|^n 2^{n+1} \|(\mathcal{T} Q)_{\text{tot}}\|_\infty$$

$$\leq \varepsilon n^2 |\mathcal{A}|^n 2^{n+1} (R_{\max} + \gamma \|V_{\text{tot}}\|_\infty)$$

$$\leq \delta. \tag{79}$$

$\square$

**Lemma 5.** *Given a dataset $D$ generated by the optimal policy $\boldsymbol{\pi}^*$ with $\epsilon$-greedy exploration, for any target value function $Q$,*

$$\forall 0 < \varepsilon \leq \frac{(1-\gamma)\mathcal{E}(Q^*)}{\gamma n^3 |\mathcal{A}|^n 2^{n+4}(R_{max}/(1-\gamma) + \gamma \|V_{tot}^{\boldsymbol{\pi}^*} - V^*\|_\infty)}, \tag{80}$$

*we have*

$$\forall \boldsymbol{x} \in \boldsymbol{\mathcal{X}}, \ \left| (\mathcal{T}_D^{LVF} Q)_{tot}(\boldsymbol{x}, \boldsymbol{\pi}^*(\boldsymbol{x})) - V^*(\boldsymbol{x}) \right| \leq \gamma \|V_{tot}^{\boldsymbol{\pi}^*} - V^*\|_\infty + \frac{1-\gamma}{8n\gamma} \mathcal{E}(Q^*), \tag{81}$$

*where $V_{tot}^{\boldsymbol{\pi}^*}(\boldsymbol{x}) = Q_{tot}(\boldsymbol{x}, \boldsymbol{\pi}^*(\boldsymbol{x}))$.*

*Proof.* $\forall \boldsymbol{x} \in \boldsymbol{\mathcal{X}}$,

$$\left| (\mathcal{T}_D^{\text{LVF}} Q)_{\text{tot}}(\boldsymbol{x}, \boldsymbol{\pi}^*(\boldsymbol{x})) - V^*(\boldsymbol{x}) \right|$$

$$\leq \left| (\mathcal{T}_D^{\text{LVF}} Q)_{\text{tot}}(\boldsymbol{x}, \boldsymbol{\pi}^*(\boldsymbol{x})) - (\mathcal{T} Q)_{\text{tot}}(\boldsymbol{x}, \boldsymbol{\pi}^*(\boldsymbol{x})) \right| + \left| (\mathcal{T} Q)_{\text{tot}}(\boldsymbol{x}, \boldsymbol{\pi}^*(\boldsymbol{x})) - V^*(\boldsymbol{x}) \right|$$

$$= \left| (\mathcal{T}_D^{\text{LVF}} Q)_{\text{tot}}(\boldsymbol{x}, \boldsymbol{\pi}^*(\boldsymbol{x})) - (\mathcal{T} Q)_{\text{tot}}(\boldsymbol{x}, \boldsymbol{\pi}^*(\boldsymbol{x})) \right| + \left| (\mathcal{T} Q)_{\text{tot}}(\boldsymbol{x}, \boldsymbol{\pi}^*(\boldsymbol{x})) - Q^*(\boldsymbol{x}, \boldsymbol{\pi}^*(\boldsymbol{x})) \right|$$

$$= \left| (\mathcal{T}_D^{\text{LVF}} Q)_{\text{tot}}(\boldsymbol{x}, \boldsymbol{\pi}^*(\boldsymbol{x})) - (\mathcal{T} Q)_{\text{tot}}(\boldsymbol{x}, \boldsymbol{\pi}^*(\boldsymbol{x})) \right| + \left| (\mathcal{T} Q)_{\text{tot}}(\boldsymbol{x}, \boldsymbol{\pi}^*(\boldsymbol{x})) - (\mathcal{T} Q^*)(\boldsymbol{x}, \boldsymbol{\pi}^*(\boldsymbol{x})) \right|$$

$$\leq \left|(\mathcal{T}_D^{\text{LVF}}Q)_{\text{tot}}(\boldsymbol{x}, \boldsymbol{\pi}^*(\boldsymbol{x})) - (\mathcal{T}Q)_{\text{tot}}(\boldsymbol{x}, \boldsymbol{\pi}^*(\boldsymbol{x}))\right| + \gamma |V_{\text{tot}}(\boldsymbol{x}') - V^*(\boldsymbol{x}')|$$

$$\leq \left|(\mathcal{T}_D^{\text{LVF}}Q)_{\text{tot}}(\boldsymbol{x}, \boldsymbol{\pi}^*(\boldsymbol{x})) - (\mathcal{T}Q)_{\text{tot}}(\boldsymbol{x}, \boldsymbol{\pi}^*(\boldsymbol{x}))\right| + \gamma |Q_{\text{tot}}(\boldsymbol{x}', \boldsymbol{\pi}^*(\boldsymbol{x}')) - V^*(\boldsymbol{x}')|$$

$$\leq \left|(\mathcal{T}_D^{\text{LVF}}Q)_{\text{tot}}(\boldsymbol{x}, \boldsymbol{\pi}^*(\boldsymbol{x})) - (\mathcal{T}Q)_{\text{tot}}(\boldsymbol{x}, \boldsymbol{\pi}^*(\boldsymbol{x}))\right| + \gamma \|V_{\text{tot}}^{\boldsymbol{\pi}^*} - V^*\|_\infty \tag{82}$$

Let $\delta = \frac{1-\gamma}{8n\gamma}\mathcal{E}(Q^*)$. According to Lemma 4, with the condition

$$0 < \varepsilon \leq \frac{\delta}{n^2|\mathcal{A}|^n 2^{n+1}(R_{\max} + \gamma\|V_{\text{tot}}\|_\infty)} = \frac{(1-\gamma)\mathcal{E}(Q^*)/(8n\gamma)}{n^2|\mathcal{A}|^n 2^{n+1}(R_{\max} + \gamma\|V_{\text{tot}}\|_\infty)}, \tag{83}$$

we have

$$\left|(\mathcal{T}_D^{\text{LVF}}Q)_{\text{tot}}(\boldsymbol{x}, \boldsymbol{\pi}^*(\boldsymbol{x})) - (\mathcal{T}Q)_{\text{tot}}(\boldsymbol{x}, \boldsymbol{\pi}^*(\boldsymbol{x}))\right| \leq \delta = \frac{1-\gamma}{8n\gamma}\mathcal{E}(Q^*). \tag{84}$$

Notice that

$$\|V_{\text{tot}}\|_\infty \leq \|V^*\|_\infty + \|V_{\text{tot}} - V^*\|_\infty \tag{85}$$

$$\leq \frac{R_{\max}}{1-\gamma} + \|V_{\text{tot}}^{\boldsymbol{\pi}^*} - V^*\|_\infty. \tag{86}$$

The overall statement is

$$\forall 0 < \varepsilon \leq \frac{(1-\gamma)\mathcal{E}(Q^*)}{\gamma n^3|\mathcal{A}|^n 2^{n+4}(R_{\max}/(1-\gamma) + \gamma\|V_{\text{tot}}^{\boldsymbol{\pi}^*} - V^*\|_\infty)} \leq \frac{(1-\gamma)\mathcal{E}(Q^*)/(8n\gamma)}{n^2|\mathcal{A}|^n 2^{n+1}(R_{\max} + \gamma\|V_{\text{tot}}\|_\infty)} \tag{87}$$

we have $\forall \boldsymbol{x} \in \boldsymbol{\mathcal{X}}$,

$$\left|(\mathcal{T}_D^{\text{LVF}}Q)_{\text{tot}}(\boldsymbol{x}, \boldsymbol{\pi}^*(\boldsymbol{x})) - V^*(\boldsymbol{x})\right|$$

$$\leq \left|(\mathcal{T}_D^{\text{LVF}}Q)_{\text{tot}}(\boldsymbol{x}, \boldsymbol{\pi}^*(\boldsymbol{x})) - (\mathcal{T}Q)_{\text{tot}}(\boldsymbol{x}, \boldsymbol{\pi}^*(\boldsymbol{x}))\right| + \gamma \|V_{\text{tot}}^{\boldsymbol{\pi}^*} - V^*\|_\infty$$

$$\leq \gamma \|V_{\text{tot}}^{\boldsymbol{\pi}^*} - V^*\|_\infty + \frac{1-\gamma}{8n\gamma}\mathcal{E}(Q^*). \tag{88}$$

$\square$

**Lemma 6.** *For any value function Q, the corresponding sub-optimality gap satisfies*

$$\mathcal{E}(\mathcal{T}Q) \geq \mathcal{E}(Q^*) - 2\gamma\|V_{tot} - V^*\|_\infty \tag{89}$$

*Proof.* With a slight abuse of notation, let $\boldsymbol{x}_1$ and $\boldsymbol{x}_2$ denote the observations at the next timestep while taking actions $\boldsymbol{\pi}^*(\boldsymbol{x})$ and $\mathbf{a}$ upon the current state, respectively. According to the definition,

$$\mathcal{E}(\mathcal{T}Q) = \max_{(\boldsymbol{x},\mathbf{a}) \in \boldsymbol{\mathcal{X}} \times (\mathcal{A}^n \setminus \{\boldsymbol{\pi}^*(\boldsymbol{x})\})} ((\mathcal{T}Q)_{\text{tot}}(\boldsymbol{x}, \boldsymbol{\pi}^*(\boldsymbol{x})) - (\mathcal{T}Q)_{\text{tot}}(\boldsymbol{x}, \mathbf{a}))$$

$$\geq \max_{(\boldsymbol{x},\mathbf{a}) \in \boldsymbol{\mathcal{X}} \times (\mathcal{A}^n \setminus \{\boldsymbol{\pi}^*(\boldsymbol{x})\})} ((\mathcal{T}Q^*)(\boldsymbol{x}, \boldsymbol{\pi}^*(\boldsymbol{x})) - (\mathcal{T}Q^*)(\boldsymbol{x}, \mathbf{a}) - \gamma\mathbb{E}\left[|V_{\text{tot}}(\boldsymbol{x}_1) - V^*(\boldsymbol{x}_1)| + |V_{\text{tot}}(\boldsymbol{x}_2) - V^*(\boldsymbol{x}_2)|\right])$$

$$\geq \max_{(\boldsymbol{x},\mathbf{a}) \in \boldsymbol{\mathcal{X}} \times (\mathcal{A}^n \setminus \{\boldsymbol{\pi}^*(\boldsymbol{x})\})} ((\mathcal{T}Q^*)(\boldsymbol{x}, \boldsymbol{\pi}^*(\boldsymbol{x})) - (\mathcal{T}Q^*)(\boldsymbol{x}, \mathbf{a}) - 2\gamma\|V_{\text{tot}} - V^*\|_\infty)$$

$$= \max_{(\boldsymbol{x},\mathbf{a}) \in \boldsymbol{\mathcal{X}} \times (\mathcal{A}^n \setminus \{\boldsymbol{\pi}^*(\boldsymbol{x})\})} (Q^*(\boldsymbol{x}, \boldsymbol{\pi}^*(\boldsymbol{x})) - Q^*(\boldsymbol{x}, \mathbf{a}) - 2\gamma\|V_{\text{tot}} - V^*\|_\infty)$$

$$= \mathcal{E}(Q^*) - 2\gamma\|V_{\text{tot}} - V^*\|_\infty \tag{90}$$

$\square$

**Lemma 7.** *Given a dataset D generated by the optimal policy $\boldsymbol{\pi}^*$ with $\epsilon$-greedy exploration, for any target value function Q,*

$$\forall \delta > 0, \ \forall 0 < \varepsilon \leq \frac{\delta}{n^2|\mathcal{A}|^n 2^n (R_{max}/(1-\gamma) + \gamma\|V_{tot} - V^*\|_\infty)}, \tag{91}$$

*we have $\forall \boldsymbol{x} \in \boldsymbol{\mathcal{X}}$, $\forall \mathbf{a} \in \mathcal{A}^n \setminus \{\boldsymbol{\pi}^*(\boldsymbol{x})\}$,*

$$(\mathcal{T}_D^{LVF}Q)_{tot}(\boldsymbol{x}, \mathbf{a}) \leq (\mathcal{T}Q)_{tot}(\boldsymbol{x}, \boldsymbol{\pi}^*(\boldsymbol{x})) - \mathcal{E}(Q^*) + 2n\gamma\|V_{tot} - V^*\|_\infty + \delta \tag{92}$$

*where $(\mathcal{T}Q)_{tot}(\boldsymbol{x}, \mathbf{a}) = r(s, \mathbf{a}) + \gamma\mathbb{E}[V_{tot}(\boldsymbol{x}')]$ denotes the regression target generated by Q.*

*Proof.* $\forall \boldsymbol{x} \in \mathcal{X}, \forall \mathbf{a} \in \mathcal{A}^n \setminus \{\boldsymbol{\pi}^*(\boldsymbol{x})\}$,

$$
\begin{aligned}
(\mathcal{T}_D^{\text{LVF}} Q)_{\text{tot}}(\boldsymbol{x}, \mathbf{a}) &= \sum_{\mathbf{a}' \in \mathcal{A}^n} f(\boldsymbol{x}, \mathbf{a}, \mathbf{a}')(\mathcal{T}Q)_{\text{tot}}(\boldsymbol{x}, \mathbf{a}') \\
&= f(\boldsymbol{x}, \mathbf{a}, \boldsymbol{\pi}^*(\boldsymbol{x}))(\mathcal{T}Q)_{\text{tot}}(\boldsymbol{x}, \boldsymbol{\pi}^*(\boldsymbol{x})) \\
&\quad + \sum_{\mathbf{a}' \in \mathcal{A}^n : h^{\boldsymbol{\pi}^*}(\boldsymbol{x}, \mathbf{a}') = n-1} f(\boldsymbol{x}, \mathbf{a}, \mathbf{a}')(\mathcal{T}Q)_{\text{tot}}(\boldsymbol{x}, \mathbf{a}') \\
&\quad + \sum_{\mathbf{a}' \in \mathcal{A}^n : h^{\boldsymbol{\pi}^*}(\boldsymbol{x}, \mathbf{a}') < n-1} f(\boldsymbol{x}, \mathbf{a}, \mathbf{a}')(\mathcal{T}Q)_{\text{tot}}(\boldsymbol{x}, \mathbf{a}')
\end{aligned} \tag{93}
$$

In the first term,

$$
\begin{aligned}
&f(\boldsymbol{x}, \mathbf{a}, \boldsymbol{\pi}^*(\boldsymbol{x}))(\mathcal{T}Q)_{\text{tot}}(\boldsymbol{x}, \boldsymbol{\pi}^*(\boldsymbol{x})) \\
&= \left( \frac{h^{\boldsymbol{\pi}^*}(\boldsymbol{x}, \mathbf{a})}{1 - \hat{\varepsilon}} - (n-1) \right) (1 - \hat{\varepsilon})^n (\mathcal{T}Q)_{\text{tot}}(\boldsymbol{x}, \boldsymbol{\pi}^*(\boldsymbol{x})) \\
&= \left( h^{\boldsymbol{\pi}^*}(\boldsymbol{x}, \mathbf{a}) - (n-1)(1 - \hat{\varepsilon}) \right) (1 - \hat{\varepsilon})^{n-1} (\mathcal{T}Q)_{\text{tot}}(\boldsymbol{x}, \boldsymbol{\pi}^*(\boldsymbol{x})) \\
&= \left( h^{\boldsymbol{\pi}^*}(\boldsymbol{x}, \mathbf{a}) - (n-1) + (n-1)(|\mathcal{A}| - 1)\varepsilon \right) (1 - \hat{\varepsilon})^{n-1} (\mathcal{T}Q)_{\text{tot}}(\boldsymbol{x}, \boldsymbol{\pi}^*(\boldsymbol{x})) \\
&\leq \left( h^{\boldsymbol{\pi}^*}(\boldsymbol{x}, \mathbf{a}) - (n-1) \right) (1 - \hat{\varepsilon})^{n-1} (\mathcal{T}Q)_{\text{tot}}(\boldsymbol{x}, \boldsymbol{\pi}^*(\boldsymbol{x})) + \varepsilon n |\mathcal{A}| \|(\mathcal{T}Q)_{\text{tot}}\|_\infty \\
&= \left( h^{\boldsymbol{\pi}^*}(\boldsymbol{x}, \mathbf{a}) - (n-1) \right) (1 + (1 - \hat{\varepsilon})^{n-1} - 1)(\mathcal{T}Q)_{\text{tot}}(\boldsymbol{x}, \boldsymbol{\pi}^*(\boldsymbol{x})) + \varepsilon n |\mathcal{A}| \|(\mathcal{T}Q)_{\text{tot}}\|_\infty \\
&\leq \left( h^{\boldsymbol{\pi}^*}(\boldsymbol{x}, \mathbf{a}) - (n-1) \right) (\mathcal{T}Q)_{\text{tot}}(\boldsymbol{x}, \boldsymbol{\pi}^*(\boldsymbol{x})) + \left| h^{\boldsymbol{\pi}^*}(\boldsymbol{x}, \mathbf{a}) - (n-1) \right| |(1 - \hat{\varepsilon})^{n-1} - 1| \|(\mathcal{T}Q)_{\text{tot}}\|_\infty + \varepsilon n |\mathcal{A}| \|(\mathcal{T}Q)_{\text{tot}}\|_\infty \\
&\leq \left( h^{\boldsymbol{\pi}^*}(\boldsymbol{x}, \mathbf{a}) - (n-1) \right) (\mathcal{T}Q)_{\text{tot}}(\boldsymbol{x}, \boldsymbol{\pi}^*(\boldsymbol{x})) + 2n \left| \sum_{\ell=1}^{n-1} \binom{n-1}{\ell} (-1)^\ell \hat{\varepsilon}^\ell \right| \|(\mathcal{T}Q)_{\text{tot}}\|_\infty + \varepsilon n |\mathcal{A}| \|(\mathcal{T}Q)_{\text{tot}}\|_\infty \\
&\leq \left( h^{\boldsymbol{\pi}^*}(\boldsymbol{x}, \mathbf{a}) - (n-1) \right) (\mathcal{T}Q)_{\text{tot}}(\boldsymbol{x}, \boldsymbol{\pi}^*(\boldsymbol{x})) + 2n \hat{\varepsilon} \left( \sum_{\ell=1}^{n-1} \binom{n-1}{\ell} \right) \|(\mathcal{T}Q)_{\text{tot}}\|_\infty + \varepsilon n |\mathcal{A}| \|(\mathcal{T}Q)_{\text{tot}}\|_\infty \\
&\leq \left( h^{\boldsymbol{\pi}^*}(\boldsymbol{x}, \mathbf{a}) - (n-1) \right) (\mathcal{T}Q)_{\text{tot}}(\boldsymbol{x}, \boldsymbol{\pi}^*(\boldsymbol{x})) + \hat{\varepsilon} n 2^n \|(\mathcal{T}Q)_{\text{tot}}\|_\infty + \varepsilon n |\mathcal{A}| \|(\mathcal{T}Q)_{\text{tot}}\|_\infty \\
&\leq \left( h^{\boldsymbol{\pi}^*}(\boldsymbol{x}, \mathbf{a}) - (n-1) \right) (\mathcal{T}Q)_{\text{tot}}(\boldsymbol{x}, \boldsymbol{\pi}^*(\boldsymbol{x})) + \varepsilon n 2^n |\mathcal{A}| \|(\mathcal{T}Q)_{\text{tot}}\|_\infty + \varepsilon n |\mathcal{A}| \|(\mathcal{T}Q)_{\text{tot}}\|_\infty
\end{aligned} \tag{94}
$$

In the second term,

$$
\begin{aligned}
&\sum_{\mathbf{a}' \in \mathcal{A}^n : h^{\boldsymbol{\pi}^*}(\boldsymbol{x}, \mathbf{a}') = n-1} f(\boldsymbol{x}, \mathbf{a}, \mathbf{a}')(\mathcal{T}Q)_{\text{tot}}(\boldsymbol{x}, \mathbf{a}') \\
&= \sum_{\mathbf{a}' \in \mathcal{A}^n : h^{\boldsymbol{\pi}^*}(\boldsymbol{x}, \mathbf{a}') = n-1} \left( \frac{h^{(1)}(\boldsymbol{x}, \mathbf{a}, \mathbf{a}')}{1 - \hat{\varepsilon}} + \frac{h^{(0)}(\boldsymbol{x}, \mathbf{a}, \mathbf{a}')}{\varepsilon} - (n-1) \right) (1 - \hat{\varepsilon})^{n-1} \varepsilon (\mathcal{T}Q)_{\text{tot}}(\boldsymbol{x}, \mathbf{a}') \\
&= \sum_{\mathbf{a}' \in \mathcal{A}^n : h^{\boldsymbol{\pi}^*}(\boldsymbol{x}, \mathbf{a}') = n-1} \left( h^{(0)}(\boldsymbol{x}, \mathbf{a}, \mathbf{a}')(1 - \hat{\varepsilon})^{n-1}(\mathcal{T}Q)_{\text{tot}}(\boldsymbol{x}, \mathbf{a}') + \left( \frac{h^{(1)}(\boldsymbol{x}, \mathbf{a}, \mathbf{a}')}{1 - \hat{\varepsilon}} - (n-1) \right) (1 - \hat{\varepsilon})^{n-1} \varepsilon (\mathcal{T}Q)_{\text{tot}}(\boldsymbol{x}, \mathbf{a}') \right) \\
&\leq \sum_{\mathbf{a}' \in \mathcal{A}^n : h^{\boldsymbol{\pi}^*}(\boldsymbol{x}, \mathbf{a}') = n-1} \left( h^{(0)}(\boldsymbol{x}, \mathbf{a}, \mathbf{a}')(1 - \hat{\varepsilon})^{n-1}(\mathcal{T}Q)_{\text{tot}}(\boldsymbol{x}, \mathbf{a}') + \left| \frac{h^{(1)}(\boldsymbol{x}, \mathbf{a}, \mathbf{a}')}{1 - \hat{\varepsilon}} - (n-1) \right| (1 - \hat{\varepsilon})^{n-1} \varepsilon \|(\mathcal{T}Q)_{\text{tot}}\|_\infty \right) \\
&\leq \sum_{\mathbf{a}' \in \mathcal{A}^n : h^{\boldsymbol{\pi}^*}(\boldsymbol{x}, \mathbf{a}') = n-1} \left( h^{(0)}(\boldsymbol{x}, \mathbf{a}, \mathbf{a}')(1 - \hat{\varepsilon})^{n-1}(\mathcal{T}Q)_{\text{tot}}(\boldsymbol{x}, \mathbf{a}') + 2n\varepsilon \|(\mathcal{T}Q)_{\text{tot}}\|_\infty \right) \\
&= \sum_{\mathbf{a}' \in \mathcal{A}^n : h^{\boldsymbol{\pi}^*}(\boldsymbol{x}, \mathbf{a}') = n-1} \left( h^{(0)}(\boldsymbol{x}, \mathbf{a}, \mathbf{a}') \left( \sum_{\ell=0}^{n-1} \binom{n-1}{\ell} (-1)^\ell \hat{\varepsilon}^\ell \right) (\mathcal{T}Q)_{\text{tot}}(\boldsymbol{x}, \mathbf{a}') + 2n\varepsilon \|(\mathcal{T}Q)_{\text{tot}}\|_\infty \right) \\
&= \sum_{\mathbf{a}' \in \mathcal{A}^n : h^{\boldsymbol{\pi}^*}(\boldsymbol{x}, \mathbf{a}') = n-1} \left( h^{(0)}(\boldsymbol{x}, \mathbf{a}, \mathbf{a}') \left( 1 + \sum_{\ell=1}^{n-1} \binom{n-1}{\ell} (-1)^\ell \hat{\varepsilon}^\ell \right) (\mathcal{T}Q)_{\text{tot}}(\boldsymbol{x}, \mathbf{a}') + 2n\varepsilon \|(\mathcal{T}Q)_{\text{tot}}\|_\infty \right)
\end{aligned}
$$

$$\leq \sum_{\mathbf{a}'\in\mathcal{A}^n:h^{\boldsymbol{\pi}^*}(\boldsymbol{x},\mathbf{a}')=n-1} \left(h^{(0)}(\boldsymbol{x},\mathbf{a},\mathbf{a}')(\mathcal{T}Q)_{\mathrm{tot}}(\boldsymbol{x},\mathbf{a}') + \left|\sum_{\ell=1}^{n-1}\binom{n-1}{\ell}(-1)^\ell\hat{\varepsilon}^\ell\right|\|(\mathcal{T}Q)_{\mathrm{tot}}\|_\infty + 2n\varepsilon\|(\mathcal{T}Q)_{\mathrm{tot}}\|_\infty\right)$$

$$= \sum_{\mathbf{a}'\in\mathcal{A}^n:h^{\boldsymbol{\pi}^*}(\boldsymbol{x},\mathbf{a}')=n-1} \left(h^{(0)}(\boldsymbol{x},\mathbf{a},\mathbf{a}')(\mathcal{T}Q)_{\mathrm{tot}}(\boldsymbol{x},\mathbf{a}') + \hat{\varepsilon}\left|\sum_{\ell=1}^{n-1}\binom{n-1}{\ell}(-1)^\ell\hat{\varepsilon}^{\ell-1}\right|\|(\mathcal{T}Q)_{\mathrm{tot}}\|_\infty + 2n\varepsilon\|(\mathcal{T}Q)_{\mathrm{tot}}\|_\infty\right)$$

$$\leq \sum_{\mathbf{a}'\in\mathcal{A}^n:h^{\boldsymbol{\pi}^*}(\boldsymbol{x},\mathbf{a}')=n-1} \left(h^{(0)}(\boldsymbol{x},\mathbf{a},\mathbf{a}')(\mathcal{T}Q)_{\mathrm{tot}}(\boldsymbol{x},\mathbf{a}') + \hat{\varepsilon}\left(\sum_{\ell=1}^{n-1}\binom{n-1}{\ell}\right)\|(\mathcal{T}Q)_{\mathrm{tot}}\|_\infty + 2n\varepsilon\|(\mathcal{T}Q)_{\mathrm{tot}}\|_\infty\right)$$

$$\leq \sum_{\mathbf{a}'\in\mathcal{A}^n:h^{\boldsymbol{\pi}^*}(\boldsymbol{x},\mathbf{a}')=n-1} \left(h^{(0)}(\boldsymbol{x},\mathbf{a},\mathbf{a}')(\mathcal{T}Q)_{\mathrm{tot}}(\boldsymbol{x},\mathbf{a}') + \varepsilon|\mathcal{A}|2^{n-1}\|(\mathcal{T}Q)_{\mathrm{tot}}\|_\infty + 2n\varepsilon\|(\mathcal{T}Q)_{\mathrm{tot}}\|_\infty\right)$$

$$= \left(\sum_{\mathbf{a}'\in\mathcal{A}^n:h^{\boldsymbol{\pi}^*}(\boldsymbol{x},\mathbf{a}')=n-1} h^{(0)}(\boldsymbol{x},\mathbf{a},\mathbf{a}')(\mathcal{T}Q)_{\mathrm{tot}}(\boldsymbol{x},\mathbf{a}')\right) + \varepsilon n|\mathcal{A}|2^{n-1}\|(\mathcal{T}Q)_{\mathrm{tot}}\|_\infty + 2n^2\varepsilon\|(\mathcal{T}Q)_{\mathrm{tot}}\|_\infty$$

$$\leq \left(\sum_{\mathbf{a}'\in\mathcal{A}^n:h^{\boldsymbol{\pi}^*}(\boldsymbol{x},\mathbf{a}')=n-1} h^{(0)}(\boldsymbol{x},\mathbf{a},\mathbf{a}')(\mathcal{T}Q)_{\mathrm{tot}}(\boldsymbol{x},\mathbf{a}')\right) + \varepsilon n^2|\mathcal{A}|2^n\|(\mathcal{T}Q)_{\mathrm{tot}}\|_\infty \tag{95}$$

In the third term,
$$\sum_{\mathbf{a}'\in\mathcal{A}^n:h^{\boldsymbol{\pi}^*}(\boldsymbol{x},\mathbf{a}')<n-1} f(\boldsymbol{x},\mathbf{a},\mathbf{a}')(\mathcal{T}Q)_{\mathrm{tot}}(\boldsymbol{x},\mathbf{a}')$$

$$\leq \sum_{a'\in\mathcal{A}^n:h^{\boldsymbol{\pi}^*}(\boldsymbol{x},\mathbf{a}')<n-1} |f(\boldsymbol{x},\mathbf{a},\mathbf{a}')(\mathcal{T}Q)_{\mathrm{tot}}(\boldsymbol{x},\mathbf{a}')|$$

$$= \sum_{\mathbf{a}'\in\mathcal{A}^n:h^{\boldsymbol{\pi}^*}(\boldsymbol{x},\mathbf{a}')<n-1} \left|\frac{h^{(1)}(\boldsymbol{x},\mathbf{a},\mathbf{a}')}{1-\hat{\varepsilon}} + \frac{h^{(0)}(\boldsymbol{x},\mathbf{a},\mathbf{a}')}{\varepsilon} - (n-1)\right|(1-\hat{\varepsilon})^{h^{\boldsymbol{\pi}^*}(\boldsymbol{x},\mathbf{a}')}\varepsilon^{n-h^{\boldsymbol{\pi}^*}(\boldsymbol{x},\mathbf{a}')}|(\mathcal{T}Q)_{\mathrm{tot}}(\boldsymbol{x},\mathbf{a}')|$$

$$\leq \sum_{\mathbf{a}'\in\mathcal{A}^n:h^{\boldsymbol{\pi}^*}(\boldsymbol{x},\mathbf{a}')<n-1} \left|\frac{h^{(1)}(\boldsymbol{x},\mathbf{a},\mathbf{a}')}{1-\hat{\varepsilon}} + \frac{h^{(0)}(\boldsymbol{x},\mathbf{a},\mathbf{a}')}{\varepsilon} + (n-1)\right|(1-\hat{\varepsilon})^{h^{\boldsymbol{\pi}^*}(\boldsymbol{x},\mathbf{a}')}\varepsilon^{n-h^{\boldsymbol{\pi}^*}(\boldsymbol{x},\mathbf{a}')}|(\mathcal{T}Q)_{\mathrm{tot}}(\boldsymbol{x},\mathbf{a}')|$$

$$\leq \sum_{\mathbf{a}'\in\mathcal{A}^n:h^{\boldsymbol{\pi}^*}(\boldsymbol{x},\mathbf{a}')<n-1} n\left(1+\frac{1}{1-\hat{\varepsilon}}+\frac{1}{\varepsilon}\right)(1-\hat{\varepsilon})^{h^{\boldsymbol{\pi}^*}(\boldsymbol{x},\mathbf{a}')}\varepsilon^{n-h^{\boldsymbol{\pi}^*}(\boldsymbol{x},\mathbf{a}')}|(\mathcal{T}Q)_{\mathrm{tot}}(\boldsymbol{x},\mathbf{a}')|$$

$$\leq \sum_{\mathbf{a}'\in\mathcal{A}^n:h^{\boldsymbol{\pi}^*}(\boldsymbol{x},\mathbf{a}')<n-1} n\left(1+\frac{2}{\varepsilon}\right)(1-\hat{\varepsilon})^{h^{\boldsymbol{\pi}^*}(\boldsymbol{x},\mathbf{a}')}\varepsilon^{n-h^{\boldsymbol{\pi}^*}(\boldsymbol{x},\mathbf{a}')}|(\mathcal{T}Q)_{\mathrm{tot}}(\boldsymbol{x},\mathbf{a}')|$$

$$\leq \sum_{\mathbf{a}'\in\mathcal{A}^n:h^{\boldsymbol{\pi}^*}(\boldsymbol{x},\mathbf{a}')<n-1} 3n\varepsilon^{n-h^{\boldsymbol{\pi}^*}(\boldsymbol{x},\mathbf{a}')-1}|(\mathcal{T}Q)_{\mathrm{tot}}(\boldsymbol{x},\mathbf{a}')|$$

$$\leq \sum_{\mathbf{a}'\in\mathcal{A}^n:h^{\boldsymbol{\pi}^*}(\boldsymbol{x},\mathbf{a}')<n-1} 3n\varepsilon\|(\mathcal{T}Q)_{\mathrm{tot}}\|_\infty$$

$$\leq 3n\varepsilon|\mathcal{A}|^n\|(\mathcal{T}Q)_{\mathrm{tot}}\|_\infty \tag{96}$$

Combining the above terms, we can get
$$(\mathcal{T}_D^{\mathrm{LVF}}Q)_{\mathrm{tot}}(\boldsymbol{x},\mathbf{a})$$

$$= f(\boldsymbol{x},\mathbf{a},\boldsymbol{\pi}^*(\boldsymbol{x}))(\mathcal{T}Q)_{\mathrm{tot}}(\boldsymbol{x},\boldsymbol{\pi}^*(\boldsymbol{x})) + \sum_{\mathbf{a}'\in\mathcal{A}^n:h^{\boldsymbol{\pi}^*}(\boldsymbol{x},\mathbf{a}')=n-1} f(\boldsymbol{x},\mathbf{a},\mathbf{a}')(\mathcal{T}Q)_{\mathrm{tot}}(\boldsymbol{x},\mathbf{a}')$$

$$+ \sum_{\mathbf{a}'\in\mathcal{A}^n:h^{\boldsymbol{\pi}^*}(\boldsymbol{x},\mathbf{a}')<n-1} f(\boldsymbol{x},\mathbf{a},\mathbf{a}')(\mathcal{T}Q)_{\mathrm{tot}}(\boldsymbol{x},\mathbf{a}')$$

$$\leq \left(h^{\boldsymbol{\pi}^*}(\boldsymbol{x},\mathbf{a})-(n-1)\right)(\mathcal{T}Q)_{\mathrm{tot}}(\boldsymbol{x},\boldsymbol{\pi}^*(\boldsymbol{x})) + \varepsilon n2^n|\mathcal{A}|\|(\mathcal{T}Q)_{\mathrm{tot}}\|_\infty + \varepsilon n|\mathcal{A}|\|(\mathcal{T}Q)_{\mathrm{tot}}\|_\infty$$

$$+ \left(\sum_{\mathbf{a}'\in\mathcal{A}^n:h^{\boldsymbol{\pi}^*}(\boldsymbol{x},\mathbf{a}')=n-1} h^{(0)}(\boldsymbol{x},\mathbf{a},\mathbf{a}')(\mathcal{T}Q)_{\mathrm{tot}}(\boldsymbol{x},\mathbf{a}')\right) + \varepsilon n^2|\mathcal{A}|2^n\|(\mathcal{T}Q)_{\mathrm{tot}}\|_\infty + 3n\varepsilon|\mathcal{A}|^n\|(\mathcal{T}Q)_{\mathrm{tot}}\|_\infty$$

$$\leq \left(h^{\boldsymbol{\pi}^*}(\boldsymbol{x},\mathbf{a})-(n-1)\right)(\mathcal{T}Q)_{\mathrm{tot}}(\boldsymbol{x},\boldsymbol{\pi}^*(\boldsymbol{x}))+\left(\sum_{\mathbf{a}'\in\mathcal{A}^n:h^{\boldsymbol{\pi}^*}(\boldsymbol{x},\mathbf{a}')=n-1}h^{(0)}(\boldsymbol{x},\mathbf{a},\mathbf{a}')(\mathcal{T}Q)_{\mathrm{tot}}(\boldsymbol{x},\mathbf{a}')\right)$$

$$+\varepsilon n^2|\mathcal{A}|^n 2^n\|(\mathcal{T}Q)_{\mathrm{tot}}\|_\infty \tag{97}$$

in which

$$\sum_{\mathbf{a}'\in\mathcal{A}^n:h^{\boldsymbol{\pi}^*}(\boldsymbol{x},\mathbf{a}')=n-1}h^{(0)}(\boldsymbol{x},\mathbf{a},\mathbf{a}')(\mathcal{T}Q)_{\mathrm{tot}}(\boldsymbol{x},\mathbf{a}')$$

$$\leq \left(\sum_{\mathbf{a}'\in\mathcal{A}^n:h^{\boldsymbol{\pi}^*}(\boldsymbol{x},\mathbf{a}')=n-1}h^{(0)}(\boldsymbol{x},\mathbf{a},\mathbf{a}')\right)\max_{\mathbf{a}'\in\mathcal{A}^n:h^{\boldsymbol{\pi}^*}(\boldsymbol{x},\mathbf{a}')=n-1}(\mathcal{T}Q)_{\mathrm{tot}}(\boldsymbol{x},\mathbf{a}')$$

$$=(n-h^{\boldsymbol{\pi}^*}(\boldsymbol{x},\mathbf{a}))\max_{\mathbf{a}'\in\mathcal{A}^n:h^{\boldsymbol{\pi}^*}(\boldsymbol{x},\mathbf{a}')=n-1}(\mathcal{T}Q)_{\mathrm{tot}}(\boldsymbol{x},\mathbf{a}')$$

$$\leq (n-h^{\boldsymbol{\pi}^*}(\boldsymbol{x},\mathbf{a}))\max_{\mathbf{a}'\in\mathcal{A}^n\setminus\{\boldsymbol{\pi}^*(\boldsymbol{x})\}}(\mathcal{T}Q)_{\mathrm{tot}}(\boldsymbol{x},\mathbf{a}')$$

$$=(n-h^{\boldsymbol{\pi}^*}(\boldsymbol{x},\mathbf{a}))\left((\mathcal{T}Q)_{\mathrm{tot}}(\boldsymbol{x},\boldsymbol{\pi}^*)-\mathcal{E}(\mathcal{T}Q)\right) \tag{98}$$

Thus $\forall\boldsymbol{x}\in\mathcal{X},\forall\mathbf{a}\in\mathcal{A}^n\setminus\{\boldsymbol{\pi}^*(\boldsymbol{x})\}$,

$$(\mathcal{T}_D^{\mathrm{LVF}}Q)_{\mathrm{tot}}(\boldsymbol{x},\mathbf{a})$$

$$\leq \left(h^{\boldsymbol{\pi}^*}(\boldsymbol{x},\mathbf{a})-(n-1)\right)(\mathcal{T}Q)_{\mathrm{tot}}(\boldsymbol{x},\boldsymbol{\pi}^*(\boldsymbol{x}))+\left(\sum_{\mathbf{a}'\in\mathcal{A}^n:h^{\boldsymbol{\pi}^*}(\boldsymbol{x},\mathbf{a}')=n-1}h^{(0)}(\boldsymbol{x},\mathbf{a},\mathbf{a}')(\mathcal{T}Q)_{\mathrm{tot}}(\boldsymbol{x},\mathbf{a}')\right)$$

$$+\varepsilon n^2|\mathcal{A}|^n 2^n\|(\mathcal{T}Q)_{\mathrm{tot}}\|_\infty$$

$$\leq \left(h^{\boldsymbol{\pi}^*}(\boldsymbol{x},\mathbf{a})-(n-1)\right)(\mathcal{T}Q)_{\mathrm{tot}}(\boldsymbol{x},\boldsymbol{\pi}^*(\boldsymbol{x}))+(n-h^{\boldsymbol{\pi}^*}(\boldsymbol{x},\mathbf{a}))\left((\mathcal{T}Q)_{\mathrm{tot}}(\boldsymbol{x},\boldsymbol{\pi}^*)-\mathcal{E}(\mathcal{T}Q)\right)+\varepsilon n^2|\mathcal{A}|^n 2^n\|(\mathcal{T}Q)_{\mathrm{tot}}\|_\infty$$

$$=(\mathcal{T}Q)_{\mathrm{tot}}(\boldsymbol{x},\boldsymbol{\pi}^*(\boldsymbol{x}))-(n-h^{\boldsymbol{\pi}^*}(\boldsymbol{x},\mathbf{a}))\mathcal{E}(\mathcal{T}Q)+\varepsilon n^2|\mathcal{A}|^n 2^n\|(\mathcal{T}Q)_{\mathrm{tot}}\|_\infty \tag{99}$$

According to Lemma 6, $\mathcal{E}(\mathcal{T}Q)\geq\mathcal{E}(Q^*)-2\gamma\|V_{\mathrm{tot}}-V^*\|_\infty$. So $\forall\boldsymbol{x}\in\mathcal{X},\forall\mathbf{a}\in\mathcal{A}^n\setminus\{\boldsymbol{\pi}^*(\boldsymbol{x})\}$,

$$(\mathcal{T}_D^{\mathrm{LVF}}Q)_{\mathrm{tot}}(\boldsymbol{x},\mathbf{a})$$

$$\leq (\mathcal{T}Q)_{\mathrm{tot}}(\boldsymbol{x},\boldsymbol{\pi}^*(\boldsymbol{x}))-(n-h^{\boldsymbol{\pi}^*}(\boldsymbol{x},\mathbf{a}))\mathcal{E}(\mathcal{T}Q)+\varepsilon n^2|\mathcal{A}|^n 2^n\|(\mathcal{T}Q)_{\mathrm{tot}}\|_\infty$$

$$\leq (\mathcal{T}Q)_{\mathrm{tot}}(\boldsymbol{x},\boldsymbol{\pi}^*(\boldsymbol{x}))-(n-h^{\boldsymbol{\pi}^*}(\boldsymbol{x},\mathbf{a}))\left(\mathcal{E}(Q^*)-2\gamma\|V_{\mathrm{tot}}-V^*\|_\infty\right)+\varepsilon n^2|\mathcal{A}|^n 2^n\|(\mathcal{T}Q)_{\mathrm{tot}}\|_\infty$$

$$\leq (\mathcal{T}Q)_{\mathrm{tot}}(\boldsymbol{x},\boldsymbol{\pi}^*(\boldsymbol{x}))-\mathcal{E}(Q^*)+2n\gamma\|V_{\mathrm{tot}}-V^*\|_\infty+\varepsilon n^2|\mathcal{A}|^n 2^n\|(\mathcal{T}Q)_{\mathrm{tot}}\|_\infty$$

$$\leq (\mathcal{T}Q)_{\mathrm{tot}}(\boldsymbol{x},\boldsymbol{\pi}^*(\boldsymbol{x}))-\mathcal{E}(Q^*)+2n\gamma\|V_{\mathrm{tot}}-V^*\|_\infty+\varepsilon n^2|\mathcal{A}|^n 2^n(R_{\max}+\gamma\|V_{\mathrm{tot}}\|_\infty)$$

$$\leq (\mathcal{T}Q)_{\mathrm{tot}}(\boldsymbol{x},\boldsymbol{\pi}^*(\boldsymbol{x}))-\mathcal{E}(Q^*)+2n\gamma\|V_{\mathrm{tot}}-V^*\|_\infty+\varepsilon n^2|\mathcal{A}|^n 2^n(R_{\max}+\gamma\|V^*\|_\infty+\gamma\|V_{\mathrm{tot}}-V^*\|_\infty)$$

$$\leq (\mathcal{T}Q)_{\mathrm{tot}}(\boldsymbol{x},\boldsymbol{\pi}^*(\boldsymbol{x}))-\mathcal{E}(Q^*)+2n\gamma\|V_{\mathrm{tot}}-V^*\|_\infty+\varepsilon n^2|\mathcal{A}|^n 2^n(R_{\max}/(1-\gamma)+\gamma\|V_{\mathrm{tot}}-V^*\|_\infty)$$

$$\leq (\mathcal{T}Q)_{\mathrm{tot}}(\boldsymbol{x},\boldsymbol{\pi}^*(\boldsymbol{x}))-\mathcal{E}(Q^*)+2n\gamma\|V_{\mathrm{tot}}-V^*\|_\infty+\delta \tag{100}$$

$\square$

**Lemma 8.** *Let $\mathcal{B}$ denote a subspace of value functions*

$$\mathcal{B}=\left\{Q\in\mathcal{Q}^{LVF}\,\middle|\,\mathcal{E}(Q)\geq 0,\ \|V_{tot}-V^*\|_\infty\leq\frac{1}{8n\gamma}\mathcal{E}(Q^*)\right\} \tag{101}$$

*Given a dataset $D$ generated by the optimal policy $\boldsymbol{\pi}^*$ with $\epsilon$-greedy exploration,*

$$\forall 0<\varepsilon\leq\frac{(1-\gamma)\mathcal{E}(Q^*)}{n^3|\mathcal{A}|^n 2^{n+4}(R_{max}/(1-\gamma)+\mathcal{E}(Q^*)/(8n))} \tag{102}$$

*we have $\forall Q\in\mathcal{B}$, $\mathcal{T}_D^{LVF}Q\in\hat{\mathcal{B}}\subset\mathcal{B}$ where*

$$\hat{\mathcal{B}}=\left\{Q\in\mathcal{Q}^{LVF}\,\middle|\,\mathcal{E}(Q)>0,\ \|V_{tot}-V^*\|_\infty\leq\frac{1}{8n\gamma}\mathcal{E}(Q^*)\right\} \tag{103}$$

*Proof.* According to Lemma 4, with the condition

$$0 < \varepsilon \le \frac{\mathcal{E}(Q^*)/4}{n^2|\mathcal{A}|^n 2^{n+1}(R_{\max}/(1-\gamma) + \mathcal{E}(Q^*)/(8n))} \le \frac{\mathcal{E}(Q^*)/4}{n^2|\mathcal{A}|^n 2^{n+1}(R_{\max} + \gamma\|V_{\text{tot}}\|_\infty)} \tag{104}$$

we have $\forall Q \in \mathcal{B}, \forall \boldsymbol{x} \in \boldsymbol{\mathcal{X}}$,

$$\left|(\mathcal{T}_D^{\text{LVF}}Q)_{\text{tot}}(\boldsymbol{x}, \boldsymbol{\pi}^*(\boldsymbol{x})) - (\mathcal{T}Q)_{\text{tot}}(\boldsymbol{x}, \boldsymbol{\pi}^*(\boldsymbol{x}))\right| \le \frac{1}{4}\mathcal{E}(Q^*) \tag{105}$$

which implies $\forall Q \in \mathcal{B}, \forall \boldsymbol{x} \in \boldsymbol{\mathcal{X}}$,

$$(\mathcal{T}_D^{\text{LVF}}Q)_{\text{tot}}(\boldsymbol{x}, \boldsymbol{\pi}^*(\boldsymbol{x})) \ge (\mathcal{T}Q)_{\text{tot}}(\boldsymbol{x}, \boldsymbol{\pi}^*(\boldsymbol{x})) - \frac{1}{4}\mathcal{E}(Q^*). \tag{106}$$

According to Lemma 7, with the condition

$$\begin{aligned}0 < \varepsilon &\le \frac{\mathcal{E}(Q^*)/4}{n^2|\mathcal{A}|^n 2^n(R_{\max}/(1-\gamma) + \mathcal{E}(Q^*)/(8n))} \\ &\le \frac{\mathcal{E}(Q^*)/4}{n^2|\mathcal{A}|^n 2^n(R_{\max}/(1-\gamma) + \gamma\|V_{\text{tot}} - V^*\|_\infty)}\end{aligned} \tag{107}$$

we have $\forall Q \in \mathcal{B}, \forall \boldsymbol{x} \in \boldsymbol{\mathcal{X}}, \forall a \in \mathcal{A}^n \setminus \{\boldsymbol{\pi}^*(\boldsymbol{x})\}$,

$$\begin{aligned}(\mathcal{T}_D^{\text{LVF}}Q)_{\text{tot}}(\boldsymbol{x}, \mathbf{a}) &\le (\mathcal{T}Q)_{\text{tot}}(\boldsymbol{x}, \boldsymbol{\pi}^*(\boldsymbol{x})) - \mathcal{E}(Q^*) + 2n\gamma\|V_{\text{tot}} - V^*\|_\infty + \frac{1}{4}\mathcal{E}(Q^*) \\ &\le (\mathcal{T}Q)_{\text{tot}}(\boldsymbol{x}, \boldsymbol{\pi}^*(\boldsymbol{x})) - \mathcal{E}(Q^*) + \frac{1}{4}\mathcal{E}(Q^*) + \frac{1}{4}\mathcal{E}(Q^*) \\ &= (\mathcal{T}Q)_{\text{tot}}(\boldsymbol{x}, \boldsymbol{\pi}^*(\boldsymbol{x})) - \frac{1}{2}\mathcal{E}(Q^*) \\ &< (\mathcal{T}_D^{\text{LVF}}Q)_{\text{tot}}(\boldsymbol{x}, \boldsymbol{\pi}^*(\boldsymbol{x}))\end{aligned} \tag{108}$$

which implies $\mathcal{E}(\mathcal{T}_D^{\text{LVF}}Q) > 0$.

According to Lemma 5, with the condition

$$\begin{aligned}0 < \varepsilon &\le \frac{(1-\gamma)\mathcal{E}(Q^*)}{\gamma n^3|\mathcal{A}|^n 2^{n+4}(R_{\max}/(1-\gamma) + \mathcal{E}(Q^*)/(8n))} \\ &\le \frac{(1-\gamma)\mathcal{E}(Q^*)}{\gamma n^3|\mathcal{A}|^n 2^{n+4}(R_{\max}/(1-\gamma) + \gamma\|V_{\text{tot}}^{\boldsymbol{\pi}^*} - V^*\|_\infty)},\end{aligned} \tag{109}$$

we have $\forall Q \in \mathcal{B}, \forall \boldsymbol{x} \in \boldsymbol{\mathcal{X}}$,

$$\begin{aligned}\left|(\mathcal{T}_D^{\text{LVF}}V)(\boldsymbol{x}) - V^*(\boldsymbol{x})\right| &= \left|(\mathcal{T}_D^{\text{LVF}}Q)_{\text{tot}}(\boldsymbol{x}, \boldsymbol{\pi}^*(\boldsymbol{x})) - V^*(\boldsymbol{x})\right| \\ &\le \gamma\|V_{\text{tot}}^{\boldsymbol{\pi}^*} - V^*\|_\infty + \frac{1-\gamma}{8n\gamma}\mathcal{E}(Q^*) \le \frac{1}{8n\gamma}\mathcal{E}(Q^*).\end{aligned} \tag{110}$$

Combing Eq. (104), (107), and (109), the overall condition is

$$0 < \varepsilon \le \frac{(1-\gamma)\mathcal{E}(Q^*)}{n^3|\mathcal{A}|^n 2^{n+4}(R_{\max}/(1-\gamma) + \mathcal{E}(Q^*)/(8n))} \tag{111}$$

$\square$

**Lemma 3.** *There exists a threshold $\delta > 0$ such that the on-policy Bellman operator $\mathcal{T}_\epsilon^{LVF}$ is closed in the following subspace $\mathcal{B} \subset \mathcal{Q}^{LVF}$, when the hyper-parameter $\epsilon$ is sufficiently small.*

$$\mathcal{B} = \left\{ Q \in \mathcal{Q}^{LVF} \,\middle|\, \boldsymbol{\pi}_Q = \boldsymbol{\pi}^*, \ \max_{\boldsymbol{x} \in \boldsymbol{\mathcal{X}}} |Q_{tot}(\boldsymbol{x}, \boldsymbol{\pi}^*(\boldsymbol{x})) - V^*(\boldsymbol{x})| \le \delta \right\}$$

*Formally, $\exists \delta > 0, \exists \epsilon > 0, \forall Q \in \mathcal{B}$, there must be $\mathcal{T}_\epsilon^{LVF}Q \in \mathcal{B}$.*

*Proof.* It is implied by Lemma 8. $\square$

**Theorem 4** (Formal version of Theorem 2). *Besides Lemma 3, Algorithm 2 will have a fixed point value function expressing the optimal policy if the hyper-parameter $\epsilon$ is sufficiently small.*

*Proof.* Notice that the observation-value function $V_{\text{tot}}$ is sufficient to determine the target values, so the subspace $\mathcal{B}$ defined in Lemma 8 is a compact and convex space in terms of $V_{\text{tot}}$. The operator $\mathcal{T}_D^{\text{LVF}}$ is a continuous mapping because it only involves elementary functions. According to Brouwer's Fixed Point Theorem [47], there exist $Q \in \mathcal{B}$ satisfying $\mathcal{T}_D^{\text{LVF}} Q \in \mathcal{B}$. In addition, according to the definition stated in Eq. (103), the fixed point must represent the unique optimal policy since it cannot lie on the boundary with $\mathcal{E}(Q) = 0$. $\square$

## D    Omitted Proofs for Theorem 3

**Lemma 9.** *The empirical Bellman operator $\mathcal{T}_D^{IGM}$ stated in Definition 2 is a $\gamma$-contraction, i.e., the following important property of the standard Bellman optimality operator $\mathcal{T}$ will hold for $\mathcal{T}_D^{IGM}$.*

$$\forall Q_{tot}, Q'_{tot} \in \mathcal{Q}, \quad \|\mathcal{T}Q_{tot} - \mathcal{T}Q'_{tot}\|_\infty \le \gamma \|Q_{tot} - Q'_{tot}\|_\infty \tag{112}$$

*Proof.* First, we want to prove $\left(\mathcal{T}_D^{\text{IGM}} Q\right)_{\text{tot}} = r(s, \mathbf{a}) + \gamma \mathbb{E}[V_{\text{tot}}(\boldsymbol{x}')]$ which indicates that the empirical Bellman error is zero:

$$err_D^{\text{IGM}} \equiv \min_{Q \in \mathcal{Q}^{\text{IGM}}} \sum_{(\boldsymbol{x}, \mathbf{a}) \in \mathcal{X} \times \mathbf{A}} p_D(\mathbf{a}|\boldsymbol{x}) \left(y^{(t)}(\boldsymbol{x}, \mathbf{a}) - Q_{\text{tot}}(\boldsymbol{x}, \mathbf{a})\right)^2 = 0. \tag{113}$$

Let $\mathbf{a}^{*,(t)} = \left[a_i^{*,(t)}\right]_{i=1}^n = \arg\max_{\mathbf{a} \in \mathbf{A}} y^{(t)}(\boldsymbol{x}, \mathbf{a})$. We construct $Q_{\text{tot}}(\boldsymbol{x}, \mathbf{a}) = y^{(t)}(\boldsymbol{x}, \mathbf{a})$ and its corresponding local action-value functions $[Q_i]_{i=1}^n$ satisfying IGM principle:

$$Q_i(x_i, a_i) = \begin{cases} 1, & \text{when } a_i = a_i^{*,(t)}, \\ 0, & \text{when } a_i \ne a_i^{*,(t)}. \end{cases} \tag{114}$$

To avoid the multiple solutions of $\arg\max$ operator in $\mathbf{a}^{*,(t)}$, we consider the lexicographic order of joint actions as the second priority. Thus, we illustrate the completeness of IGM function class in our problem setting. According to Eq. (113) and Lemma 1.5 in RL textbook [67], we can prove that $\mathcal{T}_D^{\text{IGM}}$ is a $\gamma$-contraction, $\mathcal{T}_D^{\text{IGM}}$ is a $\gamma$-contraction in Dec-ROMDPs while using reactive function classes. $\square$

**Theorem 3.** *FQI-IGM globally converges to the optimal value function in arbitrary Dec-ROMDPs.*

*Proof.* Let $Q^*(\boldsymbol{x}, \mathbf{a}) = max_{\boldsymbol{\pi} \in \Pi} Q^{\boldsymbol{\pi}}(\boldsymbol{x}, \mathbf{a})$ where $\Pi$ is the space of all policies. According to Lemma 9 and Theorem 1.4 in RL textbook [67], we have that

- There exists a stationary and deterministic policy $\boldsymbol{\pi}$ such that $Q_{\text{tot}}^{\boldsymbol{\pi}} = Q_{\text{tot}}^*$.

- A vector $Q_{\text{tot}} \in \mathbb{R}^{|\mathcal{S}| \times |\mathcal{A}|^n}$ is equal to $Q_{\text{tot}}^*$ if and only if it satisfies $Q_{\text{tot}} = \left(\mathcal{T}_D^{\text{IGM}} Q\right)_{\text{tot}}$.

- $\forall Q'_{\text{tot}} \in \mathcal{Q}^{\text{IGM}}$,

$$\begin{aligned} \left\|Q_{\text{tot}}^* - \left(\mathcal{T}_D^{\text{IGM}} Q'\right)_{\text{tot}}\right\|_\infty &= \left\|\left(\mathcal{T}_D^{\text{IGM}} Q^*\right)_{\text{tot}} - \left(\mathcal{T}_D^{\text{IGM}} Q'\right)_{\text{tot}}\right\|_\infty \\ &\le \gamma \left\|Q_{\text{tot}}^* - Q'_{\text{tot}}\right\|_\infty. \end{aligned} \tag{115}$$

Thus, FQI-IGM will globally converge to optimal value function. $\square$

## E    Experiment Settings and Implementation Details

### E.1    Implementation Details and Evaluation Setting

We adopt the PyMARL [20] implementation with default hyper-parameters to investigate state-of-the-art multi-agent Q-learning algorithms: VDN [10], QMIX [12], QTRAN [13], and QPLEX [19]. The training time of these algorithms on an NVIDIA RTX 2080TI GPU is about 4 hours to 12 hours, which is depended on the number of agents and the episode length limit of each map. The performance measure of StarCraft II tasks is the percentage of episodes in which RL agents defeat all enemy units within the limited time constraints, called *test win rate*. The dataset providing off-policy exploration is constructed by training a behavior

| Map Name | Replay Buffer Size | Behaviour Test Win Rate | Behaviour Policy |
|---|---|---|---|
| 2s3z | 20k episodes | 91.2% | VDN |
| 3s5z | 20k episodes | 77.5% | VDN |
| 2s_vs_1sc | 20k episodes | 99.6% | VDN |
| 3s_vs_5z | 20k episodes | 94.2% | VDN |
| 1c3s5z | 30k episodes | 92.1% | VDN |
| 3c7z | 30k episodes | 94.4% | VDN |
| 5m_vs_6m | 50k episodes | 61.7% | VDN |
| 10m_vs_11m | 50k episodes | 88.7% | VDN |
| 3h_vs_4z | 50k episodes | 83.1% | VDN |

Table 1: The dataset configurations of offline data collection setting.

| Map Name | Ally Units | Enemy Units |
|---|---|---|
| 2s3z | 2 Stalkers & 3 Zealots | 2 Stalkers & 3 Zealots |
| 3s5z | 3 Stalkers & 5 Zealots | 3 Stalkers & 5 Zealots |
| 2s_vs_1sc | 2 Stalkers | 1 Spine Crawler |
| 3s_vs_5z | 3 Stalkers | 5 Zealots |
| 1c3s5z | 1 Colossus, 3 Stalkers & 5 Zealots | 1 Colossus, 3 Stalkers & 5 Zealots |
| 3c7z | 3 Colossi & 7 Zealots | 3 Colossi & 7 Zealots |
| 5m_vs_6m | 5 Marines | 6 Marines |
| 10m_vs_11m | 10 Marines | 11 Marines |
| 3h_vs_4z | 3 Hydralisks | 4 Zealots |

Table 2: SMAC challenges.

policy of VDN and collecting its 20k, 30k or 50k experienced episodes. The dataset configurations are shown in Table 1. We investigate five multi-agent Q-learning algorithms over 6 random seeds, which includes 3 different datasets and evaluates two seeds on each dataset. We train 300 epochs to evaluate the learning performance with a given static dataset, of which 32 episodes are trained in each update, and 160k transitions are trained for each epoch totally. Moreover, the training process of behavior policy is the same as that discussed in PyMARL [20], which has collected a total of 2 million timestep data and anneals the hyper-parameter $\epsilon$ of $\epsilon$-greedy exploration strategy linearly from 1.0 to 0.05 over 50k timesteps. The target network will be updated periodically after training every 200 episodes. We call this period of 200 episodes an *Iteration*, which corresponds to an iteration of FQI-LVF (see Definition 1).

### E.2 Two-State Example

In the two-state example shown in Figure 1a, due to the GRU-based implementation of the finite-horizon paradigm in the above five deep multi-agent Q-learning algorithms, we assume that two agents starting from state $s_2$ have 100 environmental steps executed by a uniform $\epsilon$-greedy exploration strategy (*i.e.*, $\epsilon = 1$). We use this long-term horizon pattern and uniform $\epsilon$-greedy exploration methods to approximate an infinite-horizon learning paradigm with uniform data distribution. We adopt $\gamma = 0.9$ to implement FQI-LVF and deep MARL algorithms. In the FQI-LVF framework, $V_{max} = \frac{1}{1-\gamma} = 100$ as shown in Figure 1c. Figure 1b demonstrates that *Optimal* line is approximately $\sum_{i=0}^{99} \gamma^i = 63.4$ in one episode of 100 timesteps.

### E.3 StarCraft II

StarCraft II unit micromanagement tasks consider a combat game of two groups of agents, where StarCraft II takes built-in AI to control enemy units, and MARL algorithms can control each ally unit to fight the enemies. Units in two groups can contain different types of soldiers, but these soldiers in the same group should belong to the same race. The action space of each agent includes no-op, move [direction], attack [enemy id], and stop. At each timestep, agents choose to move or attack in continuous maps. MARL agents will get a global reward equal to the amount of damage done to enemy units. Moreover, killing one enemy unit and winning the combat will bring additional bonuses of 10 and 200, respectively. The maps of SMAC challenges in this paper are introduced in Table 2 in the episodes of 100 timesteps. To approximate the Dec-ROMDP setting, we concatenate the global state with the local observations for each agent to handle partial observability.

All networks are trained using GeForce GTX 1080 Ti and Intel(R) Xeon(R) CPU E5-2630 v4 @ 2.20GHz. Each single learning curve can be completed within 36 hours.

# F Experiments on a Two-Player Matrix Game

## F.1 Value Estimation in Multi-Agent Q-Learning Algorithms

| $a_1$ / $a_2$ | $\mathcal{A}^{(1)}$ | $\mathcal{A}^{(2)}$ | $\mathcal{A}^{(3)}$ |
|---|---|---|---|
| $\mathcal{A}^{(1)}$ | **8** | -12 | -12 |
| $\mathcal{A}^{(2)}$ | -12 | 0 | 0 |
| $\mathcal{A}^{(3)}$ | -12 | 0 | 0 |

(a) Payoff matrix

| $a_1$ / $a_2$ | $\mathcal{A}^{(1)}$ | $\mathcal{A}^{(2)}$ | $\mathcal{A}^{(3)}$ |
|---|---|---|---|
| $\mathcal{A}^{(1)}$ | **7.98** | -12.09 | -12.10 |
| $\mathcal{A}^{(2)}$ | -12.18 | -0.02 | -0.02 |
| $\mathcal{A}^{(3)}$ | -12.11 | -0.03 | -0.03 |

(b) $Q_{\text{tot}}$ of QPLEX

| $a_1$ / $a_2$ | $\mathcal{A}^{(1)}$ | $\mathcal{A}^{(2)}$ | $\mathcal{A}^{(3)}$ |
|---|---|---|---|
| $\mathcal{A}^{(1)}$ | **8.00** | -12.00 | -12.00 |
| $\mathcal{A}^{(2)}$ | -12.00 | -0.00 | 0.00 |
| $\mathcal{A}^{(3)}$ | -12.00 | 0.00 | 0.00 |

(c) $Q_{\text{tot}}$ of QTRAN

| $a_1$ / $a_2$ | $\mathcal{A}^{(1)}$ | $\mathcal{A}^{(2)}$ | $\mathcal{A}^{(3)}$ |
|---|---|---|---|
| $\mathcal{A}^{(1)}$ | -7.98 | -7.98 | -7.98 |
| $\mathcal{A}^{(2)}$ | -7.98 | -0.00 | **-0.00** |
| $\mathcal{A}^{(3)}$ | -7.98 | -0.00 | -0.00 |

(d) $Q_{\text{tot}}$ of QMIX

| $a_1$ / $a_2$ | $\mathcal{A}^{(1)}$ | $\mathcal{A}^{(2)}$ | $\mathcal{A}^{(3)}$ |
|---|---|---|---|
| $\mathcal{A}^{(1)}$ | -6.23 | -4.90 | -4.90 |
| $\mathcal{A}^{(2)}$ | -4.90 | **-3.57** | -3.57 |
| $\mathcal{A}^{(3)}$ | -4.90 | -3.57 | -3.57 |

(e) $Q_{\text{tot}}$ of VDN

| $a_1$ / $a_2$ | $\mathcal{A}^{(1)}$ | $\mathcal{A}^{(2)}$ | $\mathcal{A}^{(3)}$ |
|---|---|---|---|
| $\mathcal{A}^{(1)}$ | -6.22 | -4.89 | -4.89 |
| $\mathcal{A}^{(2)}$ | -4.89 | **-3.56** | -3.56 |
| $\mathcal{A}^{(3)}$ | -4.89 | -3.56 | -3.56 |

(f) $Q_{\text{tot}}$ of FQI-LVF

Table 3: (a) Payoff matrix of the one-step game. Boldface means the optimal joint action selection from payoff matrix. (b-f) Joint action-value functions $Q_{\text{tot}}$ estimated by a suite of algorithms. Boldface means the greedy joint action selection from $Q_{\text{tot}}$.

Table 3 reveals the following observations regarding the representational capacity of these value factorization structures:

- QPLEX and QTRAN, two algorithms using IGM value factorization structure, can almost perfectly fit the payoff matrix.
- QMIX cannot find the ground truth best actions, since its monotonic value factorization structure cannot express the given payoff matrix.
- The value estimation generated by tabular FQI-LVF matches its deep-learning-based implementation (i.e., VDN).

## F.2 The Learning Curve of Table 3e

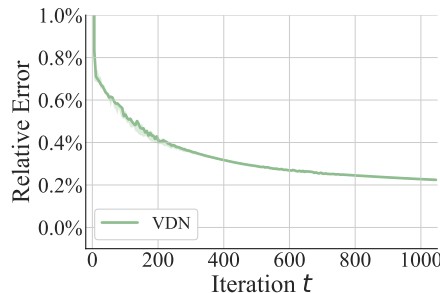

Figure 3: The learning curve of Table 3e. Every iteration contains 200 gradient steps. The relative error is defined as $\|Q_{\text{tot}}^{\text{FQI-LVF}} - Q_{\text{tot}}^{\text{VDN}}\|_\infty / \|Q_{\text{tot}}^{\text{FQI-LVF}}\|_\infty$.

# G Ablation Studies on Network Capacity

## G.1 Ablation Studies in Matrix Game

| $a_2$ \ $a_1$ | $\mathcal{A}^{(1)}$ | $\mathcal{A}^{(2)}$ | $\mathcal{A}^{(3)}$ |
|---|---|---|---|
| $\mathcal{A}^{(1)}$ | **8** | -12 | -12 |
| $\mathcal{A}^{(2)}$ | -12 | 0 | 0 |
| $\mathcal{A}^{(3)}$ | -12 | 0 | 0 |

(a) Payoff of matrix game

| $a_2$ \ $a_1$ | $\mathcal{A}^{(1)}$ | $\mathcal{A}^{(2)}$ | $\mathcal{A}^{(3)}$ |
|---|---|---|---|
| $\mathcal{A}^{(1)}$ | **7.98** | -12.09 | -12.10 |
| $\mathcal{A}^{(2)}$ | -12.18 | -0.02 | -0.02 |
| $\mathcal{A}^{(3)}$ | -12.11 | -0.03 | -0.03 |

(b) $Q_{\text{tot}}$ of QPLEX

| $a_2$ \ $a_1$ | $\mathcal{A}^{(1)}$ | $\mathcal{A}^{(2)}$ | $\mathcal{A}^{(3)}$ |
|---|---|---|---|
| $\mathcal{A}^{(1)}$ | **8.00** | -12.00 | -12.00 |
| $\mathcal{A}^{(2)}$ | -12.00 | -0.00 | 0.00 |
| $\mathcal{A}^{(3)}$ | -12.00 | 0.00 | 0.00 |

(c) $Q_{\text{tot}}$ of QTRAN

| $a_2$ \ $a_1$ | $\mathcal{A}^{(1)}$ | $\mathcal{A}^{(2)}$ | $\mathcal{A}^{(3)}$ |
|---|---|---|---|
| $\mathcal{A}^{(1)}$ | -6.24 | -4.90 | -4.90 |
| $\mathcal{A}^{(2)}$ | -4.90 | **-3.57** | -3.57 |
| $\mathcal{A}^{(3)}$ | -4.90 | -3.57 | -3.57 |

(d) $Q_{\text{tot}}$ of Large-VDN

| $a_2$ \ $a_1$ | $\mathcal{A}^{(1)}$ | $\mathcal{A}^{(2)}$ | $\mathcal{A}^{(3)}$ |
|---|---|---|---|
| $\mathcal{A}^{(1)}$ | -8.03 | -8.03 | -8.03 |
| $\mathcal{A}^{(2)}$ | -8.03 | -0.01 | **-0.01** |
| $\mathcal{A}^{(3)}$ | -8.03 | -0.01 | -0.01 |

(e) $Q_{\text{tot}}$ of Large-QMIX

Table 4: (a-c) The ground-truth payoff matrix and the joint action-value functions of QPLEX and QTRAN. (d-e) The joint action-value functions $Q_{\text{tot}}$ of Large-VDN and Large-QMIX. Boldface means the greedy joint action selection from $Q_{\text{tot}}$.

To address the concern that QPLEX naturally uses more hidden parameters than VDN and QMIX, which may also improve its representational capacity. To demonstrate that the performance gap between QPLEX and other methods does not come from the difference in term of the number of parameters, we increase the number of neurons in VDN and QMIX so that they have the comparable number of parameters as QPLEX. Formally, Large-VDN and Large-QMIX have the similar number of parameters as QPLEX. The experiment results are presented in Table 4, both the "Large-" versions of VDN and QMIX cannot represent an accurate value function in this matrix game. Increasing the number of parameters cannot address the limitations of VDN and QMIX on representational capacity.

## G.2 Ablation Studies in StarCraft II Benchmark Tasks

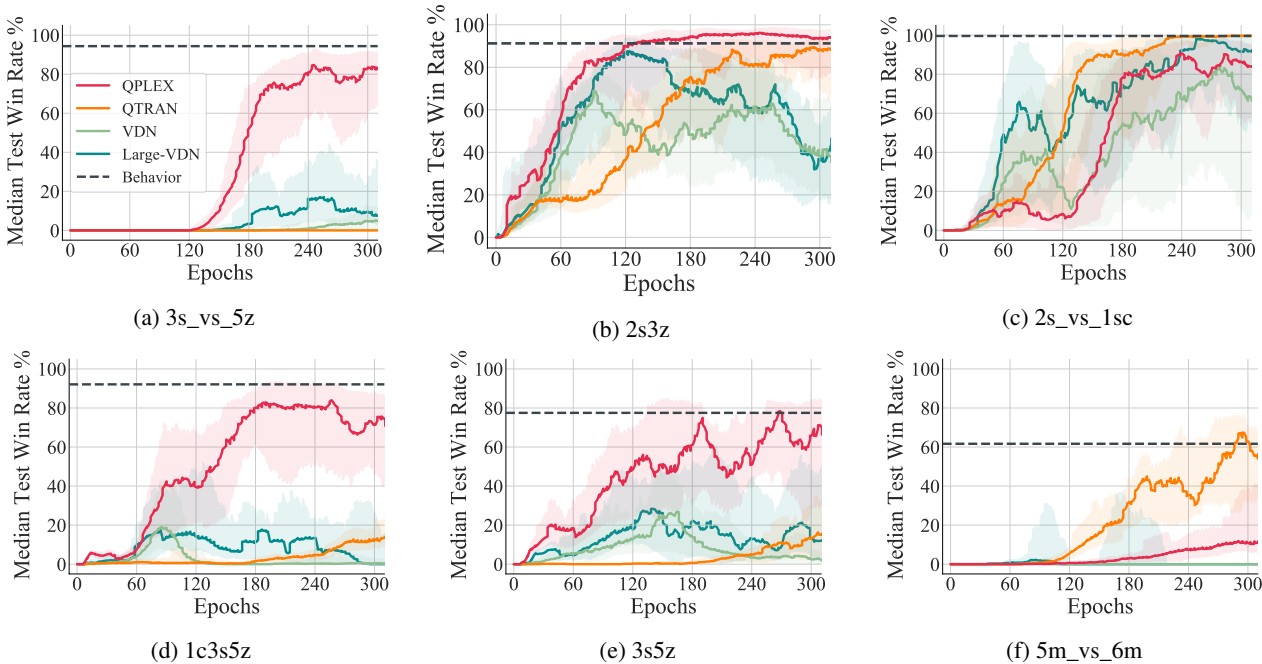

Figure 4: Evaluating the performance of Large-VDN with a given static dataset.

In addition to the ablation study in the matrix game, Figure 4 and Figure 5 present the ablation studies in StarCraft II benchmark tasks with offline data collection. In comparison to the standard versions of VDN and QMIX, we introduce Large-VDN and

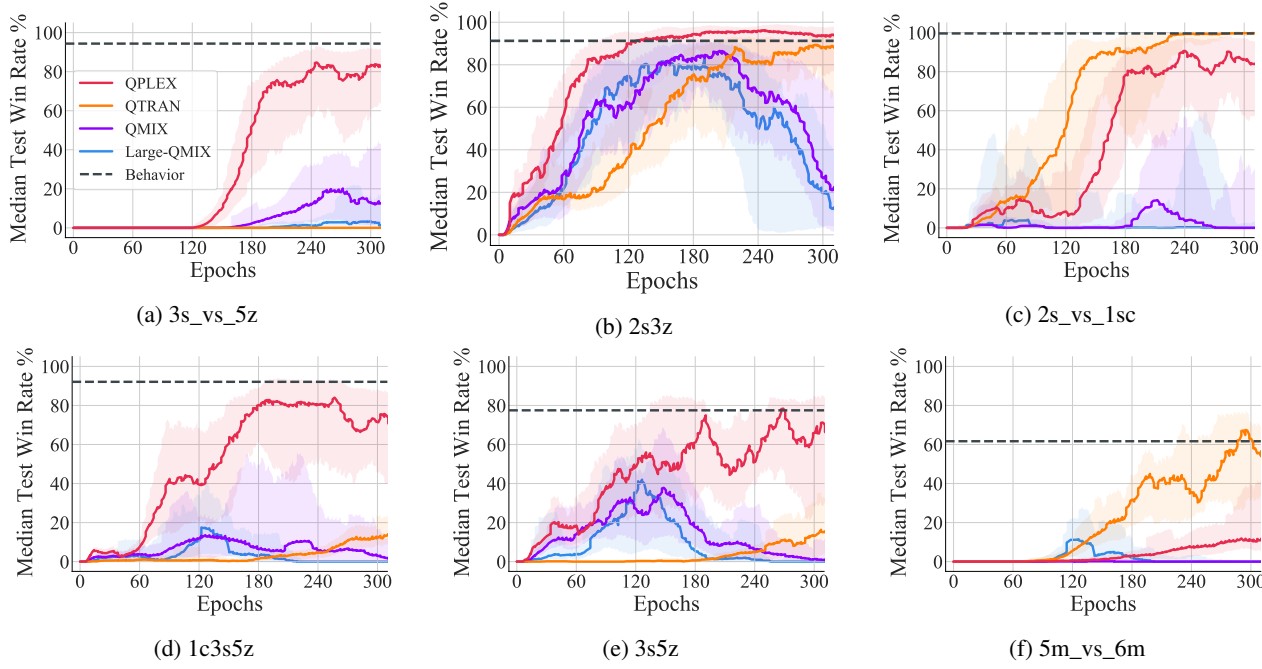

Figure 5: Evaluating the performance of Large-QMIX with a given static dataset.

Large-QMIX, which have a similar number of parameters as QPLEX. As shown in Figure 4, increasing parameters can benefit VDN in some easy maps such as 2s3z and 2s_vs_1sc, but it cannot provide fundamental improvement in harder tasks. As shown in Figure 5, the effects of increasing parameters are rather weak for QMIX. These experiments demonstrate that increasing the number of parameters cannot address the limitations of VDN and QMIX on representational capacity.

# H  Additional Experiments on Two-State Example

**Remark.**  Assumption 1 assumes that the dataset $D$ is collected by a decentrailized and exploratory policy $\boldsymbol{\pi}^D$. All algorithms discussed in the paper, including VDN, QMIX, QTRAN, and QPLEX, learn decentralized policies, which are executed in a decentralized manner. To investigate the dependency of our theoretical implications on Assumption 1, we provide an experiment to evaluate the performance of deep multi-agent Q-learning algorithms on the general datasets. Figure 6 present the learning curves of VDN, QMIX, QPLEX, and QTRAN in the example shown in Figure 1a with a specific dataset $D$ constructed by a parameter $\eta$ as follows:

$$\forall s \in \mathcal{S}, \quad p_D(\mathcal{A}^{(1)}, \mathcal{A}^{(2)} \mid s) = \begin{pmatrix} 0.5\eta + 0.25(1-\eta) & 0.25(1-\eta) \\ 0.25(1-\eta) & 0.5\eta + 0.25(1-\eta) \end{pmatrix}.$$

As shown in Figure 6, the choice of parameter $\eta$ has no impacts on the performance of QPLEX and QTRAN, which matches the fact that Theorem 3 does not rely on the assumption of the decentralized data collection. As the extension of Proposition 2, VDN and QMIX empirically suffer from unbounded divergence when the dataset is not collected by a decentrailized policy. The only exception is the case of $\eta = 1$, in which the dataset only contains two kinds of joint actions. In this case, the given example degenerates to a single-agent MDP because agents only perform the same actions in the dataset. As a result, VDN and QMIX would not diverge in this special situation.

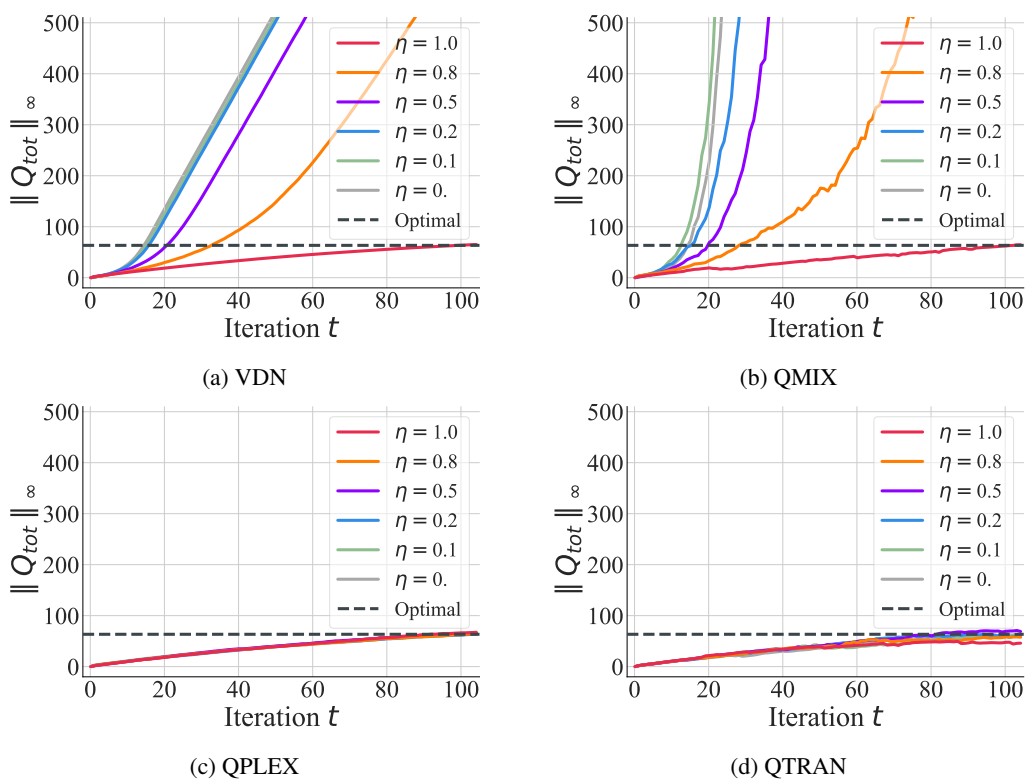

Figure 6: The learning curves of $\|Q_{\text{tot}}\|_\infty$ while running several deep multi-agent Q-learning algorithms with a new dataset.

# I   Diagnosing the Divergence of QMIX

As shown in Figure 6, QMIX also suffers from unbounded divergence in this MMDP. To investigate this phenomenon, we conduct an ablation study and find that the divergence is caused by choice of the activation function. The default implementation of QMIX uses `Elu` as activation, which contains a linear component on the positive side. We hypothesize that the behavior of this linear piece may resemble that of linear value factorization which leads to unbounded divergence. As shown in Figure 7, using `Tanh` instead of `Elu` can prevent the divergence of QMIX in this example task.

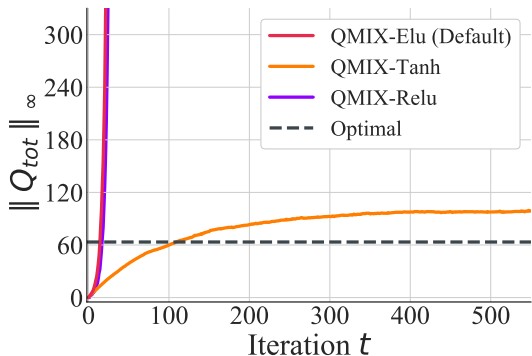

Figure 7: The learning curves of $\|Q_{\text{tot}}\|_\infty$ while running QMIX with different activation functions.