# OpenReview forum: "Towards Understanding Cooperative Multi-Agent Q-Learning with Value Factorization"
_NeurIPS.cc/2021/Conference — NeurIPS 2021 Poster_

### Official Review · Reviewer_8PFC · 2021-07-16

**Rating:** 6
**Confidence:** 3

**Summary:**

The paper proposes the multi-agent version of fitted q learning to analyze two kinds of value decomposition. They prove that on-policy training or richer joint value function classes can improve its local or global convergence properties.

**Limitations And Societal Impact:**

Yes

**Main Review:**

The method is interesting, and the paper is easy to follow. The authors provide the theoretical view of some existing methods, like VDN and QTRAN. There are some concerns.

1. Since QMIX is one of the most popular value decomposition methods, is it possible to analyze it by the theoretical framework? Any conclusion?
2. Since the analysis is based on FQI, is it possible to directly compare  FQI-LVF and FQI-IGM rather than deep versions of them? Then the result would be more convincing.


**Time Spent Reviewing:**

2 hours

---

> ### Author Response · Authors · 2021-08-10
> **Response to Reviewer 8PFC**
>
> Thanks for the comments. We provide clarification to your questions and concerns as below. If our response does not fully address your concerns, please post additional questions and we will be happy to have further discussions.
>
> **Q1: Any conclusions about QMIX?**
>
> Although we cannot derive a closed-form analysis on multi-agent Q-learning with monotonic function class (i.e., QMIX), we establish an empirical study on its behavior in the same evaluation metric as other experiments presented in this paper (see Appendix G and H). According to the experiment results, we show that the algorithmic properties of QMIX can be regarded as an interpolation between linear and IGM value factorization. Its function expressiveness is richer than that of VDN, which gives it better ability in complex domains such as StarCraft benchmark tasks. However, the monotonic function used by QMIX cannot represent all valid joint value functions. In the most unfavorable cases, QMIX may suffer from the same issue of unbounded divergence as linear value factorization (see Figure 6(b) in Appendix).
>
> **Q2: A direct comparison between FQI-LVF and FQI-IGM**
>
> Our theoretical analysis can be regarded as a direct comparison between FQI-LVF and FQI-IGM, which characterizes their convergence properties. In practice, the IGM consistency in FQI-IGM needs to be represented by a carefully designed architecture (i.e., QPLEX), which requires gradient-based optimization. That is why we consider deep-learning-based implementations for experiments.

---

### Official Review · Reviewer_1PHZ · 2021-07-16

**Rating:** 6
**Confidence:** 4

**Summary:**

This paper investigates Q-learning algorithms that factorise the joint-action Q-values in a cooperative MARL setting under CTDE.
Specifically, a fitted Q-iteration framework is used in order to focus on the theoretical properties associated with these approaches.
The theoretical analysis provides a better understanding of linear value factorisation: how it can be viewed as performing implicit credit assignment, situations where it can diverge, and the stabilising effect of training with on-policy data. Additionally, the paper theoretically demonstrates the benefits of being able to represent a wider class of joint-action Q-values by proving convergence in the same setting.

**Limitations And Societal Impact:**

Can you comment more on the results in Figure 2, specifically the scenarios where QTRAN significantly outperforms QPLEX.
Do you have any hypotheses for why this is the case? Is it a failing on the optimisation part of QPLEX (nn architectures), or is it related to the way QPLEX always maintains IGM consistency whereas QTRAN does not? I ask because this is an opportunity to discuss the potential limitations of the theory. Although QTRAN and QPLEX can both theoretically represent the same function class, their empirical performance can differ significantly. Specifically, what aspects of their respective algorithms does your theory not capture that could be useful for future work/analysis.

**Main Review:**

I liked this paper overall, but would like to see the presentation adjusted to mention the links with existing work and the importance of some results. I am open to increasing my score based on the rebuttal (and on the other reviews) since I do think this paper makes a valuable contribution to the literature.

Is the setting of a Dec-ROMDP really important for the analysis in this paper, or would an MMDP suffice?
In Appendix A.2 you mention that it is used for "bypassing the barrier of partial observability in theoretical analyses".
It doesn't appear that the particular structure/dimensionality of the observation space is used in the analysis so I don't see the necessity of using the Dec-ROMDP formalism over a simpler formalism that drops the partial-observability entirely.
Furthermore, since you make additional assumptions that only the most recent observation is required to know all about the true state I don't see much of an actual difference between your setting and just assuming full observability.

This paper is entirely missing the links to the relevant work done in the Weighted QMIX paper.
The analysis done there is very similar to the FMA-FQI considered here, except it doesn't include an explicit notion of a dataset D and instead assumes all state-action pairs are updated. I appreciate that considering the dataset brings additional complexity and richness to the analysis, but it seems an oversight to completely ignore it since it is very related prior work.
Line 360 in the conclusion suggests that the theoretical properties of QMIX have not been investigated at all in this kind of framework, but they have been in the Weighted QMIX paper.
Furthermore, for a lot of the analysis the role the dataset/exploratory policy plays is very mild which brings the two analyses even closer.
I'm not saying that there is a lack of novelty in this paper, just that there is very related work that should be also mentioned.

Could you expand on the extra importance weight (n-1)/n, and why it is more consistent/meaningful in Line 214?
In the scenario where all joint-actions lead to the same rewards, then assigning 0 is the intended behaviour of COMA (i.e. no advantage to taking any action compared to a baseline policy). I don't see why assigning 1/n to each agent is necessarily beneficial in that scenario.

For Proposition 1, could the same argument/example as in Weighted QMIX be used to show this in a much simpler manner (Section 3.1, under the paragraph "T^∗_{Qmix} is not a contraction")? Essentially that there isn't necessarily a unique minimiser of the loss.
Related to this, how do you establish which entry of the argmin to take in Eqns 3 and 6 when there are multiple?
I see that Assumption 3 buried deep in the appendix rules this out, it should be included more prominently in the main paper with a short discussion on its relevance. In particular, how strong of an assumption it is in this multi-agent setting. For example, it rules out a very simple 2 agent, 2 action matrix game in which the payoffs are the identity matrix.

In Theorem 2, the quantifier 'locally' makes the statement rather weak. Theorem 4 (formal version of Thm 2) makes this clearer: If your current policy is already sufficiently close to the optimal one, then you'll converge.
I don't quite see the importance of this result since it feels rather vacuous. However, the pages of derivation for it suggest that it is non-trivial to prove. Some extra discussion on its purpose/importance would be welcomed. Are there analogous situations in which this type of local convergence cannot be guaranteed for instance?

For the Offline RL experiments, have you ruled out problems such as over-estimation being the cause of bad performance? In appendix G.2 when the performance of some algorithms declines over the course of training, that suggests there might be other reasons for the relatively bad performance other than the representational capacity of the methods.
Related to the experiments in Section 6.3, your theoretical results suggest that the closer to training on-policy VDN is the better/more stable it is. Are you able to empirically confirm that in the same offline RL setting, by say reporting the results of training on datasets that get increasingly on-policy?

In Appendix H, can you elaborate further on why QMIX also diverges even though it is not restricted to a linear factorisation? Is it down to the particular choice of non-linearity in the mixing network (ELU) which results in a largely linear mixing? Would using a tanh change this for example?

### Minor Points

39: Since QTRAN doesn't maintain IGM (consistency between joint and individual policies) like VDN, QMIX and QPLEX do it seems a little off to mention it like this.

339: Again I don't quite agree that QTRAN acheives Q^{IGM} like QPLEX does. Unless QTRAN's optimisation problem is appropriately solved, it has no guarantee of maintaining IGM. This is in stark contrast to QPLEX which always maintains IGM by its architecture/parametrisation of Q_{tot}.

208: Reiterate that the Q_i do not have unique solutions even though Q_tot might do. Furthermore, are you sure there is always a unique minimiser (related to the point about Prop 1 earlier)?

In Appendix G, that increasing the capacity/parameters of the networks for QMIX can't help in some situations like this has already been established in other papers on this exact matrix game (see QTRAN, MAVEN and Weighted QMIX papers).

25: factorizable -> factorized

250: Convergence can be proved in other ways without contraction arguments (see "Increasing the Action Gap: New Operators for Reinforcement Learning").

### Typos

In the statement of Theorem 4, why besides? Do you mean according to Lemma 3?

# Post-rebuttal

Thanks for your replies and for clearing up some of my misunderstandings. I'm raising my score to 6 based on these. I'm not raising it to 7 because there are quite a few areas where the presentation/discussion needs to be improved that have been noted in the reviews. Whilst the authors have been receptive to making the changes, I would want to see the revision before advocating more strongly for acceptance. I realise this is a bit unfair since this review process doesn't allow for revisions.

I also still have some lingering issues about the offline experimental results.
I'm still not entirely convinced that the offline experiments are a great way to try and demonstrate the benefits of being able to represent more. Since there are many other things that could be affecting the results such as severe over-estimation which doesn't necessarily stem from the restricted function class. Whilst you cannot rule all potential problems, making sure there isn't something obvious like overestimation happening would help (graph the max Q-values or the loss or something relevant to ease a cynical reader's mind).

The difference in performance between QTRAN and QPLEX also shows that there are important considerations that aren't captured by the analysis which leaves the reader with more questions. Arguably, properly investigating these is best left for future work since it would likely focus more on the deep learning side of things, but they should be discussed a bit more.

**Time Spent Reviewing:**

4

---

> ### Author Response · Authors · 2021-08-10
> **Response to Reviewer 1PHZ**
>
> Thanks for the comments. We provide clarification to your questions and concerns as below. If our response does not fully address your concerns, please post additional questions and we will be happy to have further discussions.
>
> **Q1: The setting of Dec-ROMDP**
>
> The reviewer's understanding is correct. MMDP is a special case of Dec-ROMDP, in which all agents receive the same observations. The formulation of Dec-ROMDP is also used by a recent work [1].
>
> We consider Dec-ROMDP since it may have potential connections to related work in function approximation theories mentioned in Appendix A, which is one of our future works. We agree with the reviewer that the presentation would be more clear using MMDP. We will refine our discussions in the next revision.
>
> [1] Mahajan, Anuj, et al. "Tesseract: Tensorised Actors for Multi-Agent Reinforcement Learning." ICML 2021.
>
> **Q2: The connection to Weighted QMIX**
>
> We thank the reviewer for reminding the connection to the analysis provided by Weighted-QMIX. The theoretical analysis in Weighted-QMIX focuses on the characterization of the function expressiveness. In comparison, this paper aims to rigorously analyze the dependency of FQI performance on the expressiveness of the value given factorization class. We do reach the similar conclusions from different aspects. e.g., our Theorem 4 indicates VDN prefers on-policy data collection, which is similar to the Idealised Central Weighting introduced by Weighted-QMIX. We will include the above discussion in the next revision.
>
> **Q3: The extra importance weight $(n-1)/n$**
>
> Consider two states $s_1$ and $s_2$. All joint actions in state $s_1$ lead to a reward signal $r_1=1$. All joint actions in state $s_2$ lead to a reward signal $r_1=2>r_1$. If we use the same credit assignment as COMA, the advantage values computed in states $s_1,s_2$ are both zero. In comparison, our solution will assign the credits by $1/n$ and $2/n$, respectively. If there is a third state $s_3$ where the agent decides on moving to $\{s_1,s_2\}$ by choosing different actions, the credit assignment used by COMA cannot figure out the differences between $s_1$ and $s_2$.
>
> **Q4: The multiple minimizers**
>
> The closed-form solution in Theorem 1 characterizes the existences of multiple minimisers as s subset of the solution space. Formally, the closed-form solution of the individual value function $Q_i^{(t+1)}(x_i,a_i)$ has a residual term $w_i(x_i)$, which can be arbitrary values satisfying $\sum_i w_i(x_i)=0$ for all data points. Note that, this residual term does not depend on the agent actions, i.e., all minimisers of the loss function correspond to the same joint value function $Q^{(t+1)}({\bf x},{\bf a})$.
>
> **Q5: The importance of Theorem 4**
>
> To address this question, we first review the results in Proposition 2. We construct an MMDP where FQI-LVF diverges to infinity from arbitrary initialization $Q^{(0)}$ using uniform data distribution. In comparison, Theorem 4 shows that, by using on-policy training, FQI-LVF can at least converge in a small region. It is true that Theorem 4 does not characterize the whole learning procedure, but it indicates an important difference between on-policy and off-policy training. We will include the above discussion in the next revision.
>
> **Q6: The divergence of QMIX**
>
> The experiments in Appendix H consider the simple two-state MMDP (see Figure 1(a)). In this simple case, the expressiveness of the neural network should be sufficient. The divergence of QMIX is caused purely by the incompleteness of the monotonic value function class, i.e., it's not a contractor as pointed by Weighted QMIX.
>
> **Q7: Differences between QTRAN and QPLEX**
>
> As mentioned by the reviewer, QTRAN and QPLEX consider different methods to implement the IGM value factorization. A well-known issue of QTRAN is that it does not guarantee the IGM consistency when its loss function has not been well optimized. This issue may become critical in online learning setting, since the violation of IGM consistency may affect the efficiency of exploration. In comparison, our experiments focus on the offline learning setting, in which the dataset is fixed and does not require exploration. We hypothesize it is the major reason why QTRAN can achieve good performance in our experiments. The remaining differences may come from many aspects, such as the implementation details of optimization, and the generalizability of different network architectures.

---

> > ### Comment · Reviewer_1PHZ · 2021-08-17
> > **Questions**
> >
> > > the credit assignment used by COMA cannot figure out the differences between $s_1$ and $s_2$.
> >
> > I don't think this is true since the critic is updated using the Q-values, and as such the Q-values (and thus the advantages when used for updating the agents) for going to $s_2$ will be higher than for going to $s_1$.
> >
> > > The closed-form solution in Theorem 1 characterizes the existences of multiple minimisers as s subset of the solution space.
> >
> > Do you mean that the argmin returns a set? I don't see how this works with the iteration.
> >
> > > all minimisers of the loss function correspond to the same joint value function
> >
> > I don't see how this is true for the simple example I gave earlier of the payoffs being the identity matrix. There are clearly 2 (symmetric) solutions to the argmin.
> >
> > > It is true that Theorem 4 does not characterize the whole learning procedure, but it indicates an important difference between on-policy and off-policy training
> >
> > This still feels like an overstatement of what Thm 4 shows. Not only do you need to be appropriately close to being on-policy (I have no issues with the $\epsilon$ having to be small), but you also need to be starting your iterations sufficiently close to your target. And so, even you're training fully on-policy Thm 4 does not prove convergence starting from an arbitrary Q.
> >
> > > The divergence of QMIX is caused purely by the incompleteness of the monotonic value function class, i.e., it's not a contractor as pointed by Weighted QMIX.
> >
> > (Please correct me if I'm wrong) I don't think you provide any evidence to justify that claim. In the particular example you used, using an ELU non-linearity for QMIX's mixing function could lead to a linear mixing (which you do prove diverges). But there is no reason you have to use an ELU for QMIX. You've shown that QMIX can diverge (which is useful and interesting). But I do not think you have sufficiently demonstrated it is because of the class of Q it can represent, as opposed to particular choices for its implementation (ELU non-linearity).
> >
> > I'd also like to ask again if you have ruled out problems in your offline RL experiments that stem from overestimation (or other issues that are not to do with the representational class of Q that can be represented), particularly since the performance of some algorithms declines over the course of training.

---

> > > ### Author Response · Authors · 2021-08-21
> > > **Response to Reviewer 1PHZ**
> > >
> > > Thanks for the responsive reply. We provide clarification to your questions as below, and we will be happy to have further discussions.
> > >
> > > **Q1: Clarification on the differences from COMA.**
> > >
> > > We agree that using COMA's advantage-based credit assignment does make sense in policy gradient methods. It is reasonable to use advantage values in policy gradient, and the COMA algorithm does not have any issues in the example we discussed in the rebuttal.
> > >
> > > Our main point is that, we cannot use COMA's credit assignment to update the Q-value function. This paper focuses on the Q-learning framework where the policy is directly derived from the maximum Q-values. Theorem 1 characterizes how linear value factorization assigns the global reward signals to individual Q-values. The extra weight $(n-1)/n$ makes the credit assignment can retain the scale information of the reward signals, which is the major difference from the advantage-based credit assignment.
> > >
> > > We will refine our discussions to avoid ambiguity.
> > >
> > > **Q2\&Q3: Clarification on the multiple minimizers.**
> > >
> > > The argmin does not return a set, because all minimizers of the loss function have the same joint value function. This is true for the reviewer's example.
> > >
> > > Consider the reviewer's example in which the payoff matrix is an identity matrix $[[1,0],[0,1]]$. The optimal joint value function computed by Theorem 1 is $[[1/2,1/2], [1/2,1/2]]$. The solution space is $Q_1(a_1)=Q_1(a_2)=1/4+w_1$ and $Q_2(a_1)=Q_2(a_2)=1/4+w_2$, where $w_1$ and $w_2$ can be arbitrary values satisfying $w_1+w_2=0$. The residual terms $\\{w_1,w_2\\}$ characterize the minimizer space of the loss function, but these terms would not affect the derived joint value function because of the linear factorization, i.e., $Q_{tot} = Q_1 + Q_2$. Please correct us if we misunderstand the question or the example.
> > >
> > > **Q4: Clarification on Theorem 4.**
> > >
> > > We agree with the reviewer's understanding. Theorem 4 only considers the Q-iteration within a local region near the optimal solution. We will refine the informal statement (Theorem 2) as follows:
> > >
> > > **Theorem 2.** *(Informal)* When the hyper-parameter $\epsilon$ is sufficiently small, on-policy FQI-LVF has at least one fixed-point Q-value that derives the optimal policy.
> > >
> > > **Q5: In the particular example you used, using an ELU non-linearity for QMIX's mixing function could lead to a linear mixing (which you do prove diverges). But there is no reason you have to use an ELU for QMIX. You've shown that QMIX can diverge (which is useful and interesting).**
> > >
> > > Excellent suggestions! We carefully check our two-state MMDP example in Figure 1(a), and find its optimal value function $Q^*$ can actually be represented by a monotonic function. Following the reviewer's suggestions, we conduct an ablation study on the activation function of QMIX. Results of this experiment show that QMIX with Tanh activation would not diverge in our example.
> > >
> > > In this ablation study, we consider three activation functions (i.e., Elu, Relu, and Tanh) and increase the depth of the mixing network to enable multiple layers of non-linear activation. Formally, we use QMIX-\{Elu/Relu/Tanh\}-$k$ to denote the network using $k$ layers of corresponding activation, i.e., $[\texttt{FC}+activation]\times k+\texttt{FC}$. The default implementation of QMIX corresponds to QMIX-Elu-1. The experiment results are presented as below.
> > >
> > > We present the infinite-norm of the learned value function. When Elu or Relu are used, the value function of QMIX rapidly diverges.
> > >
> > >
> > > | Algorithms\\Steps (Milion)   | 0M | 0.5M | 1.0M |1.5M|2.0M|
> > > |  ----  | ----  |----  | ----  |----  |----  |
> > > |**QMIX-Elu-1 (Default)**| 0.0 |485.0|1.2e+5|2.5e+7|4.0e+9|
> > > |**QMIX-Elu-2**| 0.0 |1.3e+3|3.6e+5|8.3e+7|1.6e+10|
> > > |**QMIX-Elu-3**| 0.0 |1.1e+3|2.8e+5|7.6e+7|2.0e+10|
> > > |**QMIX-Relu-1**| 0.0 |377.9|1.0e+5|1.6e+7|2.6e+9|
> > > |**QMIX-Relu-2**| 0.0 |407.8|7.4e+4|1.8e+7|2.9e+9|
> > > |**QMIX-Relu-3**| 0.0 |2.3e+3|5.4e+5|1.4e+8|3.4e+10|
> > >
> > > In comparison, using Tanh can maintain numerical stability.
> > >
> > > | Algorithms\\Steps (Milion)   | 0M | 2.5M | 5M|7.5M|10M|
> > > |  ----  | ----  |----  | ----  |----  |----  |
> > > |**QMIX-Tanh-1**| 0.0 |65.9|90.2|97.2|97.7|
> > > |**QMIX-Tanh-2**| 0.0 |69.6|88.6|97.7|98.4|
> > > |**QMIX-Tanh-3**| 0.0 |73.4|88.7|94.5|95.7|
> > >
> > > Different results with Tanh from Elu and Relu may confirm the reviewer's hypothesis that "*using an ELU non-linearity for QMIX's mixing function could lead to a linear mixing.*" We will incorporate this study in the next revision.
> > >
> > >
> > > **Q6: If you have ruled out problems in your offline RL experiments?**
> > >
> > > We cannot claim we have ruled out all potential problems, but we have made efforts to remove irrelevant factors. We would like to review several considerations in the experiment design:
> > >
> > > - **(Using Diverse Dataset)** A major source of the overestimation in offline RL is the extrapolation error caused by inadequate dataset. To address these concerns, we ensure the dataset in our experiments contains a diverse set of agent behaviors. The offline dataset is collected by training a VDN agent and adopting its full experience buffer. This buffer would include the mixture of a series of policies along the whole learning procedure and also contain the exploration trajectories.
> > >
> > > - **(Using Multiple Datasets)** To reduce the bias of performing evaluation on a specific dataset, each curve is plotted by the median performance of running with multiple random seeds on three independently collected datasets.
> > >
> > > - **(Removing Partial Observability)** We consider a modification in the SMAC environment. We concatenate the local observations with the global latent state vector to ensure the assumption of rich observations. It allows us to remove the unpredictable effects of partial observability.
> > >
> > > In addition, we note that the issues of offline training in discrete-action environments are not always as severe as that in continuous control problems. For example, offline DQN can retain 80\% of its performance in Atari games [1].
> > >
> > > In our experiments, we consider the same training setting for all evaluated algorithms to establish a fair comparison. The major difference between these algorithms is their value function class. As the results shown in the paper, QPLEX and QTRAN can generally work better than other methods, which can be regarded as a support of our theoretical implications.
> > >
> > > *References*
> > >
> > > [1] Agarwal, Rishabh, Dale Schuurmans, and Mohammad Norouzi. "An optimistic perspective on offline reinforcement learning." International Conference on Machine Learning. PMLR, 2020.

---

> > > > ### Comment · Reviewer_1PHZ · 2021-08-23
> > > > **Thanks for clarification**
> > > >
> > > > Thanks for your reply.
> > > >
> > > > > Q2&Q3: Clarification on the multiple minimizes.
> > > >
> > > > My apologies, you're right. There is indeed a unique Q-value that minimises the loss.

---

> > > > > ### Author Response · Authors · 2021-08-27
> > > > > **Thanks for your reply**
> > > > >
> > > > > Thank you very much for your feedback! We are really encouraged by your positive comment "this paper makes a valuable contribution to the literature." We hope our response has addressed your concerns. We sincerely appreciate it if you could re-evaluate the rating.
> > > > >
> > > > > Thanks again for your insightful comments and discussions.
> > > > >
> > > > > Best regards,
> > > > >
> > > > > The authors

---

> ### Author Response · Authors · 2021-09-03
> **Thanks for inspiring reviews and suggestions**
>
> We thank the reviewer for inspiring comments and increasing the score. In the revision, we will incorporate our discussions in the rebuttal to clarify the questions raised in the initial submission. We conclude a list of update changes in the global response of **Summary of the revision**.
>
> Regarding the offline experiments, we thank the reviewer for pointing out some potential challenges in offline learning. In general, the value overestimation can be caused by two sources, the interpolation source and the extrapolation source. On the interpolation side, we follow the default implementation in which double Q-learning is applied to reduce the overestimation caused by the maximum bias. On the other side, as discussed in the previous response, we have made efforts in constructing diverse datasets for reducing the overestimation caused by extrapolation error. Although we cannot eliminate the effects of offline overestimation error, all algorithms are investigated under the same training setting. Given all algorithms equally suffer from well-known offline issues, our theories indicate another factor, the function representational capacity, for illustrating the performance differences between advanced deep multi-agent Q-learning algorithms in the offline setting. We agree with the reviewer that multi-agent offline reinforcement learning is an interesting and important research topic and deserves further efforts from the community to tackle other issues, e.g., the value overestimation and distributional shift.
>
> Regarding QTRAN and QPLEX, we agree that their differences are related to deep-learning-based implementation details. In this revision, we will discuss this point as a future work.

---

### Official Review · Reviewer_z1vJ · 2021-07-18

**Rating:** 6
**Confidence:** 4

**Summary:**

This paper theoretically analyses the popular linear joint Q value factorization methods via a multi-agent fitted Q-iteration framework. The results reveal the connections between the value decomposition and counterfactual credit assignment. The authors also have discussed the impacts of on/off policy training on the value decomposition methods and conducted the experiments on SC2 micromanagement tasks to examine the theoretical results.

**Main Review:**

The main contributions of the paper are:
1. A new perspective to theoretically analyze the cooperative multi-agent Q-learning with value factorization.
2. The study on the well-known IGM condition shows its global convergence under some assumptions.

Cons:
The authors claimed that IGM value factorization can enjoy the global convergence guarantee. However, both QTRAN and QPLEX have implemented the FQI-IGM, but they have quite different performances in many experiments. Therefore, it would be better to further clarify the connection between different factorization structures and where is the performance gap comes from.

**Time Spent Reviewing:**

4

---

> ### Author Response · Authors · 2021-08-10
> **Response to Reviewer z1vJ**
>
> Thanks for the comments. We provide clarification to your questions and concerns as below. If our response does not fully address your concerns, please post additional questions and we will be happy to have further discussions.
>
> **Q1: Differences between QTRAN and QPLEX**
>
> The reviewer's understanding is correct that both QTRAN and QPLEX are deep-learning-based approaches to implementing IGM value factorization, but their implementations are structurally different. QTRAN implements the IGM consistency as a soft constraint, which is realized by minimizing a loss function, i.e., the IGM consistency does not hold when the loss has not been well optimized. QPLEX ensures the IGM consistency by a well-designed network architecture, which regards the IGM consistency as a hard constraint. In practice, such as the experiments in Figure 2, the generalization ability is also an important factor that determines the testing performance. That is why their performance differs in some cases.
>
> We will include the above discussion in the next revision.

---

### Official Review · Reviewer_ZXgm · 2021-07-20

**Rating:** 4
**Confidence:** 4

**Summary:**

The authors present a theoretical framework based on the fitted Q-iteration (FQI) method as a common tool in analyzing two popular value factorization approaches in cooperative deep multi-agent Q-learning. Linear value factorization (LVF) and Individual-Global-Max (IGM) value factorization methods are studied from the FQI perspective, and their convergence properties are explained in theoretical manner. Some didactic experiments support their findings, and numerous empirical performance figures are reported to be consistent with those in previous works as well as with the theoretical proofs delivered in the appendix.

**Limitations And Societal Impact:**

If the FQI tool is found to be unfit to extend to i) monotonic value factorization methods, ii) continuous action spaces, or iii) actor-critic settings, then the authors should acknowledge the limitations accordingly.

**Main Review:**

Originality:
To the best of my knowledge, this work is the first in attempting to find and explain the common denominator among value factorization methods in SoTA MARL works based on the FQI framework. However, the authors do not make a strong case as to why FQI is the best candidate as the "frame" in the framework. Had the paper explained how FQI topped the authors' list of viable frame candidates, it would have been very helpful in understanding the motivation behind the framework development. I would imagine that one advantage of reverting to a non-deep classical method lies in the ease of deriving a closed-form solution, but I would rather read about it in the paper instead of having to rely on my guesswork.

One strong competitor would be deep convolutional graphs (DCG), which I really think should appear much earlier in the paper than in the conclusion. Despite DCG authors never calling the work a framework, by the standards of FMA-FQI, DCG would also qualify as one, but retaining the "deep" aspect of deep MARL as well as being capable of interpolating through all VDN, QMIX, and QTRAN/QPLEX and their corresponding classes of realizable value functions: additive, monotonic, and IGM.

Quality:
The paper is generally easy to follow, but some of the important proofs feel a bit too "offloaded" to the appendix, which forces the reader to repeat "believing and moving on" until the paper's end is reached. A sketch of proofs for Theorem 1 and Proposition 1 would have made the submission much more convincing.

A minor error:
Assumption 1: decentrailized --> decentralized

Clarity:
Sections 5.2 and 5.3 remain a blur even after going through the appendix. Exactly what are meant by "offline setting" and "\epsilon"?

Are the terms "offline" and "off-policy" used interchangeably in this paper? If not, Section 5.3.2 must be renamed to "online data collection..." because the "on-policy"-ness of the data collection has nothing to do with the local convergence guarantees. (e.g., an on-policy learner could still be offline when \epsilon=1.0).

If "offline" and "off-policy" are interchangeably used, (and so are the counterparts "online" and "on-policy"), then the following questions arise.
Figure 1c must be aptly renamed, as the \epsilon=1.0 case would clearly be "off-policy".
Does the \epsilon refer to the exploratory \epsilon? If so, a completely random exploration policy does not mean off-policy.
Does the \epsilon refer to the \epsilon-greedy policy of choosing the a' in s' when updating the Q-function? If so, \epsilon=1.0 would mean completely random action chosen in the next state: how does this translate to off-policy learning? In fact, for the learning process to qualify as off-policy, shouldn't the next action be chosen greedily? (\epsilon=0).

It's very difficult to say that Section 5.2 and 5.3 are convincing when the convergence guarantees seem to have little relevance to whether the learning is on- or off-policy. Rather, they only have to do with the randomness of how the next action (a') is chosen, which certainly isn't completely random in the off-policy setting, or even in the on-policy setting, unless its current policy is a completely random one as well.

On a separate note, it remains a question why the authors chose linear and IGM value factorization methods and left out monotonic value functions. It feels like quite a stretch to claim to have proposed a "unified framework" without even hinting at the generalizability/applicability of the said framework to the "middle" case of monotonic classes of functions, which would lie somewhere between linear and IGM in terms of subspace relations.

Some minor issues:
In Figure 1b, the QTRAN graph seems to be missing; so does \epsilon=0.1 in Figure 1c.

Significance:
The submission is partially successful in presenting what it set out to. However, a number of lines of research prevent the paper from really becoming a unified framework.
- Monotonic value factorization is not treated in depth. Works following QMIX outnumber others both in quantity and in citation, so it would be hard to justify leaving them out without even a hint as to how FMA-FQI can be applied to the settings of QMIX and its variants.
- Continuous action spaces are not studied. COMIX and COVDN (Schroeder de Witt et al., 2020) are examples of value-based methods modified to work with continuous action spaces. Nowhere in the FMA-FQI paper do the authors assume discrete actions, so the continuous-action counterparts must be studied accordingly if the proposed framework were to be general.
- Actor-critic settings are not studied. FACMAC (Schroeder de Witt et al., 2020) is one example of an actor-critic model incorporating value factorization. Since the authors do not assume a purely value-based setting, actor-critic counterparts deserve some attention.

On a different note, the following statement raises a question. Section 6 "Recall that the unbounded divergence of linear value factorization is caused by the projection error induced from the limited function expressiveness of Q^{LVF}." This statement suggests that Proposition 2 relies on Proposition 1; however, the appendix shows that the proof of Proposition 1 actually relies on Proposition 2 being true (and hence the counterexample brought about by P2). In other words, the paper itself states that P2 being true is caused by P1 being true, but the appendix carries P1's proof by counterexample, which could exist only if P2 were true. The two proofs need to be refined.

**Time Spent Reviewing:**

9

---

> ### Author Response · Authors · 2021-08-10
> **Response to Reviewer ZXgm**
>
> Thanks for the comments. We provide clarification to your questions and concerns as below. If our response does not fully address your concerns, please post additional questions and we will be happy to have further discussions.
>
> **Q1: The authors do not make a strong case as to why FQI is the best candidate**
>
> We never claimed FQI is the best analytical framework.
>
> We include a discussion on the background of FQI in the related work section. We choose FQI as our analytical framework since it almost becomes the default choice in studying the single-agent deep Q-learning algorithms from the perspective of function approximation (Fu et al., 2019; Levine et al., 2020).
>
> We would appreciate it a lot if the reviewer can suggest other tools for theoretical studies on Q-learning with function approximation. It will help us to enrich our discussion in the literature background.
>
> [1] Fu, Justin, et al. "Diagnosing bottlenecks in deep q-learning algorithms." International Conference on Machine Learning. PMLR, 2019.
>
> [2] Levine, Sergey, et al. "Offline reinforcement learning: Tutorial, review, and perspectives on open problems." arXiv preprint arXiv:2005.01643 (2020).
>
> **Q2: Relation to DCG**
>
> We consider DCG as a direction for future study in the conclusion section, since DCG is a value factorization structure out of the paradigm of centralized training with decentralized execution (CTDE). As we discussed in the introduction and preliminaries, this paper focuses on factorization structures under the CTDE paradigm, in which agents cannot communicate with each other. In comparison, DCG requires a messaging-passing module to decide actions in the execution stage, since its local value function considers a group of agents.
>
> In addition, DCG can be regarded as a template for designing a series of algorithms with different graph structures, but it is not a framework for theoretical analyses. From the perspective of function approximation, the coordination graph refers to a second-order value factorization structure. We can represent the value functions of DCG by a function class $\mathcal{Q}^{\text{DCG}}$ as what we did for linear value factorization and IGM factorization, i.e., DCG can be regarded as an algorithm instance, like VDN and QPLEX, which is covered in our FMA-FQI framework.
>
> **Q3: Exactly what are meant by "offline setting" and "$\epsilon$"?**
>
> In offline settings, the training data distribution is fixed and would not change during policy learning. In section 5.3.1, we study a specific case of the offline dataset, i.e., the training data are uniformly drawn from the state-action space.
>
> The parameter $\epsilon$ refers to the hyper-parameter used by $\epsilon$-greedy. When $\epsilon$ is small, we regard the generated action distribution as on-policy (see Q4).
>
> We will include clarifications for these terms in the next revision.
>
> **Q4: Are the terms "offline" and "off-policy" used interchangeably in this paper?**
>
> The terminologies are not interchangeable. We strictly follow the instructions of Levine et al., (2020) to distinguish these words.
>
> As shown in Figure 1(c), FQI-LVF can only converge when $\epsilon$ is a small value. We regard such situations as on-policy training, since the training distribution is close to the current policy. Formally, Theorem 4 in Appendix proves that, for any MDPs, there exists a small value of $\epsilon$ such that the local convergence of FQI-LVF can be guaranteed.
>
> We will refine our discussion in the next revision.
>
> [2] Levine, Sergey, et al. "Offline reinforcement learning: Tutorial, review, and perspectives on open problems." arXiv preprint arXiv:2005.01643 (2020).
>
> **Q5: The convergence guarantees seem to have little relevance to whether the learning is on- or off-policy**
>
> To address this question, we first review the results in Proposition 2. We construct an MMDP where FQI-LVF diverges to infinity from arbitrary initialization $Q^{(0)}$ using uniform data distribution, i.e., under offline training. In comparison, Theorem 4 shows that, by using on-policy training distribution, FQI-LVF can at least converge in a small region. It is true that Theorem 4 does not characterize the whole learning procedure, but it indicates an important difference between on-policy and off-policy training.
>
> Besides, we do not understand what refers to the randomness in the reviewer's question. We would appreciate it if the reviewer can expand the question a little bit.
>
> **Q6: On the monotonic value factorization**
>
> We do not cover too much on QMIX in the main text, since we have not set up rigorous theories for the monotonic value factorization. In Appendix G and H, we establish an empirical study on its behavior in the same evaluation metric as other experiments presented in this paper. According to the experiment results, we show that the algorithmic properties of QMIX can be regarded as an interpolation between linear and IGM value factorization. Its function expressiveness is richer than that of VDN, which gives it better ability in complex domains such as StarCraft benchmark tasks. However, the monotonic function used by QMIX cannot represent all valid joint value functions. In the most unfavorable cases, QMIX may suffer from the same issue of unbounded divergence as linear value factorization (see Figure 6(b) in Appendix).
>
> **Q7: On the continuous action space and the actor-critic framework**
>
> As we state in our title, this paper focuses on the theoretical analyses of cooperative multi-agent Q-learning. We thank the reviewer to give several citations of empirical related work that studied these problems separately, but we do not think a thorough discussion over these big problems can be contained in a single conference paper. In addition, we do not agree that studies on continuous action space and actor-critic methods are critical for a theory paper on Q-learning algorithms.

---

> > ### Comment · Reviewer_ZXgm · 2021-08-31
> > **Raising the score**
> >
> > Thank you, authors, for the detailed feedback on the issues raised.
> >
> > I up my rating from 2 to 3, believing that the authors have addressed the comments and stated a further revision.
> >
> > Q1 & Q7
> > I understand that the authors do not claim that FQI is the best fit framework for the purpose of the study. Neither of the two cited works was a satisfactorily supportive clarification. As can be seen in Fu2019, there seems to be nothing about the paper being theoretical or Q-learning-focused that prevents it from discussing actor-incorporated methods. Furthermore, Fu2019's take on Q-learning with function approximation (of which FQI is a representative example) rarely encounter divergence, a contrasting claim to that of the FMA-FQI paper. If FQI is an existing framework and it is shown to rarely exhibit divergence, then perhaps arguing for the convergence of FMA-FQI merits little credit unless the challenges unique to the multi-agent setting are highlighted e.g., non-stationarity. However, such challenges seem to deserve more attention.
> >
> > The Levine2020 paper taxonomizes FQI and DQN as special cases of a generic Q-learning algorithm, so it was hard to convince myself that FQI is specifically singled out as the default choice for studying single-agent q-learning with function approximation. Even if it is, some discussion as to how that advantage carries over to the multi-agent setting is necessary, as the plurality of agents may compromise the theoretical soundness of single-agent tools.
> >
> > On a separate note, FQI and DQN being two different special cases of the generic Q-learning algorithm, and the VDN-QMIX-QTRAN line being an extension of the DQN, suggest to me that the methods named FQI-LVF and FQI-IGM are really more than variants of just the FQI, as they encompass more than the FQI aspect of the generic Q-learning to include the DQN-specific elements. This generality of FMA-FQI in my opinion is one reason for the increase in rating.
> >
> > Q3~Q5:
> > Figure 1c remains a mystery. How is it titled "On-policy FQI-LVF" if, as the authors explain, larger values of \epsilon represent off-policy cases? Are there multiple \epsilons?
> > Is the goal of Section 5.3.2 to show the convergence guarantee in on-policy data collection (as in its title) or in online data collection (as in its first paragraph), or perhaps in an on-policy AND online data collection setting? These terms need urgent clarification.
> >
> > Q2 & Q6:
> > As per the conclusion paragraph, DCG is one future direction of FMA-FQI study. According to the authors' feedback, DCG is an algorithm instance, among others such as VDN and QPLEX. I think this is all the more reason to include some analyses on either DCG or QMIX. My intuition suggests that DCG being a template from which other instances could be derived carries some interesting insights therein.

---

> > > ### Author Response · Authors · 2021-08-31
> > > **Response to Reviewer ZXgm**
> > >
> > > Thank you for your efforts and additional feedback. We are pleased that our initial response has addressed some of your concerns. However, we find you may still have misunderstandings about the literature background of FQI, and your suggestions of investigating actor-critic methods and deep coordination graphs are beyond the scope of this paper. We provide detailed clarification below and really appreciate it if you could re-evaluate our work.
> > >
> > > **(Q1\&Q7)-1: Motivation of using the FQI framework. The Levine2020 paper taxonomizes FQI and DQN as special cases of a generic Q-learning algorithm.**
> > >
> > > First, we would like to clarify the differences between FQI and DQN from the point of view of generic Q-learning. The generic Q-learning framework characterizes several components of Q-learning algorithms:
> > >
> > > - **(Function Approximation)** DQN uses a deep neural network as the function approximator. In comparison, the FQI framework does not specify the function class. In this paper, different value factorization structures correspond to different function classes, e.g., $\mathcal{Q}^{\text{LVF}}$ and $\mathcal{Q}^{\text{IGM}}$. This component is our main focus of study.
> > >
> > > - **(Data Collection)** DQN is a practical algorithm that uses a replay buffer to collect recently experienced data, i.e., using off-policy data distribution. In comparison, the standard FQI considers a fixed data distribution, i.e., an offline dataset.
> > >
> > > - **(Optimization)** DQN uses mini-batch gradient descent to optimize the loss function. In comparison, the FQI framework assumes we can find the optimal minimizer of the loss function.
> > >
> > > In summary, FQI is an analytical framework that focuses on the effects of different function classes. It simplifies the discussions on the special issues caused by exploration strategies, off-policy replay buffer, and gradient-based optimization. In this paper, we aim to analyze the algorithmic properties of multi-agent value factorization structures, which correspond to different factorized value function classes. From this perspective, FQI would be a natural choice for our analytical framework.
> > >
> > > Given that FQI is widely considered by many single-agent RL works, it is reasonable to study its variants in multi-agent RL. We are pleased that the reviewer appreciates our work as the first in attempting to analyze SoTA MARL algorithms based on the FQI framework. But we are confused why the reviewer doubts the usage of FQI.
> > >
> > >
> > > **(Q1\&Q7)-2: As can be seen in Fu2019, there seems to be nothing about the paper being theoretical or Q-learning-focused that prevents it from discussing actor-incorporated methods.**
> > >
> > > We agree that investigating actor-critic would be valuable and interesting, but it is outside the scope of this paper.
> > >
> > > In some simplified situations, we can see some connections between Q-learning and actor-critic methods. e.g., in the discussion of Fu et al. (2019), the actor function is an approximation of $\arg\max_a Q(s,a)$ (see section 3.4 of Fu et al. (2019)), i.e., they do not consider the policy gradient component of the actor-critic framework. When assuming the actor approximates the maximum-value action, there are almost no differences between Q-learning and actor-critic methods. However, such analyses are direct extensions and do not provide special insights in the point of view of actor-critic methods.
> > >
> > > In addition, Levine et al. (2020) mentioned that "*actor-critic algorithms are closely related with another class of methods that frequently arises in dynamic programming, called policy iteration.*" That is why Levine et al. (2020) discussed actor-critic as a parallel framework to fitted Q-iteration.
> > >
> > > **(Q1\&Q7)-3: Fu2019's take on Q-learning with function approximation (of which FQI is a representative example) rarely encounters divergence.**
> > >
> > > The reviewer may miss an important condition of this claim in Fu et al. (2019). In section 1 of Fu et al. (2019), they claimed that "*We find, somewhat surprisingly, that divergence rarely occurs, and that function approximation error is not a major problem in Q-learning algorithms **when the function approximator is powerful**.*" The representational capacity of the function class is a critical factor to the convergence of Q-learning algorithms. Single-agent FQI may suffer from divergence when using a poor function approximator (see appendix B.2 of Fu et al. (2019) and section 11.3 of Sutton \& Barto (2018)). DQN hardly diverges in practice since the neural network is sufficiently powerful.
> > >
> > > **(Q1\&Q7)-4: Perhaps arguing for the convergence of FMA-FQI merits little credit unless the challenges unique to the multi-agent setting are highlighted e.g., non-stationarity.**
> > >
> > > The divergence of FQI-LVF is caused by the value factorization rather than non-stationarity, which is a special technique of cooperative multi-agent Q-learning.
> > >
> > > As discussed in the previous question, the stability of fitted Q-iteration relies on a powerful function approximator. In the single-agent case, a deep neural network can effectively approximate the value function. In comparison, when using linear value factorization in multi-agent Q-learning, the factorized function class cannot represent all possible value functions, i.e., all mappings from the joint state-action space to the value. In this paper, we show that the poor representational capacity of linear value factorization may lead to divergence in offline training, and using on-policy data can resolve this issue.
> > >
> > > Note that our work focuses on the current popular multi-agent learning paradigm of Centralized Training with Decentralized Execution (CTDE), which avoids the non-stationarity issue.
> > >
> > > **(Q3-Q5): Clarification on section 5.3.2.**
> > >
> > > We thank the reviewer to point out the mistake. The term "online" in the first paragraph of section 5.3.2 should be "on-policy". We consider online data collection only in deep MARL experiments.
> > >
> > > Section 5.3.2 considers a special variant of FQI where the data distribution is not fixed. Formally, at each iteration, the dataset is re-collected by an $\epsilon$-greedy policy. We name this case "on-policy" since the data distribution is determined by the current policy (assume $\epsilon<1$) and the most relevant setting is $\epsilon\to 0$.
> > >
> > > Figure 1c presents an experiment that verifies the claims of Theorem 2. It shows that a small value of $\epsilon$ can prevent FQI-LVF from divergence. To remove ambiguity, we would modify the caption of Figure 1c to "comparing different choices of $\epsilon$".
> > >
> > > **(Q2\&Q6): DCG is one future direction of FMA-FQI study.**
> > >
> > > We agree that DCG is quite valuable for future studies, but it is outside the scope of this paper. As we discussed before, coordination through message-passing is one of the major components of the DCG algorithm. This communication structure is beyond the CTDE paradigm and quite different from the focus of this paper.
> > >
> > > *References*
> > >
> > > [1] Fu, Justin, et al. "Diagnosing bottlenecks in deep q-learning algorithms." International Conference on Machine Learning. PMLR, 2019.
> > >
> > > [2] Levine, Sergey, et al. "Offline reinforcement learning: Tutorial, review, and perspectives on open problems." arXiv preprint arXiv:2005.01643 (2020).
> > >
> > > [3] Sutton, Richard S., and Andrew G. Barto. Reinforcement learning: An introduction. MIT press, 2018.

---

> > > > ### Comment · Reviewer_ZXgm · 2021-09-01
> > > > **Further incrementing the score**
> > > >
> > > > Thank you, authors, for the clarification and feedback.
> > > >
> > > > I up my rating a further one point from 3 to 4. Terms on/offline, on/off-policy, and \epsilon have all been clarified and corrected where necessary, and the representational capacity limitations of FQI (that render convergence a problem of prime importance) have been laid out in contrast to another variant of the generic Q-learning.
> > > >
> > > > At this point, I am not all-in on rejecting the paper, but I believe that a future version of the paper having addressed the issues I have raised will enrich and consolidate the MARL literature much better than the current submission would.
> > > > I think I should leave it to the collective decision of the other reviewers and the area chair since I have sensed that the authors and I have come to an unbridgeable gap:
> > > >
> > > > I have to say that my biggest question mark lingers on the authors' stance on actor-critic methods and DCG.
> > > >
> > > > It is known that one of the strongest motivators for devising actor-critic methods is to handle continuous control problems. A SMAC environment poses several factors that hinder the convergence of agents: continuity of the space, agent plurality, non-stationarity, etc. Studying actor-critic methods through the lens of the FQI framework would be enriching to find out i) whether the achieved convergence properties of FMA-FQI are of the same nature as those of actor-critic methods, in which case their final performance (win rate in SMAC) would be similar between FMA-FQI and actor-critic (FMA-actor_critic, perhaps) OR ii) whether the achieved convergence properties of FMA-FQI is an alternative way to approach convergence than that of actor-critic methods, in which case FMA-actor_critic would converge faster or perform better (since it is benefitting from both its actor-critic structure and its "FQI-ness"). As the authors quote, actor-critic methods do have a policy iteration component, but they do not rid of the value iteration component, which is why I think that comparing the merits of actor-critic methods and those of FMA-FQI side by side in a multi-agent continuous control task is necessary.
> > > >
> > > > As to DCG, if DCG is a future study direction, and it is an algorithm instance just like the others (QPLEX, VDN) encompassed by Q^{QPLEX, VDN} or some other Q^{some algorithm}, then it probably deserves slightly more attention. However, I also agree with the authors that DCG being a CTDE work should be positioned separately. This contradiction forces me to wonder whether DCG actually qualifies at all as a future study direction of FQI because continuing/extending a no-communication algorithm at the execution time (such as FMA-FQI) will certainly not evolve the work into a message-passing algorithm. They really might just belong in two different lines of research, in which case defining a Q^{DCG} in terms of the FMA-FQI framework, even if it were possible, would not further credit the generality of FMA-FQI.

---

> > > > > ### Author Response · Authors · 2021-09-01
> > > > > **Response to Reviewer ZXgm**
> > > > >
> > > > > We thank the reviewer for additional feedback and increasing the score. We are very pleased that most questions and concerns about FQI are well addressed. We also appreciate the reviewer now being open to the decision on this work.
> > > > >
> > > > > Regarding the remaining question, actor-critic and DCG algorithms, we agree with the reviewer these topics have generally strong impacts on the MARL community. However, as mentioned by the reviewer, both these methods have special components that should be positioned separately. Deriving complete and rigorous theoretical analyses for these methods requires resolving a series of technical challenges outside the focused scope of this paper.
> > > > >
> > > > > Formally, discussing actor-critic and DCG requires extending FMA-FQI to include more algorithmic components, such as the actor function and message-passing. In this paper, we focus on FQI since our main purpose is to analyze value factorization from the point of view of function approximation. Note that, when function approximation becomes the focus, many prior works would choose FQI as the analytical framework to make the underlying insights more accessible (see lines 134-135), rather than always consider a more general and complex framework such as generic Q-learning. That is why we leave actor-critic and DCG as future work rather than the direct extensions of this paper.
> > > > >
> > > > > In the next revision, we will enrich the discussions on the empirical connections between our current results and the related literature (including actor-critic and DCG). We will also pay special attention to refining the usage of terminologies we discussed in the rebuttal (e.g., on-policy vs. online).
> > > > >
> > > > > Thank you again for taking time and efforts in engaging discussions!
> > > > >
> > > > > Best,
> > > > >
> > > > > The Authors

---

### Author Response · Authors · 2021-09-03
**Summary of the revision**

Thank AC for leading the discussions! Thank all the reviewers for the constructive comments and discussions! We are encouraged all the reviewers found our work making valuable contributions to theoretical analyses on factorized multi-agent Q-learning. We are also pleased that our responses have addressed reviewers' concerns. We will incorporate reviewers' suggestions and our rebuttal responses into the final version to improve the presentation of this paper. Here is a brief summary of the major updates we will make in the revision.

- To address the concerns of Reviewer ZXgm, we will enrich the discussions of the literature background of fitted Q-iteration (FQI). In addition, we will discuss the empirical connections between our FMA-FQI formulation and other related methods, such as actor-critic and deep coordination graphs.

- We will include a simplified statement of Theorem 1 under the notations of MMDP to make the intuition more accessible. Note MMDP is a special case of the Dec-ROMDP setting discussed in the paper.

- We will refine the statement and discussions of Theorem 2 as presented in the responses to Reviewer 1PHZ and ZXgm. We will also discuss some connections to the results in the Weighted QMIX paper as suggested by Reviewer 1PHZ.

- We will include a clarification of the offline experiment setting as what we presented in the response to Reviewer 1PHZ. We will also remark on the differences between QTRAN and QPLEX.

- In Appendix, we will incorporate additional experiments results on QMIX (presented in the response to Reviewer 1PHZ) and discuss the implementation subtleties of the selection of activation functions.

To avoid ambiguities, we will also correct the usage of some terminologies as suggested by reviewers.

- We will arrange lines 205-208 to a new paragraph to highlight the discussion of multiple loss-minimizers.

- In the discussion of the extra importance weight $(n-1)/n$ in counterfactual credit assignment, we will focus on the discussions on value-based MARL, since we do not consider the policy gradient component of COMA.

- In section 5.3.2, we will correct the term "online" to "on-policy" at line 273. We will also modify the caption of Figure 1c as suggested by Reviewer ZXgm.

Thank AC and all the reviewers again for their time and efforts in reviewing and discussing our work. Any additional suggestions or comments are well appreciated.


Best,

The authors

---

### Decision · Program_Chairs · 2021-09-27

**Decision:**

Accept (Poster)

**Comment:**

The reviewers and AC discussed the paper. There was agreement that the approach leads to useful insight about value function decomposition in MARL. The author response was also very helpful in clarifying several points about the paper. Still, a number of improvements need to be made. Some of these updates have already been discussed in the author response (e.g., clarifications to the theory and fitted Q-iteration, discussion of QTRAN and QPLEX), but the authors should thoroughly update the paper as suggested by the reviewers.